# Conversational content is organized across multiple timescales in the brain

Masahiro Yamashita[1,2], Rieko Kubo[2,3] & Shinji Nishimoto [1,2] ✉

The evolution of conversation facilitates the exchange of intricate thoughts and emotions. The meaning is progressively constructed by integrating both produced and perceived speech into hierarchical linguistic structures across multiple timescales, including words, sentences and discourse. However, the neural mechanisms underlying these interactive sense-making processes remain largely unknown. Here we used functional magnetic resonance imaging to measure brain activity during hours of spontaneous conversations, modelling neural representations of conversational content using contextual embeddings derived from a large language model (GPT) at varying timescales. Our results reveal that linguistic representations are both shared and distinct between production and comprehension, distributed across various functional networks. Shared representations, predominantly localized within language-selective regions, were consistently observed at shorter timescales, corresponding to words and single sentences. By contrast, modality-specific representations exhibited opposing timescale selectivity: shorter for production and longer for comprehension, suggesting that distinct mechanisms are involved in contextual integration. These findings suggest that conversational meaning emerges from the interplay between shared linguistic codes and modality-specific temporal integration, facilitating context-dependent comprehension and adaptive speech production.

Humans can share complex thoughts and emotions through conversation, from casual greetings to formal discussions. Successful communication assumes a shared understanding of the context and goals, which provide a framework for what is talked about (content) and how the conversation unfolds (process)[1]. Conversational content can be conveyed not only through linguistic alignment between interlocutors[2–4] but also through extralinguistic abilities such as social cognition, world knowledge and situation modelling[5–7]. By contrast, the process of conversation requires temporal alignment, enabling the seamless alternation between language production and comprehension[8,9]. Back-channel responses (for example, 'yeah') and fillers (for example, 'uh') play critical roles in reinforcing shared understanding and facilitating

the coordination of speech planning and listening[10–12]. Moreover, conveying complex ideas often relies on structured narratives, which form the foundation of effective knowledge sharing[13]. Thus, conversation emerges as a multidimensional (linguistic and extralinguistic) and multitimescale joint activity involving the intricate interplay of language production and comprehension.

The neural underpinnings of language processing have been extensively investigated using functional magnetic resonance imaging (fMRI), particularly in the context of naturalistic narrative comprehension. Comprehension can be viewed as the transformation of low-level sensory inputs (for example, speech sounds and written text) into high-level hierarchical linguistic structures[14–20]. These

[1]Graduate School of Frontier Biosciences, The University of Osaka, Suita, Japan. [2]Center for Information and Neural Networks (CiNet), National Institute of Information and Communications Technology (NICT), Suita, Japan. [3]Graduate School of Medical and Dental Sciences, Institute of Science Tokyo, Tokyo, Japan. ✉e-mail: nishimoto.shinji.fbs@osaka-u.ac.jp

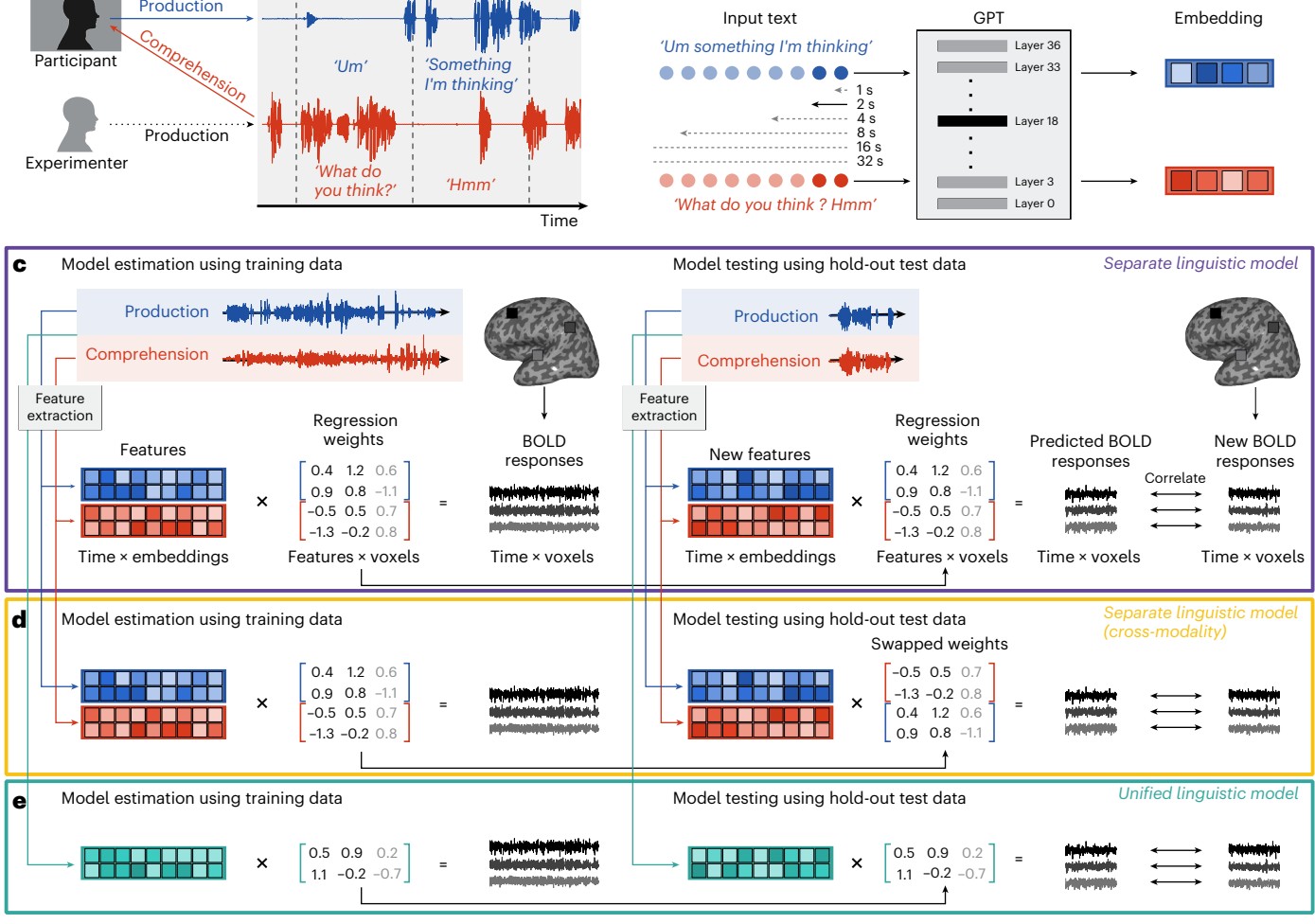

**Fig. 1 | Dialogue experiment and voxel-wise encoding models for predicting brain activity from contextual linguistic embeddings. a**, Participants ($N = 8$) engaged in natural conversations on topics provided by the experimenter while undergoing fMRI. Speech was transcribed and temporally segmented for each fMRI TR (1,000 ms intervals). **b**, Utterances were concatenated with prior context (1–32 s) and processed by an instruction-tuned GPT model. The embeddings were extracted from layers in multiples of three for subsequent analysis. **c**, Voxel-wise encoding modelling in the separate linguistic model. The GPT embeddings were modality specific, combined across modalities and utilized to train a FIR regression model, using banded ridge regularization for each voxel. The prediction performance was evaluated by correlating predicted and observed BOLD responses on held-out test data. **d**, The cross-modality prediction. The model weights were exchanged between modalities, and the prediction performance was assessed. **e**, The unified linguistic model. The GPT embeddings were extracted uniformly from concatenated content spanning both modalities.

structures are created by integrating linguistic information—both semantic and syntactic—across multiple timescales, ranging from word-level semantics to sentence meaning, ultimately culminating in a coherent narrative[14–24]. While these studies have highlighted the brain's capacity to encode hierarchical linguistic structures across various sensory modalities (for example, listening and reading) and timescales (words, sentences, discourse and narrative arcs), the functioning of these mechanisms during real-time conversations remains largely unexplored.

In this study, we address this gap by investigating the neural representation of conversational content, focusing on two key hypotheses. The first posits the existence of a unified linguistic representation that integrates information across production and comprehension. This hypothesis aligns with evidence of shared neural representations for semantic and syntactic information across modalities[25–29]. The second proposes distinct representations for production and comprehension, enabling differentiation between self-generated speech and that of an interlocutor[30]. This dual representation aligns with findings of

modality-specific syntactic processes[31] and mechanisms that facilitate concurrent speech planning during listening[32], along with the predictive coding of an interlocutor's speech[20,23].

To test these hypotheses, we used fMRI during natural conversations and utilized voxel-wise encoding modelling[33,34] to characterize neural representations of conversational content across multiple timescales, ranging from 1 to 32 s. Our analyses addressed two key questions: (1) Are linguistic representations shared between production and comprehension? (2) Are certain linguistic representations uniquely associated with production, comprehension or both? Throughout the study, the term linguistic encompasses both semantic and syntactic dimensions, reflecting their tight interdependence[35,36].

## Results

### Shared representations between production and comprehension

We collected fMRI data from eight native Japanese speakers during spontaneous conversations with an experimenter (Fig. 1a). Each

participant talked about 27 topics across 27 runs (Supplementary Table 5), with each run lasting 7 min and 10 s, resulting in approximately 3 h of data per participant over three to four sessions. Despite efforts to balance the amount of speech production and comprehension, individual variation in speech samples was observed (Supplementary Fig. 1). Rigorous preprocessing minimized potential confounds from motion artifacts and peripheral articulatory and auditory processes, effectively isolating blood-oxygen-level-dependent (BOLD) responses associated with higher-level linguistic representations (Supplementary Figs. 2–5 and Supplementary Results).

We used voxel-wise encoding modelling, utilizing GPT embeddings extracted from conversational content as linguistic features (Fig. 1b). The transcriptions were transformed into contextual embeddings using an instruction-tuned GPT model[37] that was fine-tuned for interactive language tasks. This GPT model comprises an input embedding layer and 36 transformer layers, each containing 2,816 hidden units. We extracted embeddings from 13 hierarchical layers (the input layer and every third transformer layer) across six context lengths (1, 2, 4, 8, 16 and 32 s). We averaged the embeddings across all tokens within each segment (or fMRI volume, repetition time (TR) of 1,000 ms), resulting in 78 feature combinations per modality (13 layers × 6 context lengths). These features were integrated into a joint model, referred to as the separate linguistic model, with 5,632 features (2 modalities × 2,816 features). A finite impulse response (FIR) model was used to predict BOLD responses with delays ranging from 2 to 7 s (5,632 features × 6 delays = 33,792 features). This joint model was fit to BOLD responses using banded ridge regression[38,39] for each voxel. We used the leave-one-session-out cross-validation to train the model on $N$-1 sessions and test it on one session. The prediction accuracy was evaluated using Pearson's correlation coefficients between observed and predicted BOLD responses. To account for autocorrelation, the data were divided into 20-s blocks, permuted to estimate the null distribution, and the correlations were calculated across 1,000 permutations to obtain $P$ values. The reported prediction accuracy is derived from averaging across cross-validation folds, and the combined $P$ values were calculated using Fisher's method.

The separate linguistic model achieved good prediction accuracy across extensive cortical regions. For example, in participant P7, embeddings derived from an 8-s context at layer 18 exhibited high prediction accuracy in the bilateral prefrontal, temporal and parietal cortices (Fig. 2a and see Supplementary Figs. 6 and 9 for results from individual layers and participants). To investigate whether the average prediction accuracy across the cortex was influenced by layer position and context length, we conducted a linear mixed-effects (LME) model analysis. The participants were specified as random effects, permitting variations in the effects of context length and its squared term across participants (Methods). We found an inverted U-shaped relationship across timescales (context length: $t(7) = 4.41$, $P = 0.0031$, $\beta = 0.74$, 95% confidence interval (CI) 0.39 to 1.08; context length squared: $t(7) = -4.54$, $P = 0.0027$, $\beta = -0.43$, 95% CI −0.63 to −0.24, summarized in Supplementary Table 1) and across layers (layer position: $t(597) = 11.36$, $P < 0.001$, $\beta = 0.15$, 95% CI 0.12 to 0.17; layer position squared: $t(597) = -17.95$, $\beta = -0.26$, $P < 0.001$, 95% CI −0.29 to −0.23). In addition, we found a significant interaction effect between context length and layer position ($t(597) = -6.95$, $P < 0.001$, $\beta = -0.09$, 95% CI −0.11 to −0.06).

Next, we addressed the question of how neural linguistic representations are shared between production and comprehension. To evaluate this, we assessed cross-modality prediction accuracy by interchanging the model weights between production and comprehension (Fig. 1d). This analysis was restricted to 'linguistic voxels,' defined as those exhibiting significant prediction accuracy in the separate linguistic model. We found a notable reduction in cross-modality prediction accuracy (Fig. 2b and see Supplementary Figs. 7 and 9 for individual layers and participants), consistent with findings from an

LME model (actual model: $t(7) = 7.05$, $P < 0.001$, $\beta = 0.057$, 95% CI 0.040 to 0.074). Significantly predicted voxels were scattered across prefrontal, temporal, parietal and occipital cortices (Fig. 2a). Notably, cross-modality prediction accuracy increased with longer context lengths as revealed by significant fixed effects for both context length ($t(7) = 4.01$, $P = 0.0052$, $\beta = 0.40$, 95% CI 0.15 to 0.66). These findings suggest that while linguistic representations are partially shared between production and comprehension, the topographic organization is modulated by the timescales.

Although these results demonstrated generalizable linguistic representations across modalities, another critical question arises: is a unified linguistic representation sufficient for accurate predictions? To address this, we developed a unified linguistic model that extracted GPT embeddings from combined transcripts (Fig. 1e). Compared with the separate linguistic model, the unified linguistic model showed a slight reduction in prediction accuracy (Fig. 2b and see Supplementary Figs. 8 and 9 for individual layers and participants), as indicated by a significant fixed effect of model type (separate linguistic model: $t(1227) = 4.01$, $P < 0.001$, $\beta = 3.4 \times 10^{-3}$, 95% CI $2.7 \times 10^{-3}$ to $4.1 \times 10^{-3}$). Further LME analysis of the unified linguistic model revealed significant effects of layer position (layer position: $t(597) = 7.45$, $P < 0.001$, $\beta = 0.08$, 95% CI 0.06 to 0.11; layer position squared: $t(597) = -22.79$, $P < 0.001$, $\beta = -0.29$, 95% CI −0.31 to −0.26) and its interaction with context length ($t(597) = -9.45$, $P < 0.001$, $\beta = -0.11$, 95% CI −0.13 to −0.08), whereas no significant fixed effect of context length was detected. These findings suggest that the lower prediction performance of the unified linguistic model, relative to the Separate Linguistic model, may be attributable to its limited ability to leverage longer contextual information to enhance predictions.

Next, we quantified the similarities in the linguistic representations across modalities. Given the considerably decreased prediction accuracy from same-modality to cross-modality predictions, we hypothesized that these voxels might exhibit similar yet unique linguistic tuning for each modality (that is, a weak positive correlation). Here, we focus on 'cross-modal voxels' that demonstrated robust prediction in both same- and cross-modality conditions ($r > 0.05$). The weight correlation was calculated using Pearson's correlation coefficient for the separate linguistic model weights for each voxel. Cross-modal voxels exhibited moderately positive correlations across layers and timescales (Fig. 2d and see Supplementary Fig. 11 for individual layers and participants) and showed higher weight correlations compared with linguistic voxels (cross-modal voxels: $t(7) = 11.35$, $P < 0.001$, $\beta = 0.20$, 95% CI 0.17 to 0.24). Notably, positively correlated voxels clustered within the prefrontal, temporal and parietal cortices at shorter timescales (1–4 s) (Fig. 2c and see Supplementary Fig. 14 for individual participant data). By contrast, at longer timescales (16–32 s), positively correlated voxels were more diffusely distributed and idiosyncratic among participants. These findings suggest that while cross-modal voxels may share some aspects of linguistic representation, they also exhibit unique tuning across modalities.

## Modality-specific timescale selectivity

We next explored modality-specific linguistic representations by fitting the production-only and comprehension-only linguistic models. These models utilized modality-specific contextual embeddings to quantify the variance in BOLD responses that could be uniquely attributed to each modality. Variance partitioning[16,40] was used to assign variance to either production or comprehension using the following equations

$$\text{Production} \backslash \text{comprehension} = \text{production} \cup \text{comprehension}$$
$$- \text{comprehension}$$

$$\text{Comprehension} \backslash \text{production} = \text{production} \cup \text{comprehension}$$
$$- \text{production}.$$

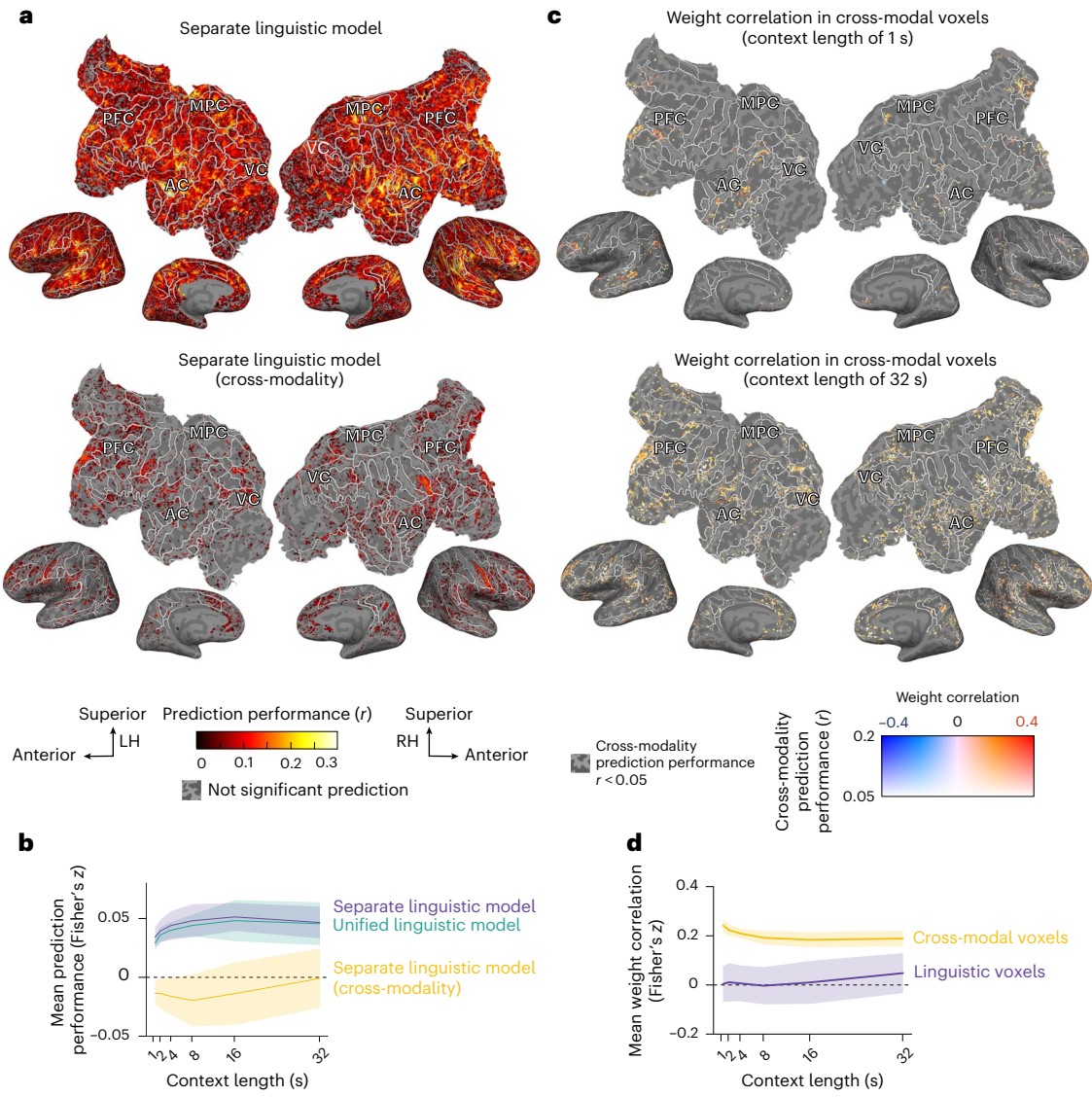

**Fig. 2 | Shared linguistic representations between production and comprehension. a**, The prediction accuracy of the separate linguistic model is visualized on the flattened cortical surface of one participant under a specific condition (P7, context length of 32 s, layer 18) (see Supplementary Figs. 6 and 9 for individual layers and participants). The voxels with significant prediction accuracy (one-sided permutation test, $P < 0.05$, FDR corrected) are displayed. PFC, prefrontal cortex; MPC, medial parietal cortex; AC, auditory cortex; VC, visual cortex. **b**, The mean prediction performance across participants, averaged over voxels and layers. The cross-modality prediction was significantly less accurate than same-modality predictions (actual model: $t(7) = 7.05$, two-sided parametric test, $P = 2.0 \times 10^{-4}$, $\beta = 0.057$, 95% CI 0.040 to 0.074). **c**, Cross-modal

voxel weight correlations at two context lengths (1 and 32 s), shown for one participant under a specific layer condition (P7, layer 18) (see Supplementary Fig. 14 for individual participant data). Only the voxels with prediction accuracy above 0.05 in both actual and cross-modality conditions are shown. **d**, The mean weight correlations across participants, highlighting linguistic (purple) and cross-modal (yellow) voxels. The cross-modal voxels exhibited significantly higher weight correlations (cross-modal voxels: $t(7) = 11.35$, two-sided parametric test, $P = 9.2 \times 10^{-6}$, $\beta = 0.20$, 95% CI 0.17 to 0.24). The shaded regions in **b** and **d** represent the standard deviation across participants. LH, left hemisphere; RH, right hemisphere.

We found that production explained more variance at shorter timescales (1–4 s) (Fig. 3a and see Supplementary Fig. 10 for individual layers and participants). The LME analysis revealed significant effects of context length ($t(7) = -3.87$, $P = 0.0061$, $\beta = -0.40$, 95% CI −0.62 to −0.18), layer position (layer position: $t(597) = 3.72$, $P < 0.001$, $\beta = 0.049$, 95% CI 0.023 to 0.075; layer position squared: $t(597) = -8.82$, $P < 0.001$, $\beta = -0.13$, 95% CI −0.16 to −0.10) and their interaction ($t(597) = -5.27$, $P < 0.001$, $\beta = -0.070$, 95% CI −0.096 to −0.044). By contrast, comprehension explained more variance at longer timescales (16–32 s). The LME analysis indicated an inverted U-shaped relationship for context length (context length: $t(7) = 4.55$, $P = 0.0026$, $\beta = 1.02$, 95% CI 0.55 to 1.49; context length squared: $t(7) = -2.42$, $P = 0.046$,

$\beta = -0.41$, 95% CI −0.75 to −0.06) and layer position (layer position: $t(598) = 8.33$, $P < 0.001$, $\beta = 0.11$, 95% CI 0.08 to 0.13; layer position squared: $t(598) = -11.57$, $P < 0.001$, $\beta = -0.16$, 95% CI −0.19 to −0.13). Across participants, the context length that maximized prediction accuracy consistently varied between modalities, with production peaking at shorter timescales and comprehension at longer timescales (Fig. 3b). These findings suggest distinct timescale selectivity for production and comprehension.

To further investigate modality-specific timescale selectivity, we compared the weights of the unified linguistic model weights to those of the separate linguistic models by calculating voxel-wise weight correlations. For production, the LME analysis revealed a

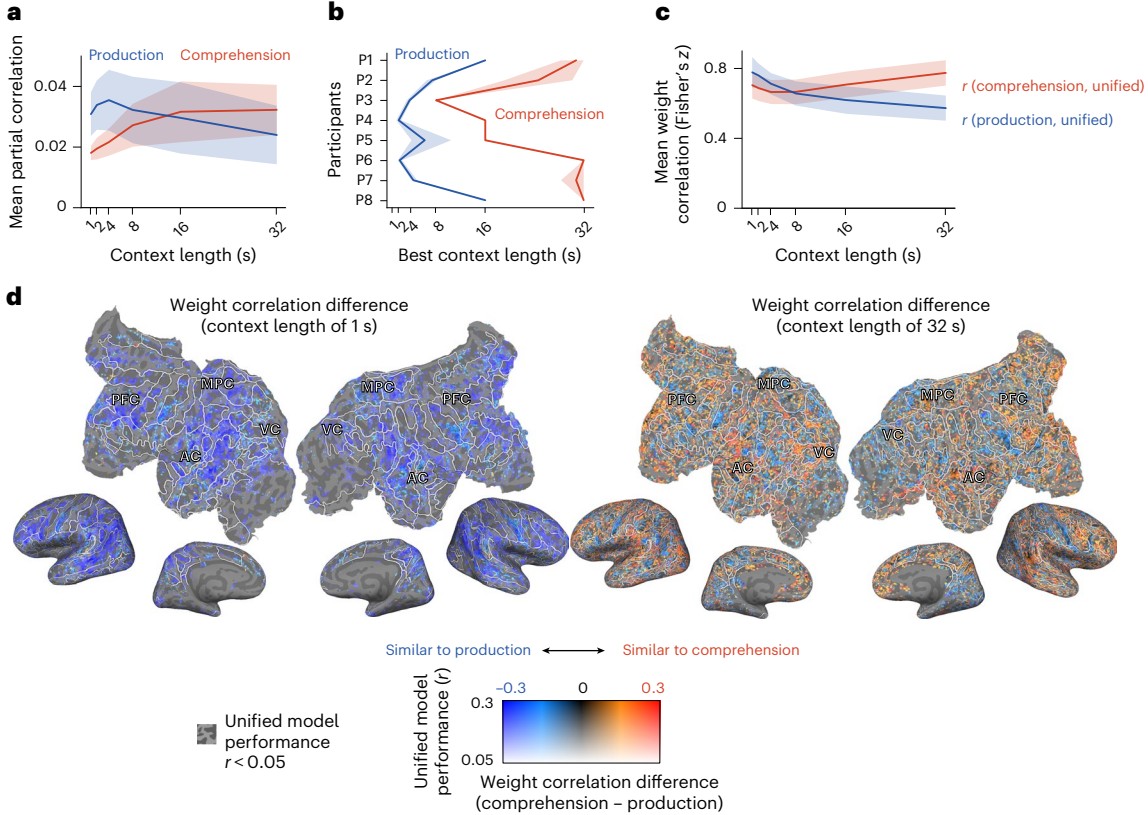

**Fig. 3 | Timescale selectivity for short-context in production and long-context in comprehension. a**, The mean variance (across voxels and layers) uniquely explained by production or comprehension, averaged across participants. **b**, The mean context lengths (across layers) maximizing unique variance explained for each modality and participant. **c**, The mean weight correlation (across voxels and layers) between the unified and separate linguistic models, calculated for each modality and averaged across participants. **d**, The changes in weight correlations between the unified and separate linguistic models for two context lengths (1 and 32 s) on the flattened cortical surface of one participant under one condition (P7, layer 18) (see Supplementary Fig. 15 for individual participant data). The voxels with good unified model prediction accuracy ($r \geq 0.05$) are shown. PFC, prefrontal cortex; MPC, medial parietal cortex; AC, auditory cortex; VC, visual cortex. The shaded areas in **a**–**c** represent the standard deviation across layers.

U-shaped relationship across timescales (context length: $t(7) = -8.59$, $P < 0.001$, $\beta = -1.01$, 95% CI −1.26 to −0.77; context length squared: $t(7) = 5.59$, $P < 0.001$, $\beta = 0.32$, 95% CI 0.20 to 0.44) (Fig. 3c and see Supplementary Fig. 12 for individual layers and participants). A similar U-shaped relationship was observed for comprehension (context length: $t(7) = 0.70$, $P = 0.50$, $\beta = 0.12$, 95% CI −0.23 to 0.47; context length squared: $t(7) = 3.30$, $P = 0.013$, $\beta = 0.23$, 95% CI 0.08 to 0.37). The weights of the unified linguistic model were more closely aligned with production at shorter timescales, while they resembled comprehension at longer timescales (Fig. 3d and Supplementary Fig. 12 for individual layers and participants). These results underscore modality-specific timescale selectivity, with production favoring shorter contexts and comprehension benefiting from longer contexts.

To mitigate potential biases in variance partitioning results due to disparities in sample sizes, we examined the correlation between production-to-comprehension sample size ratios and the corresponding variance explained. A significant correlation was observed for early layers at a 1-s context length, with Spearman's rank correlation rho of 1.00 for layer 0 and 0.93 for layer 3 ($P < 0.05$, false discovery rate (FDR) corrected). The participants who produced more speech demonstrated greater variance explained by production under these conditions. Importantly, the sample proportions were balanced overall, with four participants producing more speech (P3, P4, P6 and P7) and the remaining four comprehending more (P1, P2, P5 and P8). These findings confirm that variance partitioning results were not systematically biased towards either modality across participants.

## Dual linguistic representations in bimodal voxels

After analysing the variance uniquely explained by production and comprehension, we investigated the shared variance explained by both modalities. The shared variance was calculated as follows

$$\text{Production} \cap \text{comprehension} = \text{production} + \text{comprehension}$$

$$-\text{production} \cup \text{comprehension}.$$

We found that shared variance increased progressively with longer context lengths, peaking at an average of 8 s (Fig. 4a and see Supplementary Fig. 10 for individual layers and participants). Notably, for all participants, the context length that maximized shared variance exceeded 8 s (Fig. 4b). The LME analysis revealed an inverted U-shaped relationship across timescales (context length: $t(7) = 4.55$, $P < 0.001$, $\beta = 1.02$, 95% CI 0.68 to 1.36; context length squared: $t(7) = -4.59$, $P = 0.0025$, $\beta = -0.49$, 95% CI −0.71 to −0.27). A significant quadratic effect of layer position was also observed (layer position squared: $t(597) = -12.30$, $P < 0.001$, $\beta = -0.20$, 95% CI −0.23 to −0.17).

To map the topographic organization of selectivity to a single modality or shared between both, we created cortical maps depicting the patterns that explained the largest variance for each voxel. These maps revealed that voxels with the largest shared variance, hereafter referred to as 'bimodal voxels,' were distributed across various cortical regions (Fig. 4c and see Supplementary Fig. 16 for individual layers and participants). Notably, contextual information spanning 8 s or longer appeared to drive substantial bimodal responses, suggesting the integration of linguistic information across modalities.

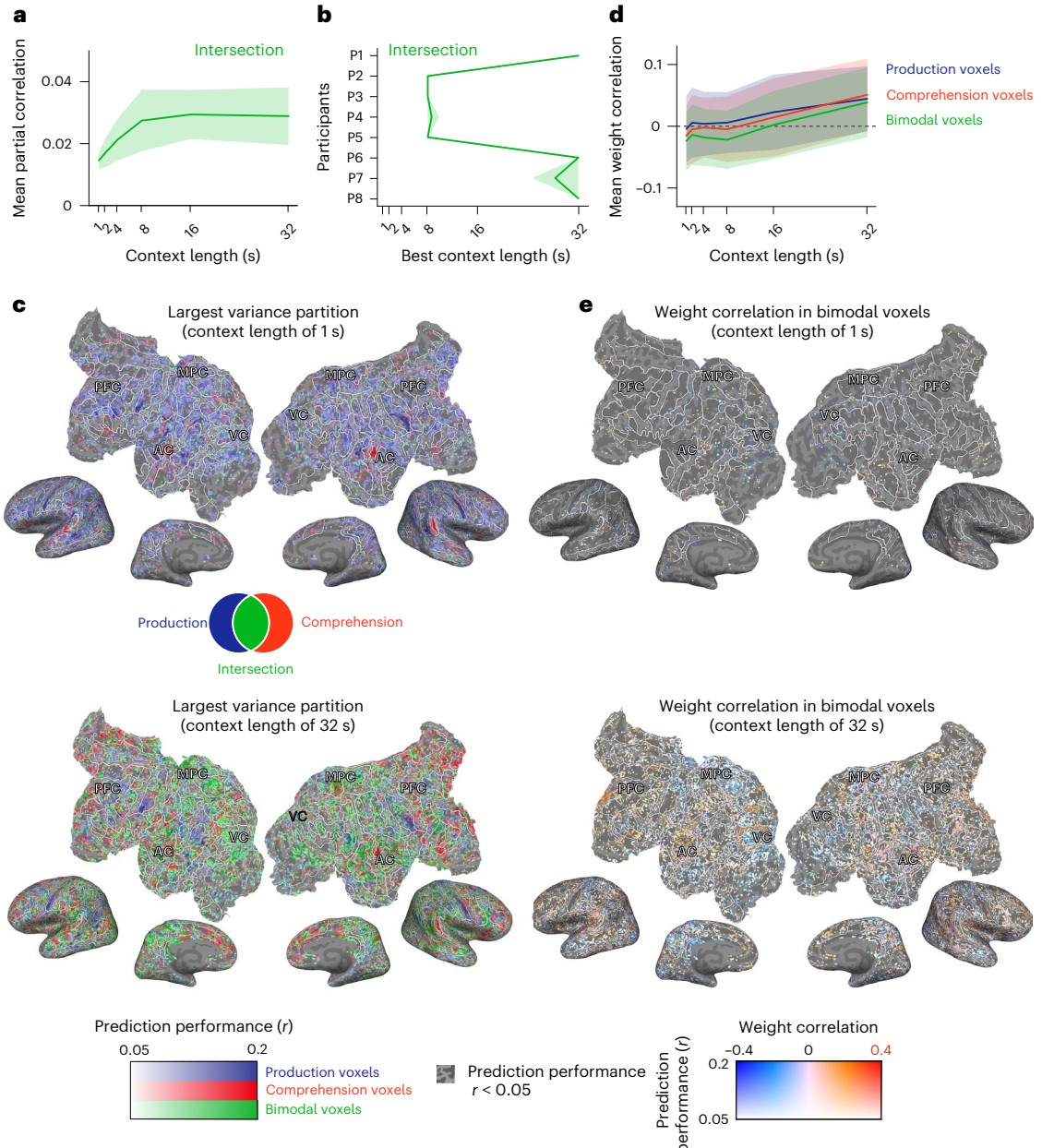

**Fig. 4 | Dual linguistic representations in bimodal linguistic voxels. a**, The mean variance (across voxels and layers) uniquely explained by the intersection of production and comprehension, averaged across participants. **b**, The mean context lengths (across layers) maximizing unique variance explained by the intersection for each participant. **c**, The cortical surface showing the best variance partition at two context lengths (1 and 32 s) for one participant in one condition (P7, layer 18) (see Supplementary Fig. 16 for individual layers and participants). The voxels with good prediction accuracy ($r > 0.05$) in the separate linguistic model are shown. PFC, prefrontal cortex; MPC, medial parietal cortex; AC, auditory cortex; VC, visual cortex. **d**, The mean weight correlation (across voxels and layers) for bimodal voxels compared with Production-only and

Comprehension-only voxels, averaged across participants. The bimodal voxels showed significantly lower weight correlation than Production-only (production: $t(8.4) = 11.99$, two-sided parametric test, $P = 1.4 \times 10^{-6}$, $\beta = 0.038$, 95% CI 0.032 to 0.045) and Comprehension-only voxels (comprehension: $t(1227) = -21.00$, two-sided parametric test, $P < 2.2 \times 10^{-16}$, $\beta = 0.021$, 95% CI 0.019 to 0.023). **e**, The weight correlations for bimodal linguistic voxels at two context lengths (1 and 32 s) shown for one participant in one condition (P7, layer 18) (see Supplementary Fig. 16 for all participants). The voxels with good prediction accuracy ($r > 0.05$) in the separate linguistic model are shown. The shaded areas in **a**, **b** and **d** represent the standard deviation across layers.

We then examined whether bimodal voxels exhibited distinct linguistic tuning for production and comprehension. To achieve this, we calculated Pearson's correlations of the separate linguistic model weights for production and comprehension specifically for bimodal voxels (Fig. 4c). The correlations were slightly negative and close to zero (Fig. 4d and see Supplementary Fig. 13 for individual layers and participants), suggesting that bimodal voxels are independently or

dissimilarly tuned for the two modalities. In comparison with unimodal voxels—those with the largest unique variance for either production or comprehension—the bimodal voxels exhibited more negative correlations. The LME analysis confirmed this difference (production: $t(8.4) = 11.99$, $P < 0.001$, $\beta = 0.038$, 95% CI 0.032 to 0.045; comprehension: $t(1227) = -21.00$, $P < 0.001$, $\beta = 0.021$, 95% CI 0.019 to 0.023). These findings indicate that the bimodal voxels are independently tuned for

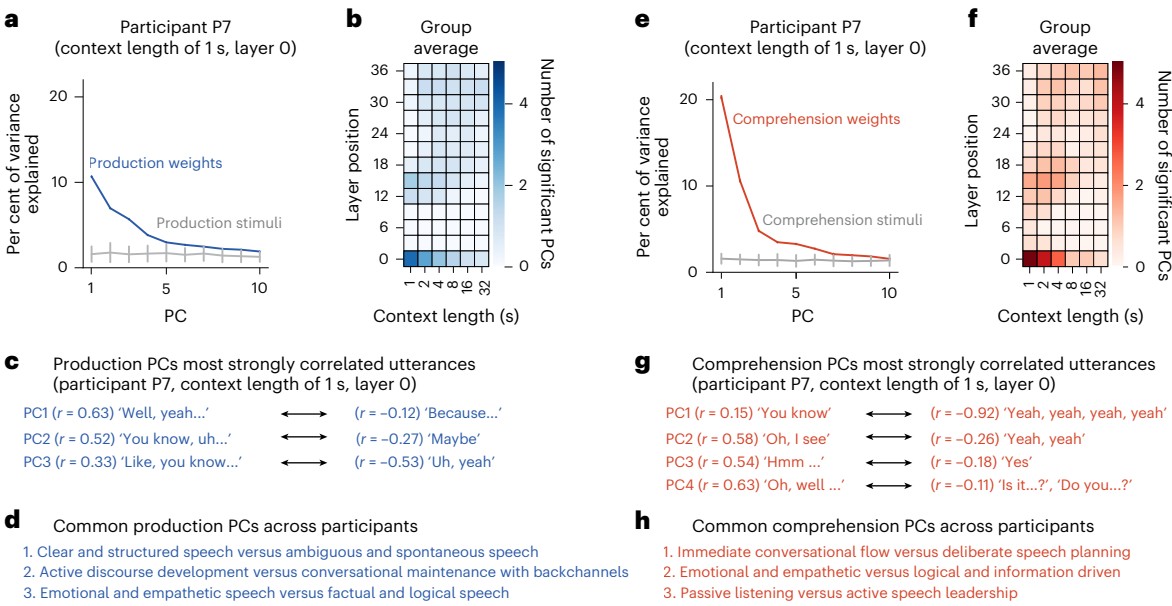

**Fig. 5 | PCs highlight conversation-specific linguistic representations.**
Production PC results are shown in a–d; comprehension PC results are shown in e–h. **a,e**, The variance explained in the separate linguistic model weights by each of the top ten PCs identified for a participant (P7) at one context-layer condition. The grey lines represent the variance explained by the PCs of the conversation content. The stimulus PCs were aligned with weight PCs using the Gale–Shapley stable-match algorithm. The data are presented as the original PCA results, with the error bars representing the standard deviation across 1,000 bootstrap samples. The standard deviations for the model weight PCs are very small. **b,f**, The mean number of significant PCs across participants (see Supplementary Fig. 17 for individual participant data). **c,g**, The utterances most strongly correlated with significant PCs for a participant (P7). The numbers in parentheses indicate the correlations between PC coefficients and GPT embeddings. **d,h**, The common PCs across participants identified for a 1-s context and layer 0 using ChatGPT.

each modality, reflecting the distinct linguistic demands of production and comprehension.

To ensure that these findings were not influenced by the instruction-tuned GPT model[41], we replicated the core analyses using a base GPT model before instruction tuning. The results were consistent, confirming that instruction tuning did not affect the observed effects (Supplementary Fig. 18).

### Revealing semantic tuning to interactive languages
To elucidate the linguistic organization underlying conversational content, we conducted principal component analysis (PCA) on the 2,816-dimensional separate linguistic model weights for each modality and participant. Building on previous research that mapped cortical semantic representations during natural speech comprehension[21,42], we adapted this framework for natural conversations. Due to variability in conversational content, a PCA was conducted separately for each participant and independently for production and comprehension, yielding modality-specific principal components (PCs). To assess the statistical robustness of the identified PCs, we performed a comparable PCA on GPT embeddings of the corresponding speech stimuli ('stimulus PCA'). We quantified the variance explained by each PC through 1,000 bootstrap resampling iterations to establish statistical significance.

Our analysis revealed the highest number of significant PCs in the embedding layer (layer 0) at a context length of 1 s for both production and comprehension (Fig. 5b,f). For production, four PCs were identified in four participants, three PCs in three participants and five PCs in one participant ($P < 0.001$, bootstrap test) (Supplementary Fig. 19). For comprehension, five PCs were identified in four participants, while four PCs were identified in the remaining four participants ($P < 0.001$, bootstrap test).

To interpret these significant PCs, we analysed the conversation content most strongly correlated with each PC. The majority of highly correlated content comprised words and phrases characteristic of interactive contexts, such as backchannel responses and conversational fillers (for example, participant P7) (Fig. 5c,g). To identify PCs consistently observed across participants, we utilized ChatGPT to analyse the top 20 most strongly correlated phrases for each PC, summarizing the common patterns. For production, four components emerged: (1) clear and structured speech versus ambiguous and spontaneous speech, (2) active discourse development versus conversational maintenance with backchannels, (3) emotional and empathetic speech versus factual and logical speech and (4) cautious speech with high cognitive load versus fluent and cooperative speech. For comprehension, four components were identified: (1) immediate conversational flow versus deliberate speech planning, (2) emotional and empathetic versus logical and information-driven, (3) passive listening versus active speech leadership and (4) concrete, experience-based versus abstract, conceptual speech. These findings demonstrate that both production and comprehension are tuned to the semantic demands of interactive language, revealing shared lexical–semantic components across participants. This highlights a consistent semantic organization that supports real-time social communication.

## Discussion
This study explored the neural representations of conversational content across production and comprehension modalities and multiple timescales. We identified shared linguistic representations exhibiting timescale-dependent topographic organization (Fig. 2). For shorter contexts (1–4 s), corresponding to words and single sentences, shared representations were localized in higher-order association cortices, including the prefrontal, temporal and parietal regions. By contrast, for longer contexts (16–32 s), spanning multiple conversational turns, these shared representations were more distributed and idiosyncratic among participants. Furthermore, modality-specific

timescale selectivity revealed enhanced encoding for shorter contexts during production and for longer contexts during comprehension (Fig. 3), suggesting distinct temporal integration processes. We also identified dual linguistic representations in bimodal voxels, encoding modality-specific information for both production and comprehension (Fig. 4). Despite these timescale-specific patterns, our analysis of low-dimensional linguistic representations revealed lexical–semantic components predominantly associated with shorter timescales (Fig. 5).

Theoretical models for the neural mechanism of conversation have proposed a common neural basis for language production and comprehension[2,43]. Empirical studies have adopted two primary approaches to examine this commonality: (1) the between-subjects approach, which examines the transmission of messages from speaker to listener[3,4,44] and (2) the within-subject approach, which investigates shared neural mechanisms within individuals[25–29]. Our study contributes to the within-subject approach by revealing both shared and distinct neural representations and their modulation by contextual timescales within individual participants.

Previous neuroimaging studies, using the within-subject approach, have manipulated the semantic and syntactic dimensions of stimuli to reveal shared representations[25–28]. Recent research utilizing spontaneously generated sentences and conversations has enhanced ecological validity[45,46], uncovering shared semantic and syntactic representations during natural language use[29,31]. For instance, recent research[29] used electrocorticography during natural conversations and modelled transient neural activity before and after word onset, identifying overlapping regions for word production and comprehension. However, two critical questions remain unresolved: (1) whether linguistic representations generalize across modalities and (2) how these shared representations vary across timescales. Our study addresses these gaps, demonstrating the generalizability of shared representations and their modulation by the amount of contextual information.

The topographic organization of shared neural representations varied across multiple timescales. At shorter timescales (1–4 s), corresponding to the duration of words and single sentences, shared representations were localized in higher-order brain regions, including the bilateral prefrontal, temporal and parietal cortices. This finding aligns with previous studies that mapped neural representations of intermediate linguistic structures, such as words and single sentences, onto these regions during naturalistic narrative comprehension[14,15,19,24]. These regions have consistently been associated with sentence-level processing in traditional neuroimaging studies of isolated sentences presented at shorter timescales (less than 6 s)[25,27,28]. Furthermore, these brain regions are partially overlap with those involved in linguistic knowledge and processes that are shared across both production and comprehension[26,47]. Therefore, the shared representations observed at shorter timescales suggest the presence of a common neural code for sentence-level linguistic information ('sentence meaning').

By contrast, at longer timescales (16–32 s), spanning multiple conversational turns, shared representations were distributed across broader cortical regions, exhibiting notable interindividual variability. Some participants (P1, P2, P4, P6 and P7) demonstrated shared representations extending into brain regions associated with the default mode network and the theory of mind (ToM) network. The default mode network has been implicated in representing higher-order discourse and narrative frameworks by integrating extrinsic information (for example, utterances) with intrinsic information (that is, prior context and memory)[7,14,48]. Similarly, the ToM network supports reasoning about others' mental states, a critical function in both language production and comprehension during conversations[47,49,50]. This network is particularly engaged in inferring the mental states of conversational partners, thereby facilitating pragmatic inferences about that particular individual[49,51,52]. These findings suggest that shared representations at longer timescales support the integration of incoming conversational content with prior conversational context, as well as with broader social knowledge and beliefs. Such integration may support the formation of a psychological model of the situation, enabling inferences about the interlocutor's intended meaning ('speaker meaning'). Individual differences in the spatial distribution of these shared representations may reflect variability in discourse-level integration strategies.

The contrasting timescale selectivity between production and comprehension may reflect their distinct functional demands in processing linguistic input and output. Our findings demonstrate that language comprehension exhibits enhanced encoding for longer timescales, consistent with the requirements of real-world language comprehension. Effective comprehension necessitates the integration of linguistic input with world knowledge, beliefs and memory to extract meaning from extended contexts[5,7,48]. This is consistent with evidence indicating that the brain prioritizes understanding broader discourse-level and overarching meanings over shorter units, such as individual words or sentences[14,19]. By contrast, we found that language production is characterized by enhanced encoding for shorter timescales. Production involves extensive preparatory processes, including ideation, lexical selection, syntactic structuring and speech planning, all of which occur before speech output[9,32,51,53–56]. Furthermore, production must dynamically adapt to the interlocutor's immediate reactions, ensuring fluid and responsive communication[57,58]. These demands suggest that production prioritize responsiveness and flexibility over reliance on extended contextual information.

Dual representations in the bimodal voxels exhibited selectivity for longer timescales (exceeding 8 s), corresponding to the integration of multiple sentences into coherent discourse. These representations probably facilitate the ability to maintain and distinguish perspectives, a critical function during conversation[30]. Conversations inherently require participants to navigate distinct perspectives, requiring differentiation at the neural level. Such interpersonal cognitive processes, integral to managing multiple perspectives, are probably not limited to external communication but may also underpin internal speech processes[59].

Despite this modality-specific timescale selectivity, our PCA identified similar lexical semantic components across modalities within the embedding layer at short timescales (1–4 s). Our results potentially extend the seminal work of Huth and colleagues[21], which comprehensively mapped fine-grained semantic representations during natural speech listening using word embeddings. Specifically, that study identified the first PC differentiating between 'humans and social interaction' and 'perceptual descriptions, quantitative descriptions and setting', thereby separating social content from physical content. Our conversational data offered a unique opportunity to examine the semantic space surrounding social words in greater depth. Notably, our identified PCs reflected social interaction nuances, such as backchannels, confirmations and fillers. These elements require minimal cognitive effort yet are vital for maintaining conversational flow[1,10–12]. By contrast, PCs linked to 'factual and logical speech' or 'logical and information-driven', such as referring to locations or objects, were identified as opposite axis of the social components ('emotion and empathetic'). This suggests that interactive language enhances the neural representation of social content, highlighting the interplay between semantic representations and social cognition.

Several limitations of the present study should be noted. We did not conduct functional localizer tasks to delineate specific functional networks, such as the language network and ToM network. Thus, our analysis could not precisely attribute voxel clusters to specific functional networks.

Our findings shed light on temporally hierarchical neural linguistic representations underlying both sentence meaning and speaker meaning during real-world conversations. Modality-aligned representations were primarily localized to brain regions involved in processing word- and sentence-level linguistic information over shorter timescales, while modality-specific representations exhibited distinct timescale selectivity: shorter contexts for production and longer contexts for

comprehension. These findings emphasize the importance of investigating the neurobiological basis of language within socially interactive contexts to comprehensively understand human language use.

## Methods

### Participants

Eight healthy, right-handed native Japanese speakers (P1–P8) participated in the fMRI experiment. The participants comprised five males (P1: age 22, P2: age 22, P3: age 23, P5: age 20 and P8: age 20) and three females (P4: age 22, P6: age 20 and P7: age 20). All participants were confirmed as right-handed through the Edinburgh Handedness Inventory[60] (with a laterality quotient score of 75–100), and they had normal hearing as well as normal or corrected-to-normal vision. The experimental protocol was approved by the Ethics and Safety Committee of the National Institute of Information and Communications Technology, Osaka, Japan. Written informed consent was obtained from all participants before the experiment.

### Natural dialogue experiment

The experiment consisted of 27 conversation topics, including self-introduction and favourite classes (Supplementary Table 5). These topics were selected to cover a wide range of semantic domains relevant to daily life, such as knowledge, memory, imagination and temporal and spatial cognition, referencing the Corpus of Everyday Japanese. Each fMRI run lasted 7 min and 10 s and focused on a specific topic. The participants engaged in unscripted, natural dialogues, freely expressing their thoughts and emotions while responding in real time to their interlocutor's input. Speech was delivered and recorded via fMRI-compatible insert earphones and a noise-cancelling microphone, respectively. Both the participants' and interlocutor's speech were recorded separately for subsequent analysis. Each participant completed 27 runs across four sessions, except for P3, who completed three sessions. Due to the collection of a single valid run in one session, only three sessions were analysed for P2 and P5, and the analysis included two to ten runs per session (Supplementary Table 4). On average, the participants produced speech during $217.1 \pm 26.0$ (mean ± standard deviation) fMRI volumes per run (range 170.6–262.1), while comprehending speech during $214.4 \pm 11.8$ volumes (range 199.2–234.0) (Supplementary Fig. 1).

### MRI data acquisition

Magnetic resonance imaging (MRI) data were collected on a 3T MRI scanner at CiNet. Participants P1–P5 were scanned on a Siemens MAGNETOM Prisma, while P6–P8 were scanned on a Siemens MAGNETOM Prisma Fit, both equipped with 64-channel head coils. Functional images were acquired using a T2-weighted gradient echo multiband echo-planar imaging sequence[61] in interleaved order, covering the entire brain. The imaging parameters were as follows: TR of 1.0 s, echo time (TE) of 30 ms, flip angle of 60°, matrix size of 96 × 96, field of view of 192 mm × 192 mm, voxel size of 2 mm × 2 mm × 2 mm, slice gap of 0 mm, 72 axial slices, multiband factor of 6. High-resolution anatomical images were obtained using a T1-weighted MPRAGE sequence with the following parameters: TR of 2.53 s, TE of 3.26 ms, flip angle of 9°, matrix size of 256 × 256, field of view of 256 mm × 256 mm, voxel size of 1 mm × 1 mm × 1 mm.

### fMRI data preprocessing

The fMRI data were preprocessed using the Statistical Parametric Mapping toolbox (SPM8). Motion correction was applied to each run, aligning all volumes to the first echo-planar imaging frame for each participant. To remove low-frequency drift, we used a median filter with a 120-s window. The response for each voxel was then normalized by subtracting the mean response and scaling to unit variance. The cortical surfaces were identified using FreeSurfer[62,63], which registered the anatomical data with the functional data. For each participant, only voxels identified within the cerebral cortex were included in the analysis, ranging from 64,072 to 72,018 voxels per participant. The flatmaps were generated by projecting

voxel values onto cortical surfaces using Pycortex[64]. Cortical anatomical parcellation was performed using the Destrieux Atlas[65], and the resulting parcellations were visualized on cortical surface maps.

### Transcription and temporal alignment

Conversational speech was transcribed morphologically using the Microsoft Azure Speech-to-Text, followed by manual correction for accuracy. The morphemes were grouped into meaningful semantic chunks, approximating the fMRI TR (1,000 ms) and temporally aligned to the corresponding fMRI volumes using the midpoint of each chunk's duration.

### Contextual embedding extraction

To extract contextual embeddings from the content of conversations, we utilized an instruction-tuned language model (GPT) fine-tuned specifically for Japanese[37] (https://huggingface.co/rinna/japanese-gpt-neox-3.6b-instruction-sft). This model is built on the open-source GPT-NeoX architecture[66] and was pretrained to predict the next word on the basis of preceding context using 312.5 billion tokens from various Japanese text datasets: Japanese CC-100, Japanese C4 and Japanese Wikipedia. For comparative purposes, we also replicated our analysis using the non-instruction-tuned version of the model (https://huggingface.co/rinna/japanese-gpt-neox-3.6b) as detailed in Supplementary Fig. 18. Instruction tuning was performed using datasets translated into Japanese, including Anthropic HH RLHF data, FLAN Instruction Tuning data and the Stanford Human Preferences Dataset. The resulting model architecture comprises 36 transformer layers with hidden unit dimensions of 2,816.

We processed transcribed utterances using GPT-NeoX with context lengths of 1, 2, 4, 8, 16 and 32 s, extracting embeddings by averaging the internal representations of all tokens within each utterance. To investigate differences in prediction accuracy across model layers, we extracted embeddings from the input layer (embedding layer), as well as every third layer within the model. As a control to account for predictions potentially driven by low-level sensory or motor brain activity, we generated random normal embeddings[4] with the same dimensionality as the GPT embeddings (2,816 features). These embeddings were matched to individual utterance instances corresponding to each TR (for example, 'something I'm thinking' in Fig. 1a).

### Head motion model construction

To account for BOLD signal variance attributable to head motion, six translational and rotational motion parameters estimated during preprocessing were included as regressors. Frame-wise displacement values, calculated following previous research[67], were also incorporated. A distance of 50 mm between the cerebral cortex and the head centre was assumed in accordance with a prior study[67].

### Separate and unified linguistic model construction

We constructed two linguistic models to evaluate hypotheses regarding the neural representation of language production and comprehension. For the separate linguistic model, it assumes independent representations for production and comprehension, combining contextual embeddings extracted separately for each modality. Each embedding set comprised 2,816 features, derived from combinations of 13 layers (0, 3, …, 36) and 6 context lengths (1, 2, 4, 8, 16, 32), yielding 78 feature combinations per modality and a total of 5,632 features. Identical feature pairs were used across both modalities. For the unified linguistic model, it assumes shared neural representations for production and comprehension. It utilized 2,816 contextual embeddings derived from combined speech content of both modalities within each TR. If only one modality was present, embeddings were derived solely from that modality.

### Voxel-wise model estimation and testing

To model cortical activity in individual voxels, we used a FIR model, which accounts for the slow hemodynamic responses and their

coupling to neural activity. Although the canonical hemodynamic response function is widely used in fMRI studies, it assumes a uniform HRF shape across cortical voxels. This simplification can result in inaccuracies, given that the shape of the hemodynamic response varies across cortical regions[68]. To address this variability, we concatenated 5,632 linguistic features with time delays spanning two to seven samples (2–7 s), yielding a total of 34,932 features. We modelled the BOLD responses as a linear combination of these features, with weights estimated using banded ridge regression, implemented via Himalaya package[38,39]. Regularization parameters were optimized through fivefold cross-validation, exploring ten values between $10^{-2}$ and $10^{7}$. Model testing utilized leave-one-session-out cross-validation, in which one session was withheld for testing while the remaining sessions served as training data. The prediction accuracy was evaluated by calculating the Pearson's correlation coefficient between observed and predicted BOLD responses in the test dataset. The statistical significance was determined through a one-sided permutation test. A null distribution was generated by permuting 20-TR blocks (20 s) of the left-out test data 1,000 times, recalculating the correlation for each permutation. Multiple comparisons were corrected using the FDR procedure[69].

### LME model analysis

To explore how timescales and layer positions influence prediction accuracy and weight correlations, we conducted LME model analyses using the lmer function from the lmerTest package (version 3.1-3)[70] in R (version 4.3.3). Fixed effects included layer position, context length, their interaction and quadratic terms for both predictors to capture potential non-linear relationships. The models included by-participant random intercepts and random slopes for context length and its quadratic term, allowing for individual variability in the effects of contextual information. To assess the influence of encoding model type or voxel type, we extended the LME model structure by adding type as both a fixed effect and a by-participant random slope. Finally, we simplified the models by stepwise removal of non-significant predictors, selecting the model structure with the lowest Akaike Information Criterion values using the Kenward–Roger approximation. The *P* values smaller than $2.2 \times 10^{-16}$ are reported as $< 2.2 \times 10^{-16}$, which is the lower limit of the default precision in R.

### Variance partitioning

We performed variance partitioning to quantify the unique contributions of linguistic features to BOLD responses in production, comprehension and their intersection. Following methods from previous voxel-wise modelling studies[16,40], we used three models: a production-only model, a comprehension-only model and their combination (that is, the separate linguistic model). We used set theory to calculate the unique and common variances explained as follows. Unique variance was calculated as follows

$$\text{Production} \backslash \text{comprehension} = \text{production} \cup \text{comprehension} \\ - \text{comprehension}$$

$$\text{Comprehension} \backslash \text{production} = \text{production} \cup \text{comprehension} \\ - \text{production}.$$

Shared variance was calculated as follows

$$\text{Production} \cap \text{comprehension} = \text{production} + \text{comprehension} \\ - \text{production} \cup \text{comprehension}.$$

While variance partitioning is typically reported using $R^2$ values, we report the square roots of these values to align with our primary evaluation metric—correlation coefficients—thereby facilitating direct

comparison and consistent interpretation across all reported results. Variance partitioning was applied to all layer-context combinations. In principle, variance partitioning assumes equal sample sizes across conditions. However, in our naturalistic dialogue experiment, individual fMRI frames (TRs) may correspond to production, comprehension, both or neither, resulting in inherent unequal sample sizes across conditions. To preserve the ecological validity of the dataset and avoid imposing artificial constraints, we applied variance partitioning uniformly across all TRs.

### PCA

To identify low-dimensional representations of the separate linguistic model weights, we performed a PCA following previous studies[21,42] separately for production and comprehension. Model weights, averaged across six delays for each feature (33,792/6 weights = 5,632 mean weights) and sessions, were scaled by prediction accuracy to reduce contributions from voxels with lower prediction accuracy. A PCA was performed separately for each modality on these scaled weights in all cortical voxels (2,816 weights × all cortical voxels), yielding 2,816 orthogonal PCs. We assessed the significance of the first 20 weight PCs by comparing their explained variance with the first 20 stimulus PCs (derived from GPT embeddings) using bootstrapping (1,000 iterations). Correspondence between weight and stimulus PCs was enhanced using the Gale–Shapley stable marriage algorithm. The PCs were deemed significant if the stimulus PC never explained more variance than the corresponding weight PC in all bootstrap samples ($P < 0.001$).

For our current analysis, we focused on a context length of 1 s for layer 0, which yielded the highest number of significant PCs across participants (Supplementary Fig. 19). Interpretation involved three steps: (1) Identification of correlated utterances: for each PC, the top 20 positively and negatively correlated utterances were identified for each participant (Fig. 5c,g). (2) Interpretation using ChatGPT: utterances and correlation coefficients were input into ChatGPT (GPT-4o) for consistent interpretations across PCs, modalities and participants. (3) Synthesis of common components: ChatGPT synthesized interpretations to identify common components across participants.

### Reporting summary

Further information on research design is available in the Nature Portfolio Reporting Summary linked to this article.

## Data availability

The MRI data and preprocessed stimulus features used in the current study are available via OpenNeuro at https://openneuro.org/datasets/ds004669. The Destrieux Atlas can be accessed via the FreeSurfer software package (https://surfer.nmr.mgh.harvard.edu/fswiki/CorticalParcellation). The Corpus of Everyday Japanese is available from the National Institute for Japanese Language and Linguistics (https://www2.ninjal.ac.jp/conversation/cejc-monitor.html). Because the free-form conversations include complex details that could reveal participants' identities, the raw speech data—after removal of personal identifiers—will be provided only to researchers who (1) contact the corresponding author (S.N.) and (2) sign a data-sharing agreement that complies with the regulations of the relevant ethics committees and with applicable privacy laws.

## Code availability

The custom code for this study is available via GitHub at https://github.com/yamashita-lang/dialogue. All model fitting and analyses were performed using custom software written in Python, utilizing libraries such as NumPy[71], SciPy[72], Scikit-learn[73], Matplotlib[74], Himalaya[38] and Pycortex[64]. An exception was the LME analysis, which was performed using the lmertTest[70] package in R. The GPT models were accessed via Huggingface[75].

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

## Acknowledgements

This work was supported by JST ERATO (grant no. JPMJER1801 to S.N.), AIP Acceleration Research (grant no. JPMJCR24U2 to S.N.), MIRAI (grant no. JPMJMI19D1 to S.N.) and JSPS (grant nos. JP23H05493 and JP24H00619 to S.N.). The funders had no role in study design, data collection and analysis, decision to publish or preparation of the manuscript.

## Author contributions

S.N. conceptualized the experiment. R.K. and S.N. designed the experiment. R.K. performed the experiment. M.Y., R.K. and S.N. analysed and interpreted the results. M.Y. and S.N. wrote the paper. S.N. supervised the study.

## Competing interests

The authors declare no competing interests.

## Additional information

**Correspondence and requests for materials** should be addressed to Shinji Nishimoto.

# Reporting Summary

## Statistics

For all statistical analyses, confirm that the following items are present in the figure legend, table legend, main text, or Methods section.

| n/a | Confirmed | |
|-----|-----------|---|
| ☐ | ☒ | The exact sample size (*n*) for each experimental group/condition, given as a discrete number and unit of measurement |
| ☐ | ☒ | A statement on whether measurements were taken from distinct samples or whether the same sample was measured repeatedly |
| ☐ | ☒ | The statistical test(s) used AND whether they are one- or two-sided <br> *Only common tests should be described solely by name; describe more complex techniques in the Methods section.* |
| ☐ | ☒ | A description of all covariates tested |
| ☐ | ☒ | A description of any assumptions or corrections, such as tests of normality and adjustment for multiple comparisons |
| ☐ | ☒ | A full description of the statistical parameters including central tendency (e.g. means) or other basic estimates (e.g. regression coefficient) AND variation (e.g. standard deviation) or associated estimates of uncertainty (e.g. confidence intervals) |
| ☐ | ☒ | For null hypothesis testing, the test statistic (e.g. *F*, *t*, *r*) with confidence intervals, effect sizes, degrees of freedom and *P* value noted <br> *Give P values as exact values whenever suitable.* |
| ☒ | ☐ | For Bayesian analysis, information on the choice of priors and Markov chain Monte Carlo settings |
| ☒ | ☐ | For hierarchical and complex designs, identification of the appropriate level for tests and full reporting of outcomes |
| ☐ | ☒ | Estimates of effect sizes (e.g. Cohen's *d*, Pearson's *r*), indicating how they were calculated |

*Our web collection on statistics for biologists contains articles on many of the points above.*

## Software and code

Policy information about availability of computer code

| | |
|---|---|
| Data collection | Siemens MAGNETOM Prisma (syngo MR E11) and Siemens MAGNETOM Prisma Fit (syngo MR E11). |
| Data analysis | The code supporting the findings of this study is available on GitHub: https://github.com/yamashita-lang/dialogue. <br> Initial transcription was performed using Microsoft Azure Speech-to-Text (Speech SDK version 1.16.0; https://learn.microsoft.com/en-us/azure/ai-services/speech-service/). <br> fMRI data preprocessing was conducted using MATLAB (R2019b, MathWorks Inc.) in combination with SPM8 (https://www.fil.ion.ucl.ac.uk/spm/). <br> Banded ridge regression was implemented in Python 3 using the Himalaya package (version 0.4.2; https://gallantlab.org/himalaya/) with PyTorch (version 2.1.2+cu118; https://pytorch.org/). <br> Language data were processed using various tools, including Japanese GPT models (https://huggingface.co/rinna/japanese-gpt-neox-3.6b-instruction-sft; https://huggingface.co/rinna/japanese-gpt-neox-3.6b) and code adapted from https://github.com/mtoneva/brain_language_NLP. <br> Data analysis was carried out using NumPy (version 1.26.4; https://numpy.org/), SciPy (version 1.12.0; https://scipy.org/), and Scikit-learn (version 1.4.0; https://scikit-learn.org/). <br> Cortical surface visualizations were generated using FreeSurfer (version 6.0; https://surfer.nmr.mgh.harvard.edu/) and Pycortex (version 1.2; https://github.com/gallantlab/pycortex). <br> Plots were created with the Matplotlib (version 3.8.2; https://matplotlib.org/) and seaborn (version 0.13.2; https://seaborn.pydata.org/). <br> Linear mixed-effects modeling was conducted in R (version 4.3.3; https://cran.r-project.org) using the lmerTest package (version 3.1-3; https://cran.r-project.org/web/packages/lmerTest/index.html). |

For manuscripts utilizing custom algorithms or software that are central to the research but not yet described in published literature, software must be made available to editors and reviewers. We strongly encourage code deposition in a community repository (e.g. GitHub). See the Nature Portfolio guidelines for submitting code & software for further information.

## Data

Policy information about availability of data

All manuscripts must include a data availability statement. This statement should provide the following information, where applicable:
  - Accession codes, unique identifiers, or web links for publicly available datasets
  - A description of any restrictions on data availability
  - For clinical datasets or third party data, please ensure that the statement adheres to our policy

MRI data and preprocessed stimulus features used in the current study are available via OpenNeuro at https://openneuro.org/datasets/ds004669.
The Destrieux Atlas can be accessed via the FreeSurfer software package (https://surfer.nmr.mgh.harvard.edu/fswiki/CorticalParcellation).
The Corpus of Everyday Japanese is available from the National Institute for Japanese Language and Linguistics (https://www2.ninjal.ac.jp/conversation/cejc-monitor.html).
Because the free-form conversations include complex details that could reveal participants' identities, the raw speech data—after removal of personal identifiers—will be provided only to researchers who (i) contact the corresponding author (S.N.) and (ii) sign a data-sharing agreement that complies with the regulations of the relevant ethics committees and with applicable privacy laws.

## Research involving human participants, their data, or biological material

Policy information about studies with human participants or human data. See also policy information about sex, gender (identity/presentation), and sexual orientation and race, ethnicity and racism.

| | |
|---|---|
| Reporting on sex and gender | The participants were selected to include male and female. Gender was determined based on self-reporting. Since these variables were outside of this study, they were not considered in our analysis. |
| Reporting on race, ethnicity, or other socially relevant groupings | Racial or ethnic information was not collected. Participants self-reported to be a native speaker of the language (Japanese) used in the experiment. |
| Population characteristics | Participants (3 female, 5 male) between 20 and 23 years of age, with normal hearing, normal or corrected-to-normal vision, without history of diagnosis of language or hearing diseases. |
| Recruitment | Participants were recruited through social networking sites (Twitter). They were undergraduate students from the nearby area. There was no self-selection bias. |
| Ethics oversight | The experiment was approved by the Ethical Committee of National Institute of Information and Communications. Technology. |

Note that full information on the approval of the study protocol must also be provided in the manuscript.

# Field-specific reporting

Please select the one below that is the best fit for your research. If you are not sure, read the appropriate sections before making your selection.

☒ Life sciences  ☐ Behavioural & social sciences  ☐ Ecological, evolutionary & environmental sciences

For a reference copy of the document with all sections, see nature.com/documents/nr-reporting-summary-flat.pdf

# Life sciences study design

All studies must disclose on these points even when the disclosure is negative.

| | |
|---|---|
| Sample size | Data analysis and statistics were examined and confirmed for each participant separately. The sample size of the test data (at least 860 samples) was determined to perform proper predictive and statistical analysis for encoding models and to match our prior attempts (e.g., Nakai and Nishimoto, 2020 Nature Communications). |
| Data exclusions | Some data were excluded from the analysis if only one run of fMRI data was available within one session (fourth session of participant P2 and fourth session of participant P5). |
| Replication | Encoding models were fit and evaluated for each participant, and the results were consistent across the 8 participants. The sample size is comparable to previous studies using voxel-wise encoding model (e.g., Nakai and Nishimoto, 2020 Nature Communications). |
| Randomization | Randomization was not relevant to this study as participants were not allocated into experimental groups. |
| Blinding | Blinding was not relevant to this study as participants were not allocated into experimental groups. |

# Reporting for specific materials, systems and methods

We require information from authors about some types of materials, experimental systems and methods used in many studies. Here, indicate whether each material, system or method listed is relevant to your study. If you are not sure if a list item applies to your research, read the appropriate section before selecting a response.

## Materials & experimental systems

| n/a | Involved in the study |
|-----|----------------------|
| ☒ | ☐ Antibodies |
| ☒ | ☐ Eukaryotic cell lines |
| ☒ | ☐ Palaeontology and archaeology |
| ☒ | ☐ Animals and other organisms |
| ☒ | ☐ Clinical data |
| ☒ | ☐ Dual use research of concern |
| ☒ | ☐ Plants |

## Methods

| n/a | Involved in the study |
|-----|----------------------|
| ☒ | ☐ ChIP-seq |
| ☒ | ☐ Flow cytometry |
| ☐ | ☒ MRI-based neuroimaging |

## Magnetic resonance imaging

### Experimental design

| | |
|---|---|
| Design type | Building voxel-wise encoding models using task-evoked brain activity (Nishimoto et al., 2011 Current Biology). |
| Design specifications | The experiment was conducted in three or four separate fMRI sessions. Each session collected 2 to 10 runs. The total of 27 runs were acquired across the sessions. A single run consisted of 430 seconds. |
| Behavioral performance measures | Behavioral performance was not measured quantitatively. The experimenter who engaged in the conversation judged based on whether the participant continued the conversation. |

### Acquisition

| | |
|---|---|
| Imaging type(s) | functional, structural. |
| Field strength | 3T. |
| Sequence & imaging parameters | Functional data: A multiband gradient echo-planar imaging sequence (TR = 1,000 ms, TE = 30 ms, flip angle = 60°; voxel size = 2 × 2 × 2 mm3, matrix size = 96 × 96, 72 axial slices, FOV = 192 × 192 mm2, multiband factor = 6).<br><br>Structural data: T1-weighted MPRAGE (TR = 2530 ms, TE = 3.26 ms, flip angle = 9°, voxel size = 1 × 1 × 1 mm3, matrix size = 256 × 256, 256 axial slices, FOV = 256 × 256 mm2). |
| Area of acquisition | A whole-brain scan was used. |

Diffusion MRI    ☐ Used    ☒ Not used

### Preprocessing

| | |
|---|---|
| Preprocessing software | SPM8 (motion correction) and FreeSurfer 5.3.0 (anatomical registration, cortical surface reconstruction, cortical segmentation, and subcortical segmentation). |
| Normalization | Data were not normalized. Data for each participant were analyzed individually. |
| Normalization template | The data were not normalized. |
| Noise and artifact removal | Motion correction (6DOF) was performed by aligning all of the EPI data to the first image from the first scan for each subject. For each voxel, responses were normalized by subtracting the mean response across all time points, and trend was removed using a median filter (120-s time window). These processes were performed using in-house MATLAB codes (Cukur et al., 2016 The Journal of Neuroscience). No spatial smoothing procedure was performed. |
| Volume censoring | No censoring was performed and all data were used for the study. |

### Statistical modeling & inference

| | |
|---|---|
| Model type and settings | Multivariate, predictive. Feature-based encoding models were built using the training data, and the modeling accuracy was examined by using the held-out test data. Leave-one-session-out cross-validation was performed to estimate average modeling accuracy across the sessions. |

Effect(s) tested Prediction performance was calculated by Pearson correlation coefficient between measured and predicted BOLD responses for each voxel.

Specify type of analysis: ☒ Whole brain ☐ ROI-based ☐ Both

Statistic type for inference voxel-wise.

(See <u>Eklund et al. 2016</u>)

Correction False-discovery rate (FDR) correction (Benjamini and Hochberg, 1995).

## Models & analysis

| n/a | Involved in the study |
|---|---|
| ☒ ☐ | Functional and/or effective connectivity |
| ☒ ☐ | Graph analysis |
| ☐ ☒ | Multivariate modeling or predictive analysis |

Multivariate modeling and predictive analysis We built voxel-wise encoding models (Naselaris et al., 2011 NeuroImage; Nishimoto et al., 2011 Current Biology) to explain the BOLD responses using speech-related features. Linguistic features of each stimulus utterance were extracted from a pre-trained GPT language model. Model weights were estimated using a L2-regularized linear regression procedure (Huth et al., 2012 Neuron) for training data (6,880 to 10,750 samples). The regularization parameter was optimized via 5-fold cross validation using the training data. The prediction accuracy of each voxel model was quantified by a Pearson's correlation coefficients between the measured and the predicted BOLD responses for test data (860 to 4,300 samples).

