## [Peer Review File · Nature Human Behaviour]

Conversational content is organized across multiple timescales in the brain

Corresponding Author: Professor Shinji Nishimoto

Version 0:

Decision Letter:

19th January 2024

Dear Prof Nishimoto,

Thank you once again for your manuscript, entitled "Distinct yet intertwined semantic representations in the cortex during natural dialogue", and for your patience during the peer review process.

Your Article has now been evaluated by 3 referees. You will see from their comments copied below that, although they find your work of considerable potential interest, they have raised quite substantial concerns. In light of these comments, we cannot accept the manuscript for publication, but would be interested in considering a revised version if you are willing and able to fully address reviewer and editorial concerns.

We hope you will find the referees' comments useful as you decide how to proceed. If you wish to submit a substantially revised manuscript, please bear in mind that we will be reluctant to approach the referees again in the absence of major revisions. We are committed to providing a fair and constructive peer-review process. Do not hesitate to contact us if there are specific requests from the reviewers that you believe are technically impossible or unlikely to yield a meaningful outcome.

To guide the scope of the revisions, the editors discuss the referee reports in detail within the team, including with the chief editor, with a view to (1) identifying key priorities that should be addressed in revision and (2) overruling referee requests that are deemed beyond the scope of the current study. We hope that you will find the prioritized set of referee points to be useful when revising your study. Please do not hesitate to get in touch if you would like to discuss these issues further.

In particular, we ask you to [1] provide a detailed analysis to reassure of the potential concerns with head motion artifacts (comments by Reviewer #2); [2] answer in full all questions about the methodological details that are currently unclear or missing, especially regarding the correlation between features, the variance partitioning, and the PC analyses (see comments by Reviewers #2 and #3); [3] perform the additional behavioral experiment to improve the labelling of the semantic features (as requested by Reviewer #1).

If you wish to submit a suitably revised manuscript, we would hope to receive it within 4 months. I would be grateful if you could contact us as soon as possible if you foresee difficulties with meeting this target resubmission date.

- Include a "Response to the editors and reviewers" document detailing, point-by-point, how you addressed each editor and referee comment. If no action was taken to address a point, you must provide a compelling argument. When formatting this document, please respond to each reviewer comment individually, including the full text of the reviewer comment verbatim followed by your response to the individual point. This response will be used by the editors to evaluate your revision and sent back to the reviewers along with the revised manuscript.
- Highlight all changes made to your manuscript or provide us with a version that tracks changes.

Link Redacted

Thank you for the opportunity to review your work. Please do not hesitate to contact me if you have any questions or would like to discuss the required revisions further.

Sincerely,

Nature Human Behaviour

Reviewer expertise:

Reviewer #1: NLP, semantic representations, encoding models

Reviewer #2: Language comprehension, encoding models, fMRI

Reviewer #3: Neurolinguistics, semantic representations

REVIEWER COMMENTS:

Reviewer #1:

Remarks to the Author:

In this work, the authors used voxelwise encoding modeling to explore brain representations during real-world conversations. Specifically, the study comprised human participants engaging in natural conversations with an experimenter across a variety of topics, while their BOLD activity was being recorded with fMRI. To model this data, the authors extracted features-of-interest for the word transcripts in each TR, doing so separately for produced and perceived instances. These features included part-of-speech, morpheme count etc. as well as the hidden state of GPTNeoX (upper-middle layer). Then, these features were used to build various linearized predictive models of the BOLD activity of each voxel individually and the models were evaluated on a held-out test set. The authors found good prediction performance broadly across the cerebral cortex. Next, they evaluated if the encoding model weights were transferable between production and perception, and what the correlation between these weights were. Overall, the weights were not transferable and were negatively correlated across cortex barring low-level auditory regions around the superior temporal gyrus. Then the authors evaluated the unique variance explained by production and perception, finding that a majority of voxels were well predicted by the linguistic content in both conditions while a handful of voxels were only predicted by the linguistic content of one or the other. Finally, the authors did PCA on the encoding weights for each condition and interpreted the information captured by each significant PC. They found that the PCs largely captured information related to conversations like turn-taking, backtracking etc. as opposed to the semantic distinctions reported in prior studies that consider pure perception.

Strengths:

This paper brings interesting methods to the nascent field of neural representations during naturalistic conversations. To the best of my knowledge, it is the first study to apply voxelwise encoding modeling analyses to such an experimental design. This preserves individual participant resolution lost in prior studies that instead mainly relied on inter-subject correlation. The study also highlights an important gap in the field by showing that many of the variations in voxels function during conversations do not in fact correspond to purely semantic variations observed in studies that only study perception. By jointly studying perception and production, we can make stronger claims about the functional role of each brain region.

I particularly found that lack of weight transfer between the two conditions and the conversation (but not semantic category) based interpretations of the encoding weight PCs very useful in understanding how voxels engaged in both production and perception might represent different, conversation-related information during each stage.

Weaknesses & Questions: Overall, the conclusions regarding purported functional roles of voxels during perception and production was confusing, as were the interpretations of the weights-based analyses.

- Why is Fig. 3 labeled as "semantic selectivity"? Also, wouldn't this analysis be sensitive to whether both production and perception happened every TR? Can the authors elaborate on the statistics of the transcripts, like the word rate for production and perception, if there are TRs comprising only one condition and how such TRs were treated?
- Can the authors elaborate on the lack of successful weight transfer between production and comprehension even in voxels that jointly represent the two? This is very closely related to the correlation experiment and should be grouped together. Also, aren't the strong "positive" correlations, suggestive of preserved semantic selectivity, only around auditory cortex (expectedly due to perception and auditory feedback during production)?
- In the Fig. 5 experiment:
 - Were the PCs estimated on weights of the joint model, or production and perception utterances separately? Which voxels went into the estimation for each condition? i.e., were the prod PCs estimated on "prod-only" and "both" voxels but not "comp only", or, on all significantly predicted voxels in this condition?
 - The correlation between interjections and ProdPC 1 seems to be very small. From Extended Fig. 10, it also seems that this PC is high in "comprehension only" regions like parts of the STG. How do the authors reconcile this with the claim that the first PCs capture turn-taking.

- It could be interesting to plot the distribution of each PC across voxels that are “production-only”, “comprehension-only” or both. Are there consistent patterns here? One might expect that “prod-only” voxels have very high ProcPC1 and low CompPC1, for example.
- The labels of the “semantic” features are based on the authors’ qualitative interpretations as far as I can tell. I would recommend running a more systematic behavioral experiment across multiple labelers as in Huth et al., 2016. Can the authors also elaborate why they refer to these PCs as “semantic”? How can we be assured that these differences are purely semantic in nature? This also affects the speculative discussion on why these PCs might arise in the first place. Without a strong assessment for interpreting each PC, it is difficult to understand their cortical distributions.
- While an interesting and novel analyses, the conclusion is not clear. What does it mean for the perception and production phases to have different PC interpretations? How does this relate to purported shared/different computational mechanisms in each voxel for the two conditions?
- What types of semantic concepts did the conversations cover? I am interested in understanding why the traditional semantic PCs uncovered during pure perception (for ex., “social” vs. “visual” vs. “tactile”) don’t come up here even in the later components- is this an artifact of not covering semantically-rich content, or, is it fundamental to real-world communication that earlier studies have missed? This also relates to a potentially critical claim the paper makes re “our results suggest that the foundational components of cortical semantic representations are formed based on universal systematic communication rules, regardless of the specific semantic content.” What does this result say about the purported function of voxels ascribed a specific semantic selectivity in prior work and the conversation-related selectivity found here?
- Throughout this study, the visual cortex is significantly predicted by the encoding models and is even attributed specific colors in the RGB PC space. Could this be because of eye motion during the task that need to be regressed out?
- Nits:
 - Which areas correspond to the “language network” being referred to in lines 369-378? I believe it is not Ev Fedorenko’s definition but perhaps “all regions activated during language production and comprehension”?
 - Perhaps reducing the curvature contrast will improve the visibility of RGB colors in the flatmaps.
- It might be useful to add one sentence about the FIR model while explaining the encoding model in the main text.

Reviewer #2:

Remarks to the Author:

The exploration of semantic representations during natural dialogue remains a relatively understudied yet highly pertinent area of research. Understanding how the human brain encodes linguistic information in real-world conversational settings remains unresolved. Previous linguistic studies have predominantly concentrated on language comprehension, often employing controlled linguistic stimuli like isolated words or sentences. Recent studies in language comprehension, utilizing narrative stories together with voxelwise encoding modeling, have shown a distributed network of brain regions responsible for representing semantic information, largely irrespective of the presentation modality. Nevertheless, the scarcity of studies that examined natural language production and/or interactive conversations has been limiting our understanding of linguistic representations in the brain.

The authors of this study are tackling this research gap within language research asking participants to engage in natural conversations on predefined topics while recording their brain activity using functional MRI. Subsequently, they extracted language comprehension and production features derived from spoken stimuli (either spoken by the participant or the experimenter) using large language models, such as GPT and applied a voxelwise encoding modeling framework to map semantic representations onto brain activity. The authors systematically explore both the distinct and shared semantic representations during language comprehension and production and report both distinct and overlapping representations. Importantly, their analysis extends to a detailed examination of voxel-level representations of semantic information, interpreting brain regions responsive to turn-taking interactions, backchannels, and self-mentalizing.

One reason why the current literature has been rather scarce to study brain representations of natural conversations is that it is really hard to conduct functional MRI experiments where subjects can freely speak during the experiment without fundamentally distorting the acquired signal through head motion. It is therefore crucial to present that potential motion artefacts have been investigated thoroughly. In its current form the manuscript is lacking the necessary detail to understand the affects of motion and low-level information on the results. Hence, I have several questions and comments pertaining to the data analysis, which I have listed below.

(1) Given that head motion can have a significant effect on acquired BOLD signal, it is important see more detailed analysis of potential motion artifacts and how they may be affecting the results. It would be a good start to examine the motion traces per individual in the cortical voxels. The motion parameters have been added to the analysis as nuisance regressors. However, I am very surprised to see in Extended Data Fig. 3 that motion (and other features but see my next comment for the other features) have no unique variance explained that seems to be very unlikely, especially for the production data. Did the authors plot head motion model separately in a ridge model? Did the authors have systematically and separately investigate the effect of speaking in the scanner?

(2) I have a hard time following what the authors did in their variance partitioning. It seems that the authors only computed the difference between the full model and a reduced model. Has this been done for all the different feature spaces separately? The methods section lacks the appropriate detail here and needs improvement. Please see the cited papers Le Bel et al. 2021 and de Heer et al. 2017 for a better description. Based on this I am also curious whether the authors fit each model as a separate ridge model? Given de Heer et al. 2017 it is very surprising that the low-level features are very poorly predicting in speech-related regions (Extended Data Fig. 3). Given that there may be voxels that share representations of GPT features and e.g. part-of-speech features it would be more beneficial to use a banded ridge model [Ja Tour et al. 2022; Nunez-Elizalde et al. 2019] and examine the split-correlations of the prediction accuracies across different models.

References:

la Tour, T.D., Eickenberg, M., Nunez-Elizalde, A.O. and Gallant, J.L., 2022. Feature-space selection with banded ridge regression. *NeuroImage*, 264, p.119728.

Nunez-Elizalde, A.O., Huth, A.G. and Gallant, J.L., 2019. Voxelwise encoding models with non-spherical multivariate normal priors. *Neuroimage*, 197, pp.482-492.

(3) Related to my comment in (2) did the authors examine the stimulus-correlations? It seems that the GPT features are strongly correlated with the low-level features and that the model with a lot of dimensions in the ridge-model explains most of the variance. Have the authors examined the implications of feature dimensions on model weights further? Given the uncertainties in the main model, I also have difficulties following the interpretations made in Fig 5 about semantic representations.

(4) The main results that the authors show is based on the reduced semantics model, however, I am not convinced that the low-level features explain the necessary low-level information (see my previous comments in (1-3) that are correlated in the results and therefore the conclusions hold.

(5) The authors used as semantic features the features extracted from one specific GPT-layer (layer 27) with the reasoning that other studies have shown that "semantic features extracted from middle layers are more accurate in predicting brain activity". Although I agree with the authors that this has been to some extent replicated using different large language models in different studies, layer-wise representational effects may remain in this unique dataset. In addition, the studies mentioned in the manuscript use smaller GPT-models or other contextual language models such as BERT and ELMo (LeBel et al. 2021, Caucheteux et al. 2023 use 12-layer GPT model, and Toneva and Wehbe, 2019 use other contextual language models such as BERT and ELMo). It would therefore still be important to run a systematic layer-by-layer analysis of the results. One possible way to present the results would be to show per layer the average prediction accuracy across significantly predicted voxels (per subject), and the average across subjects and this separately for comprehension and production. Only then can the importance of layer 27 vs. other layers or all-layers predictions can be systematically assessed, which is important for understanding how different layers relate to the semantic representation in both conversation models (comprehension vs. production) in this dataset.

(6) Following on my previous point in (6) it is advisable to create a correlation matrix of significantly predicted voxels between comprehension and production per-layer to examine the per-layer similarity between the modes. Especially, because given the complex and not-immediately interpretable representations of large language models, I am not sure why we should assume that a random middle layer 27 performs similarly for both comprehension and production. Also, have the author compared their contextual model to other contextual models?

(7) It reads like (L655-657) that the authors performed a parametric statistical test. However, this test may inflate the results and may be insufficient. A non-parametric permutation test while accounting for autocorrelation of the BOLD response is more appropriate.

(8) Have the authors looked in more detail into speech errors? In day-to-day communications it is very common that we make speech errors, so did the authors investigate the distributions of participant's speech errors and fit a speech error model?

(9) Extended Data Fig. 6 & 7 show the opposite results for P3 and P4 do the authors have an explanation for this behavior?

(10) The Semantic model construction section in Methods needs necessary details on how fine-tuning was performed. Also, the wording in L597-598 mentioned that the authors first pre-train a model wasn't the huggingface model used?

Reviewer #3:

Remarks to the Author:

Review of "Distinct yet intertwined semantic representations in the cortex during natural dialogues" submitted to *Nature Human Behavior*

Masahiro Yamashita, Rieko Kubo & Shinji Nishimoto

Brief summary and overall assessment

This paper investigates the similarity/separability of semantic representations computed by the human brain during language production and comprehension in the context of natural dialogue. To investigate this question, the authors use a voxel-wise LLM-to-brain encoding approach. The authors present a novel fMRI dataset, consisting of BOLD responses from 8 Japanese participants as they engage in 3h long spontaneous, naturalistic (topic-restricted, but unscripted/uncontrolled) conversations with the experimenter. Neural responses were recorded in participants during both production and comprehension. Conversations were transcribed and semantic representations were obtained from an openly available Japanese variant of GPT-NeoX, fine-tuned to serve as an instruction-following conversational agent (also in Japanese). The key findings include i) good predictivity from GPT model representations across many brain areas, ii) substantial overlap between comprehension and production, but also some spatial separability, and iii) a small number of (putatively interpretable) components explain neural responses in the voxels that are well-predicted by the language model.

The article tackles an ambitious and timely question and presents both an interesting novel dataset and a range of interesting analyses. The scope of the undertaking is commendable, and the quality of the figures is high. The alleged specialization for comprehension vs. production in the semantic space is intriguing. However, the manuscript suffers from a number of conceptual issues, both with the framework presented and the conclusions drawn from the results. Moreover, many analyses are not clearly justified and not well described; critical controls are often missing; the explanations of the results are often unclear and lack coherence; and the work is not well positioned within the recent literature.

More extensive paper summary

First, the authors argue that representations from a particular middle layer (layer 27) of the GPT model accurately predict voxel responses in participants across a range of cortical areas, including putative semantic areas but also sensorimotor, auditory, and visual areas. They predict responses both for comprehension and production and use leave-one-session-out cross-validation (holding out a single fMRI session and training on the rest).

Next, they ask whether the semantic representations for production and comprehension are shared or distinct. To evaluate this question, they exchange the learned regression weights for production vs. comprehension inputs at test time. They show a significant decrease in brain predictivity for the cross-modal setup, relative to the high prediction performance in the non-swapped case. Based on this, they argue that semantic representations in a voxel differ between production and comprehension.

They go on to investigate if voxels show a preference for production or comprehension, or whether each voxel represents both. They apply unique variance partitioning, and claim that low-level linguistic features, such as part-of-speech or number of morphemes do not contribute to model prediction performance. They neglect these features going forward and rely "only" on the GPT representations, which they dub "semantic representation" throughout the manuscript. Using the GPT features, the authors investigate the variance explained in each voxel by the full model (GPT features for production & comprehension), and for the two "nested models", i.e., the production/comprehension models individually. They report (a) spatially dissociated production- vs. comprehension-selective voxels, and (b) that most voxels are implicated in both production and comprehension. These results, they claim, both (a) "align with the classical framework that separates production and comprehension" and (b) "provide evidence that these semantic processes [production & comprehension] partially overlap and are interwoven in the same networks".

Next, they provide a correlational analysis as evidence for the existence of a "semantic hub", tasked with modality-invariant semantic processing: They correlate the weights of the semantic model for production and comprehension in each voxel and find positive correlations within the amodal regions (including STG/STS) and negative correlations otherwise (though the statement is unclear, see comments below).

Lastly, they investigate the specific semantic contents predominantly represented in each voxel, via a principal component analysis. They identify four significant PCs for production and six for comprehension which they claim were structured into systematic conversational interactions and social cognition, encoding things like turn-taking and backchanneling.

Major comments

I. Critical implementational details and controls are missing

The paper fails to mention or justify key implementational details and does not include key control conditions/models.

- In the Methods section, the authors report that voxel responses are modeled using an FIR model, and that the slow hemodynamic response is accounted for through concatenating features from the previous 2-7s to account for the delay ($TR=1s$). This means that instead of predicting a voxel's response from the 2,816-dimensional vector representation of GPT-NeoX (or even the concatenation of production & comprehension representations: $2 * 2,816 = 5,632$ features), the voxel response is instead predicted using $5,632 * 6 = 34,962$ features (concatenating the production & comprehension features within the 6s in the delay window, from seconds 2-7 before the current TR). That is a lot of features!

- Of course, using a lagged feature representation for FIR modeling is well-established. However, what is the justification of using concatenation over averaging or summing the feature representations within a window (e.g., Wehbe et al., 2014; Caucheteux, Gramfort & King, 2021), or down-sampling them (e.g., Jain & Huth, 2018)? And how do either of the other methods fare?

- For the PCA analysis, the authors indeed change the representation and average the features within a given time lag ("to eliminate temporal information"). Why is this approach reasonable here but not elsewhere?

- Further, given the large number of features, how does a control model, such as an untrained model of the same architecture, or a random embedding with the same number of features fare? Is the sheer number of parameters maybe already enough to achieve high predictivity? These controls are critical to be able to interpret the reported results.

- Likewise, the variance partitioning analysis, aimed at isolating the variance explained by different "low-level linguistic features", leaves several questions unanswered.

- First, these features are represented as just one, or very few features in the model (e.g., number of morphemes (an integer), or part of speech (a string)). Given the large number of what the authors call "semantic features" (i.e., GPT features) it seems unsurprising that removing these few features, while keeping the full set of GPT features (~35,000!), would lead to a negligible drop in predictivity.

- This is especially true considering that many of these features are represented (in a distributed way) in the GPT features, too. Thus, arguing that such low-level linguistic features do not contribute to the LLM-to-brain alignment is unwarranted, based on these results. In fact, other approaches have arguably tried to test similar claims more directly, by eliminating/marginalizing over information related to specific linguistic properties in the language model representations (see, e.g., Caucheteux et al., 2021 (<https://proceedings.mlr.press/v139/caucheteux21a/caucheteux21a.pdf>); Oota et al., 2022 (<https://arxiv.org/pdf/2212.08094.pdf>)).

- The manuscript underspecifies how the input was provided to the GPT model, how representations were obtained, and whether, empirically (not only conceptually), the use of fine-tuning the model as a conversational agent makes a difference. Specifically:

- The model used was trained using a specific input format (<https://huggingface.co/rinna/japanese-gpt-neox-3.6b-instruction-sft#io-format>). Were inputs prepared in the same way? I.e., indicating different speakers? It looks like the production/comprehension streams were just separated directly and fed individually to the model; see, e.g., Fig. 1). If the nested production/comprehension models did not have access to the interlocutor's content, and thus, the larger conversational context, how much do we benefit from

using the instruction-tuned model? Using this kind of model seems conceptually reasonable, but the choice should be justified better, including through prior literature (e.g., Aw et al, 2023, see below for full reference), and possibly through a comparison to a base GPT model without instruction tuning.

--It is unclear how model representations were obtained for the "semantic chunks"? Is the representation of the last token obtained (as is common for GPT models?), or is the representation of tokens summed, averaged, concatenated? This information should be added to the methods.

- Further, while the previous literature indeed shows that middle layers typically perform best in ANN-to-brain comparisons, the choice of layer 27 seems arbitrary. Especially given the novel paradigm, an across-layers comparison should be performed to confirm this choice empirically.

- Lastly, we were surprised to see the model's very high (and, to the best of our knowledge, not normalized relative to a ceiling, which is typically computed by how well a given participant's data can be predicted from other participants' data) predictivity performance (~.6). This is substantially higher than most other papers report, which suggests that some overfitting is likely happening. What is the signal-to-noise ratio in the dataset? I.e., how much variance can be explained overall? The high predictivity is especially surprising to see this given the unscripted conversations and the leave-one-session out cross-validation. Presumably most content discussed in this left out session will not be shared (except for backchanneling, etc., more on this point below).

II. Explanations/interpretations of results are not justified or are incoherent

- Through correlating the weights of the semantic model for production and comprehension, the authors report evidence in favor of "modality-invariant semantic processing" in the amodal regions, including STG/STS. However, in the section before they argue that these regions are selective for comprehension and defend a separation between production and comprehension. How do these results fit together?

- The authors report the intriguing finding that "semantic representations in a single voxel can differ between production and comprehension, even when using the same words or phrases". This conclusion is drawn based on the drop in prediction accuracy for a mapping model with swapped regression weights between production vs. comprehension at test time. This analysis is clever and the results are intriguing, but

--what data support this result (even when using the same words or phrases)? Did the authors subset the data for overlap with training utterances and showed that for this dataset, too, prediction performance dropped?

--Later in the paper the authors show that different voxels seem to be responsible for representing production vs. comprehension semantic information. What do these results look like when subsetting for voxel preference? Would the expectation be that voxels which encode both "modalities" are better predicted by this model? And what are the implications if this is/is not so?

- The authors claim that they "aimed to identify the specific semantic content that is predominantly represented in each voxel.". However, they use a group-level analysis. Given the anatomical differences of individuals' brains it seems unlikely that the same voxels would represent the same information, leaving the aim unsupported.

III. PC analysis

The methodology and interpretation of the PC analysis are unclear. This is especially important given that the PC analysis (while only being one of many analyses presented in the paper) is the main focus of the discussion section, accounting for ~3/4 of that section.

- The part-of-speech weights estimated in the voxel-wise encoding model (that were then discarded for the encoding analysis) were correlated with PC scores for the participants. Why were they included here when they were shown to not contribute to the alignment? This point is critical given that most of the conclusions are based on the analysis of the POS (and number of morphemes).

- The PC analysis returns backchanneling as a primary factor. How much do the authors think this result is an artifact of the methodology? In particular, in leave-one-out cross-validation, backchanneling arguably provides some of the few utterances that would be shared across training and test sets, and they are usually quite frequent, too. If the authors want to make some claim about this factor being important, they need to rule out this trivial explanation.

- Further, some of the conclusions drawn from the correlation analyses are premature. For example: "both PC1 for production and comprehension displayed a negative correlation solely with the weights of interjections (Fig. 5 c, f). Simultaneously, they exhibited a strong positive correlation with the weights of other parts of speech. Moreover, these components showed little association with specific semantic content. This suggests that both PC1s represent turn-taking" > why is this necessarily the case? Many other explanations are possible.

- Many of the categories identified through the PC analysis do not seem immediately meaningful: "there was a preference for proper nouns, particularly names of universities or specific locations, with fewer morphemes" >> why should this be represented? What are the implications of such a result? This and other such findings are just stated without any discussion/elaboration.

The idea of speaker- vs. listener-specific semantic interpretations needs more discussion. What does it mean, theoretically, if participants represent the same content differently when they speak it/listen to it? How does this relate to current proposals about the functional architecture of language and meaning, most of which assume that the meaning representations are shared between comprehension and production?

IV. Relevant literature is missing

The paper does not consider many relevant, recent references, which makes it difficult to situate the paper within the recent research discourse. We are providing a (non-exhaustive) list of relevant references below:

- ANN-to-brain modeling / disentangling the contribution of different features
 - o Caucheteux, C., Gramfort, A., & King, J. R. (2021, July). Disentangling syntax and semantics in the brain with deep networks. In International conference on machine learning (pp. 1336-1348). PMLR.
 - o Oota, S. R., Gupta, M., & Toneva, M. (2022). Joint processing of linguistic properties in brains and language models. arXiv preprint arXiv:2212.08094.
 - o M. Schrimpf, I. A. Blank, G. Tuckute, C. Kauf, E. A. Hosseini, N. Kanwisher, J. B. Tenenbaum, and E. Fedorenko. The neural architecture of language: Integrative modeling converges on predictive processing. *Proceedings of the National Academy of Sciences*, 118(45), 2021.
- Instruction-tuning improves LLM-to-brain encoding performance
 - o Aw, K. L., Montariol, S., AIKhamissi, B., Schrimpf, M., & Bosselut, A. (2023). Instruction-tuning Aligns LLMs to the Human Brain. arXiv preprint arXiv:2312.00575.
- Production and comprehension load onto the same network
 - o Menenti, L., Gierhan, S. M., Segaert, K., & Hagoort, P. (2011). Shared language: overlap and segregation of the neuronal infrastructure for speaking and listening revealed by functional MRI. *Psychological science*, 22(9), 1173-1182.
 - o Hu, J., Small, H., Kean, H., Takahashi, A., Zekelman, L., Kleinman, D., ... & Fedorenko, E. (2023). Precision fMRI reveals that the language-selective network supports both phrase-structure building and lexical access during language production. *Cerebral Cortex*, 33(8), 4384-4404.
- fMRI responses to dialogue vs. monologue are similar
 - o Olson, H. A., Chen, E. M., Lydic, K. O., & Saxe, R. R. (2023). Left-Hemisphere Cortical Language Regions Respond Equally to Observed Dialogue and Monologue. *Neurobiology of Language*, 1-36.
- Neural basis of speech in everyday conversations
 - o Goldstein, A., Wang, H., Niekerken, L., Zada, Z., Aubrey, B., Sheffer, T., ... & Hasson, U. (2023). Deep speech-to-text models capture the neural basis of spontaneous speech in everyday conversations. *bioRxiv*, 2023-06.

Minor comments

Clarification questions

- “By correlating the weights of the semantic model for production and comprehension (2,816 dimensions) for each voxel that exhibited significant predictions (Fig. 1d), we found positive correlations within the amodal regions ($P < 0.05$, two-sided, FDR corrected), including clusters in STG/STS (Fig. 4a and Extended Data Fig. 6). In contrast, negative correlations were observed throughout the cortex, both inside and outside the amodal regions ($P < 0.05$, two-sided, FDR corrected), encompassing clusters in the CeS and SMA (Fig. 4b and Extended Data Fig. 7).”
 - o the statement “positive correlations within amodal regions” but negative correlations [...] inside [...] the amodal regions” is confusing; please clarify where positive/negative correlations were found

Typos

- Page 2: listening to speech listening
- Page 13: First, both PC1 > should be “both PC1s”

Miscellaneous

- The term “weights of the semantic model” is unclear/ambiguous throughout the manuscript. In the Methods, semantic model refers to GPT, but in certain contexts, the term seems to refer to the learned regression weights, such as here:
 - o By correlating the weights of the semantic model for production and comprehension (2,816 dimensions) for each voxel that exhibited significant predictions [...]

Version 1:

Decision Letter:

Our ref: NATHUMBEHAV-23113888A

18th March 2025

Dear Dr Nishimoto,

Thank you for submitting your revised manuscript "Distinct yet intertwined semantic representations in the cortex during natural dialogue" (NATHUMBEHAV-23113888A). It has now been seen by two of the original referees and their comments are below. As you can see, the reviewers find that the paper has improved in revision. We will therefore be happy in principle to publish it in *Nature Human Behaviour*, pending minor revisions to satisfy the referees' final requests and to comply with our editorial and formatting guidelines.

Please note that Reviewer #2 raises a concern about how well the random embeddings model can serve as a control for low-level sensory or motor information. We ask you to provide a compelling response to this concern upon receiving our checklist.

We are now performing detailed checks on your paper and will send you a checklist detailing our editorial and formatting requirements within two weeks. Please do not upload the final materials and make any revisions until you receive this additional information from us.

Sincerely,

Nature Human Behaviour

Reviewer #1 (Remarks to the Author):

Thank you for addressing comments and questions. The new analyses and interpretations, especially with head motion correction/random embeddings/banded ridge make the story clearer and delineate how this new study fits into the context of prior perception only work.

Reviewer #2 (Remarks to the Author):

First, I would like to thank the authors for revising the manuscript to systematically address for variance associated with head motion information. I am also happy to see the layer-by-layer representations and context-length analysis. In the following I have a few remaining comments:

Comment 1

It is very surprising that the without production condition produced more head motion. How do the authors explain this? In Supplemental Fig. 2 due to the outliers, the differences between the medians are hard to compare. Could the authors create this figure without the red outlier circles. It would also be helpful to compute the statistical difference in the individual subject level instead of group average.

Comment 2

I appreciate that the authors build a separate model where they remove the head motion artifacts, as from Supplemental Fig. 3 one can see that this model predicts well in language related areas (e.g. P2, P3, P5, P7, P8). However, the description of how the motion parameters were considered needs a better explanation than the following sentence: "The predicted timeseries were subtracted from the preprocessed fMRI data to isolate the neural signals of interest."

Comment 3

The authors write in L617-618

"As a control to account for predictions potentially driven by low-level sensory or motor brain activity, we generated random normal embeddings with the same dimensionality as the GPT embeddings (2,816 features)."

Although this model is a neat model, and looking at Supplementary Fig 4 suggests that sensory regions are well predicted. I doubt it is a good model for low-level sensory information. There are remaining stimulus correlations that won't be addressed with such a random model.

Comment 4

Regarding the description about model fitting:

The authors write in the revised Results section (L676-679):

"We performed variance partitioning to quantify the unique contributions of linguistic features to BOLD responses in production, comprehension, and their intersection. Following methods from previous voxelwise modeling studies^{16,40}, we used three models: a production-only model, a comprehension-only model, and their combination (i.e., the Separate Linguistic model)."

So, did the authors run a simple regression model with only comprehension and only production data and the corresponding embeddings separately?

$Y_{\text{comprehension}} = X_{\text{comprehension}} * w + \text{epsilon}$

$Y_{\text{production}} = X_{\text{production}} * w + \text{epsilon}$

What do the authors mean by the following to statements:

- L685-686: "To maintain consistency with correlation coefficients, square roots of variance components were reported."

- L686-688: "For TRs containing single (either production or comprehension) or overlapping conditions (production and comprehension), we applied variance partitioning consistently."

Comment 5

L627-628: Following statement needs citations:

"A distance of 50 mm between the cerebral cortex and the head center was assumed in accordance with prior studies."

Comment 6

EVC labels in all the Figures where the cortical flatmaps are shown are quite misplaced.

Version 2:

Decision Letter:

Dear Prof Nishimoto,

We are pleased to inform you that your Article "Conversational content is organized across multiple timescales in the brain", has now been accepted for publication in Nature Human Behaviour.

With best regards,

Nature Human Behaviour

P.S. Click on the following link if you would like to recommend Nature Human Behaviour to your librarian
<http://www.nature.com/subscriptions/recommend.html#forms>

** Visit the Springer Nature Editorial and Publishing website at http://editorial-jobs.springernature.com?utm_source=ejp_NHumB_email&utm_medium=ejp_NHumB_email&utm_campaign=ejp_NHumB for more information about our career opportunities. If you have any questions please click [here](mailto:editorial.publishing.jobs@springernature.com). **

REVIEWER COMMENTS:

Reviewer #1:

Remarks to the Author:

In this work, the authors used voxelwise encoding modeling to explore brain representations during real-world conversations. Specifically, the study comprised human participants engaging in natural conversations with an experimenter across a variety of topics, while their BOLD activity was being recorded with fMRI. To model this data, the authors extracted features-of-interest for the word transcripts in each TR, doing so separately for produced and perceived instances. These features included part-of-speech, morpheme count etc. as well as the hidden state of GPTNeoX (upper-middle layer). Then, these features were used to build various linearized predictive models of the BOLD activity of each voxel individually and the models were evaluated on a held-out test set. The authors found good prediction performance broadly across the cerebral cortex. Next, they evaluated if the encoding model weights were transferable between production and perception, and what the correlation between these weights were. Overall, the weights were not transferable and were negatively correlated across cortex barring low-level auditory regions around the superior temporal gyrus. Then the authors evaluated the unique variance explained by production and perception, finding that a majority of voxels were well predicted by the linguistic content in both conditions while a handful of voxels were only predicted by the linguistic content of one or the other. Finally, the authors did PCA on the encoding weights for each condition and interpreted the information captured by each significant PC. They found that the PCs largely captured information related to conversations like turn-taking, backtracking etc. as opposed to the semantic distinctions reported in prior studies that consider pure perception.

Strengths:

This paper brings interesting methods to the nascent field of neural representations during naturalistic conversations. To the best of my knowledge, it is the first study to apply voxelwise encoding modeling analyses to such an experimental design. This preserves individual participant resolution lost in prior studies that instead mainly relied on inter-subject correlation. The study also highlights an important gap in the field by showing that many of the variations in voxels function during conversations do not in fact correspond to purely semantic variations observed in studies that only study perception. By jointly studying perception and production, we can make stronger claims about the functional role of each brain region.

I particularly found that lack of weight transfer between the two conditions and the conversation (but not semantic category) based interpretations of the encoding weight PCs very useful in understanding how voxels engaged in both production and perception might represent different, conversation-related information during each stage.

Weaknesses & Questions: Overall, the conclusions regarding purported functional roles of voxels during perception and production was confusing, as were the interpretations of the weights-based analyses.

- Why is Fig. 3 labeled as "semantic selectivity"? Also, wouldn't this analysis be sensitive to whether both production and perception happened every TR? Can the authors elaborate on the statistics of the transcripts, like the word rate for production and perception, if there are TRs comprising only one condition and how such TRs were treated?

- Can the authors elaborate on the lack of successful weight transfer between production and comprehension even in voxels that jointly represent the two? This is very closely related to the correlation experiment and should be grouped together. Also, aren't the strong "positive" correlations, suggestive of preserved semantic selectivity, only around auditory cortex (expectedly due to perception and auditory feedback during production)?

- In the Fig. 5 experiment:

- Were the PCs estimated on weights of the joint model, or production and perception utterances separately? Which voxels went into the estimation for each condition? I.e., were the prod PCs estimated on “prod-only” and “both” voxels but not “comp only”, or, on all significantly predicted voxels in this condition?
- The correlation between interjections and ProdPC 1 seems to be very small. From Extended Fig. 10, it also seems that this PC is high in “comprehension only” regions like parts of the STG. How do the authors reconcile this with the claim that the first PCs capture turn-taking.
- It could be interesting to plot the distribution of each PC across voxels that are “production-only”, “comprehension-only” or both. Are there consistent patterns here? One might expect that “prod-only” voxels have very high ProcPC1 and low CompPC1, for example.
- The labels of the “semantic” features are based on the authors’ qualitative interpretations as far as I can tell. I would recommend running a more systematic behavioral experiment across multiple labelers as in Huth et al., 2016. Can the authors also elaborate why they refer to these PCs as “semantic”? How can we be assured that these differences are purely semantic in nature? This also affects the speculative discussion on why these PCs might arise in the first place. Without a strong assessment for interpreting each PC, it is difficult to understand their cortical distributions.
- While an interesting and novel analyses, the conclusion is not clear. What does it mean for the perception and production phases to have different PC interpretations? How does this relate to purported shared/ different computational mechanisms in each voxel for the two conditions?
- What types of semantic concepts did the conversations cover? I am interested in understanding why the traditional semantic PCs uncovered during pure perception (for ex., “social” vs. “visual” vs. “tactile”) don’t come up here even in the later components- is this an artifact of not covering semantically-rich content, or, is it fundamental to real-world communication that earlier studies have missed? This also relates to a potentially critical claim the paper makes re “our results suggest that the foundational components of cortical semantic representations are formed based on universal systematic communication rules, regardless of the specific semantic content.” What does this result say about the purported function of voxels ascribed a specific semantic selectivity in prior work and the conversation-related selectivity found here?
- Throughout this study, the visual cortex is significantly predicted by the encoding models and is even attributed specific colors in the RGB PC space. Could this be because of eye motion during the task that need to be regressed out?
- Nits:
- Which areas correspond to the “language network” being referred to in lines 369-378? I believe it is not Ev Fedorenko’s definition but perhaps “all regions activated during language production and comprehension”?
- Perhaps reducing the curvature contrast will improve the visibility of RGB colors in the flatmaps.
- It might be useful to add one sentence about the FIR model while explaining the encoding model in the main text.

Reviewer #2:

Remarks to the Author:

The exploration of semantic representations during natural dialogue remains a relatively understudied yet highly pertinent area of research. Understanding how the human brain encodes linguistic information in real-world conversational settings remains unresolved. Previous linguistic studies have predominantly concentrated on language comprehension, often employing controlled linguistic stimuli like isolated words or sentences. Recent studies in language comprehension, utilizing narrative stories together with voxelwise encoding modeling, have shown a distributed network of brain regions responsible for representing semantic information, largely irrespective of the presentation modality. Nevertheless, the scarcity of studies that examined natural language production and/or interactive conversations has been limiting our understanding of linguistic representations in the brain.

The authors of this study are tackling this research gap within language research asking participants to engage in natural conversations on predefined topics while recording their brain activity using functional MRI. Subsequently, they extracted language comprehension and production features derived from spoken stimuli (either spoken by the participant or the experimenter) using large language models, such as GPT and applied a voxelwise encoding modeling framework to map semantic representations onto brain activity. The authors systematically explore both the distinct and shared semantic representations during language comprehension and production and report both distinct and overlapping representations. Importantly, their analysis extends to a detailed examination of voxel-level representations of semantic information, interpreting brain regions responsive to turn-taking interactions, backchannels, and self-mentalizing.

One reason why the current literature has been rather scarce to study brain representations of natural conversations is that it is really hard to conduct functional MRI experiments where subjects can freely speak during the experiment without fundamentally distorting the acquired signal through head motion. It is therefore crucial to present that potential motion artefacts have been investigated thoroughly. In its current form the manuscript is lacking the necessary detail to understand the affects of motion and low-level information on the results. Hence, I have several questions and comments pertaining to the data analysis, which I have listed below.

(1) Given that head motion can have a significant effect on acquired BOLD signal, it is important see more detailed analysis of potential motion artifacts and how they may be affecting the results. It would be a good start to examine the motion traces per individual in the cortical voxels. The motion parameters have been added to the analysis as nuisance regressors. However, I am very surprised to see in Extended Data Fig. 3 that motion (and other features but see my next comment for the other features) have no unique variance explained that seems to be very unlikely, especially for the production data. Did the authors plot head motion model separately in a ridge model? Did the authors have systematically and separately investigate the effect of speaking in the scanner?

(2) I have a hard time following what the authors did in their variance partitioning. It seems that the authors only computed the difference between the full model and a reduced model. Has this been done for all the different feature spaces separately? The methods section lacks the appropriate detail here and needs improvement. Please see the cited papers Le Bel et al. 2021 and de Heer et al. 2017 for a better description. Based on this I am also curious whether the authors fit each model as a separate ridge model? Given de Heer et al. 2017 it is very surprising that the low-level features are very poorly predicting in speech-related regions (Extended Data Fig. 3). Given that there may be voxels that share representations of GPT features and e.g. part-of-speech features it would be more beneficial to use a banded ridge model [la Tour et al. 2022; Nunez-Elizalde et al. 2019] and examine the split-correlations of the prediction accuracies across different models.

References:

- la Tour, T.D., Eickenberg, M., Nunez-Elizalde, A.O. and Gallant, J.L., 2022. Feature-space selection with banded ridge regression. *NeuroImage*, 264, p.119728.
- Nunez-Elizalde, A.O., Huth, A.G. and Gallant, J.L., 2019. Voxelwise encoding models with non-spherical multivariate normal priors. *Neuroimage*, 197, pp.482-492.

(3) Related to my comment in (2) did the authors examine the stimulus-correlations? It seems that the GPT features are strongly correlated with the low-level features and that the model with a lot of dimensions in the ridge-model explains most of the variance. Have the authors examined the implications of feature dimensions on model weights further? Given the uncertainties in the main model, I also have difficulties following the interpretations made in Fig 5 about semantic representations.

(4) The main results that the authors show is based on the reduced semantics model, however, I am not convinced that the low-level features explain the necessary low-level information (see my previous comments in (1-3) that are correlated in the results and therefore the conclusions hold.

(5) The authors used as semantic features the features extracted from one specific GPT-layer (layer 27) with the reasoning that other studies have shown that “semantic features extracted from middle layers are more accurate in predicting brain activity”. Although I agree with the authors that this has been to some extent replicated using different large language models in different studies, layer-wise representational effects may remain in this unique dataset. In addition, the studies mentioned in the manuscript use smaller GPT-models or other contextual language models such as BERT and ELMo (LeBel et al. 2021, Caucheteux et al. 2023 use 12-layer GPT model, and Toneva and Wehbe, 2019 use other contextual language models such as BERT and ELMo). It would therefore still be important to run a systematic layer-by-layer analysis of the results. One possible way to present the results would be to show per layer the average prediction accuracy across significantly predicted voxels (per subject), and the average across subjects and this separately for comprehension and production. Only then can the importance of layer 27 vs. other layers or all-layers predictions can be systematically assessed, which is important for understanding how different layers relate to the semantic representation in both conversation models (comprehension vs. production) in this dataset.

(6) Following on my previous point in (6) it is advisable to create a correlation matrix of significantly predicted voxels between comprehension and production per-layer to examine the per-layer similarity between the modes. Especially, because given the complex and not-immediately interpretable representations of large language models, I am not sure why we should assume that a random middle layer 27 performs similarly for both comprehension and production. Also, have the author compared their contextual model to other contextual models?

(7) It reads like (L655-657) that the authors performed a parametric statistical test. However, this test may inflate the results and may be insufficient. A non-parameteric permutation test while accounting for autocorrelation of the BOLD response is more appropriate.

(8) Have the authors looked in more detail into speech errors? In day-to-day communications it is very common that we make speech errors, so did the authors investigate the distributions of participant's speech errors and fit a speech error model?

(9) Extended Data Fig. 6 & 7 show the opposite results for P3 and P4 do the authors have an explanation for this behavior?

(10) The Semantic model construction section in Methods needs necessary details on how fine-tuning was performed. Also, the wording in L597-598 mentioned that the authors first pre-train a model wasn't the huggingface model used?

Reviewer #3:

Remarks to the Author:

Review of “Distinct yet intertwined semantic representations in the cortex during natural dialogues” submitted to Nature Human Behavior

Masahiro Yamashita, Rieko Kubo & Shinji Nishimoto

Brief summary and overall assessment

This paper investigates the similarity/separability of semantic representations computed by the human brain during language production and comprehension in the context of natural dialogue. To investigate this question, the authors use a voxel-wise LLM-to-brain encoding approach. The authors present a novel fMRI dataset, consisting of BOLD responses from 8 Japanese participants as they engage in 3h long spontaneous, naturalistic (topic-restricted, but unscripted/uncontrolled) conversations with the experimenter. Neural responses were recorded in participants during both production and comprehension. Conversations were transcribed and semantic representations were obtained from an openly available Japanese variant of GPT-NeoX, fine-tuned to serve as an instruction-following conversational agent (also in Japanese). The key findings include i) good predictivity from GPT model representations across many brain areas, ii) substantial overlap between comprehension and production, but also some spatial separability, and iii) a small number of (putatively interpretable) components explain neural responses in the voxels that are well-predicted by the language model.

The article tackles an ambitious and timely question and presents both an interesting novel dataset and a range of interesting analyses. The scope of the undertaking is commendable, and the quality of the figures is high. The alleged specialization for comprehension vs. production in the semantic space is intriguing. However, the manuscript suffers from a number of conceptual issues, both with the framework presented and the conclusions drawn from the results. Moreover, many analyses are not clearly justified and not well described; critical controls are often missing; the explanations of the results are often unclear and lack coherence; and the work is not well positioned within the recent literature.

More extensive paper summary

First, the authors argue that representations from a particular middle layer (layer 27) of the GPT model accurately predict voxel responses in participants across a range of cortical areas, including putative semantic areas but also sensorimotor, auditory, and visual areas. They predict responses both for comprehension and production and use leave-one-session-out cross-validation (holding out a single fMRI session and training on the rest).

Next, they ask whether the semantic representations for production and comprehension are shared or distinct. To evaluate this question, they exchange the learned regression weights for production vs. comprehension inputs at test time. They show a significant decrease in brain predictivity for the cross-modal setup, relative to the high prediction performance in the non-swapped case. Based on this, they argue that semantic representations in a voxel differ between production and comprehension.

They go on to investigate if voxels show a preference for production or comprehension, or whether each voxel represents both. They apply unique variance partitioning, and claim that low-level linguistic features, such as part-of-speech or number of morphemes do not contribute to model prediction performance. They neglect these features going forward and rely “only” on the GPT representations, which they dub “semantic representation” throughout the manuscript. Using the GPT features, the authors investigate the variance explained in each voxel by the full model (GPT features for production & comprehension), and for the two “nested models”, i.e., the production/comprehension models individually. They report (a) spatially dissociated production- vs. comprehension-selective voxels, and (b) that most voxels are implicated in both production and comprehension. These results, they claim, both (a) “align with the classical framework that separates production and comprehension” and (b) “provide evidence that these semantic processes [production & comprehension] partially overlap and are interwoven in the same networks”.

Next, they provide a correlational analysis as evidence for the existence of a “semantic hub”, tasked with modality-invariant semantic processing: They correlate the weights of the semantic model for production and comprehension in each voxel and find positive correlations within the amodal regions (including STG/STS) and negative correlations otherwise (though the statement is unclear, see comments below).

Lastly, they investigate the specific semantic contents predominantly represented in each voxel, via a principal component analysis. They identify four significant PCs for production and six for comprehension which they claim were structured into systematic conversational interactions and social cognition, encoding things like turn-taking and backchanneling.

Major comments

I. Critical implementational details and controls are missing

The paper fails to mention or justify key implementational details and does not include key control conditions/models.

- In the Methods section, the authors report that voxel responses are modeled using an FIR model, and that the slow hemodynamic response is accounted for through concatenating features from the previous 2-7s to account for the delay ($TR=1s$). This means that instead of predicting a voxel's response from the 2,816-dimensional vector representation of GPT-NeoX (or even the concatenation of production & comprehension representations: $2 * 2,816 = 5,632$ features), the voxel response is instead predicted using $5,632 * 6 = 34,962$ features (concatenating the production & comprehension features within the 6s in the delay window, from seconds 2-7 before the current TR). That is a lot of features!

--Of course, using a lagged feature representation for FIR modeling is well-established. However, what is the justification of using concatenation over averaging or summing the feature representations within a window (e.g., Wehbe et al., 2014; Caucheteux, Gramfort & King, 2021), or down-sampling them (e.g., Jain & Huth, 2018)? And how do either of the other methods fare?

--For the PCA analysis, the authors indeed change the representation and average the features within a given time lag ("to eliminate temporal information"). Why is this approach reasonable here but not elsewhere?

--Further, given the large number of features, how does a control model, such as an untrained model of the same architecture, or a random embedding with the same number of features fare? Is the sheer number of parameters maybe already enough to achieve high predictivity? These controls are critical to be able to interpret the reported results.

- Likewise, the variance partitioning analysis, aimed at isolating the variance explained by different "low-level linguistic features", leaves several questions unanswered.

--First, these features are represented as just one, or very few features in the model (e.g., number of morphemes (an integer), or part of speech (a string)). Given the large number of what the authors call "semantic features" (i.e., GPT features) it seems unsurprising that removing these few features, while keeping the full set of GPT features (~35,000!), would lead to a negligible drop in predictivity.

--This is especially true considering that many of these features are represented (in a distributed way) in the GPT features, too. Thus, arguing that such low-level linguistic features do not contribute to the LLM-to-brain alignment is unwarranted, based on these results. In fact, other approaches have arguably tried to test similar claims more directly, by eliminating/marginalizing over information related to specific linguistic properties in the language model representations (see, e.g., Caucheteux et al., 2021 (<https://proceedings.mlr.press/v139/caucheteux21a/caucheteux21a.pdf>); Oota et al., 2022 (<https://arxiv.org/pdf/2212.08094.pdf>)).

- The manuscript underspecifies how the input was provided to the GPT model, how representations were obtained, and whether, empirically (not only conceptually), the use of fine-tuning the model as a conversational agent makes a difference. Specifically:

--The model used was trained using a specific input format (<https://huggingface.co/rinna/japanese-gpt-neox-3.6b-instruction-sft#io-format>). Were inputs prepared in the same way? I.e., indicating different speakers? It looks like the production/comprehension streams were just separated directly and fed

individually to the model; see, e.g., Fig. 1). If the nested production/comprehension models did not have access to the interlocutor's content, and thus, the larger conversational context, how much do we benefit from using the instruction-tuned model? Using this kind of model seems conceptually reasonable, but the choice should be justified better, including through prior literature (e.g., Aw et al, 2023, see below for full reference), and possibly through a comparison to a base GPT model without instruction tuning.

--It is unclear how model representations were obtained for the "semantic chunks"? Is the representation of the last token obtained (as is common for GPT models?), or is the representation of tokens summed, averaged, concatenated? This information should be added to the methods.

- Further, while the previous literature indeed shows that middle layers typically perform best in ANN-to-brain comparisons, the choice of layer 27 seems arbitrary. Especially given the novel paradigm, an across-layers comparison should be performed to confirm this choice empirically.
- Lastly, we were surprised to see the model's very high (and, to the best of our knowledge, not normalized relative to a ceiling, which is typically computed by how well a given participant's data can be predicted from other participants' data) predictivity performance (~.6). This is substantially higher than most other papers report, which suggests that some overfitting is likely happening. What is the signal-to-noise ratio in the dataset? I.e., how much variance can be explained overall? The high predictivity is especially surprising to see this given the unscripted conversations and the leave-one-session out cross-validation. Presumably most content discussed in this left out session will not be shared (except for backchanneling, etc., more on this point below).

II. Explanations/interpretations of results are not justified or are incoherent

- Through correlating the weights of the semantic model for production and comprehension, the authors report evidence in favor of "modality-invariant semantic processing" in the amodal regions, including STG/STS. However, in the section before they argue that these regions are selective for comprehension and defend a separation between production and comprehension. How do these results fit together?
- The authors report the intriguing finding that "semantic representations in a single voxel can differ between production and comprehension, even when using the same words or phrases". This conclusion is drawn based on the drop in prediction accuracy for a mapping model with swapped regression weights between production vs. comprehension at test time. This analysis is clever and the results are intriguing, but --what data support this result (even when using the same words or phrases)? Did the authors subset the data for overlap with training utterances and showed that for this dataset, too, prediction performance dropped?
--Later in the paper the authors show that different voxels seem to be responsible for representing production vs. comprehension semantic information. What do these results look like when subsetting for voxel preference? Would the expectation be that voxels which encode both "modalities" are better predicted by this model? And what are the implications if this is/is not so?
- The authors claim that they "aimed to identify the specific semantic content that is predominantly represented in each voxel.". However, they use a group-level analysis. Given the anatomical differences of individuals' brains it seems unlikely that the same voxels would represent the same information, leaving the aim unsupported.

III. PC analysis

The methodology and interpretation of the PC analysis are unclear. This is especially important given that the PC analysis (while only being one of many analyses presented in the paper) is the main focus of the discussion section, accounting for ~3/4 of that section.

- The part-of-speech weights estimated in the voxel-wise encoding model (that were then discarded for the encoding analysis) were correlated with PC scores for the participants. Why were they included here when they were shown to not contribute to the alignment? This point is critical given that most of the conclusions are based on the analysis of the POS (and number of morphemes).
- The PC analysis returns backchanneling as a primary factor. How much do the authors think this result is an artifact of the methodology? In particular, in leave-one-out cross-validation, backchanneling arguably provides some of the few utterances that would be shared across training and test sets, and they are usually quite frequent, too. If the authors want to make some claim about this factor being important, they need to rule out this trivial explanation.
- Further, some of the conclusions drawn from the correlation analyses are premature. For example: “both PC1 for production and comprehension displayed a negative correlation solely with the weights of interjections (Fig. 5 c, f). Simultaneously, they exhibited a strong positive correlation with the weights of other parts of speech. Moreover, these components showed little association with specific semantic content. This suggests that both PC1s represent turn-taking” > why is this necessarily the case? Many other explanations are possible.
- Many of the categories identified through the PC analysis do not seem immediately meaningful: “there was a preference for proper nouns, particularly names of universities or specific locations, with fewer morphemes” >> why should this be represented? What are the implications of such a result? This and other such findings are just stated without any discussion/elaboration.

The idea of speaker- vs. listener-specific semantic interpretations needs more discussion. What does it mean, theoretically, if participants represent the same content differently when they speak it/listen to it? How does this relate to current proposals about the functional architecture of language and meaning, most of which assume that the meaning representations are shared between comprehension and production?

IV. Relevant literature is missing

The paper does not consider many relevant, recent references, which makes it difficult to situate the paper within the recent research discourse. We are providing a (non-exhaustive) list of relevant references below:

- ANN-to-brain modeling / disentangling the contribution of different features
 - o Caucheteux, C., Gramfort, A., & King, J. R. (2021, July). Disentangling syntax and semantics in the brain with deep networks. In International conference on machine learning (pp. 1336-1348). PMLR.
 - o Oota, S. R., Gupta, M., & Toneva, M. (2022). Joint processing of linguistic properties in brains and language models. arXiv preprint arXiv:2212.08094.
 - o M. Schrimpf, I. A. Blank, G. Tuckute, C. Kauf, E. A. Hosseini, N. Kanwisher, J. B. Tenenbaum, and E. Fedorenko. The neural architecture of language: Integrative modeling converges on predictive processing. Proceedings of the National Academy of Sciences, 118(45), 2021.
- Instruction-tuning improves LLM-to-brain encoding performance
 - o Aw, K. L., Montariol, S., Alkhamissi, B., Schrimpf, M., & Bosselut, A. (2023). Instruction-tuning Aligns LLMs to the Human Brain. arXiv preprint arXiv:2312.00575.
- Production and comprehension load onto the same network
 - o Menenti, L., Gierhan, S. M., Segaert, K., & Hagoort, P. (2011). Shared language: overlap and segregation of the neuronal infrastructure for speaking and listening revealed by functional MRI. Psychological science, 22(9), 1173-1182.

o Hu, J., Small, H., Kean, H., Takahashi, A., Zekelman, L., Kleinman, D., ... & Fedorenko, E. (2023). Precision fMRI reveals that the language-selective network supports both phrase-structure building and lexical access during language production. *Cerebral Cortex*, 33(8), 4384-4404.

- fMRI responses to dialogue vs. monologue are similar

o Olson, H. A., Chen, E. M., Lydic, K. O., & Saxe, R. R. (2023). Left-Hemisphere Cortical Language Regions Respond Equally to Observed Dialogue and Monologue. *Neurobiology of Language*, 1-36.

- Neural basis of speech in everyday conversations

o Goldstein, A., Wang, H., Niekerken, L., Zada, Z., Aubrey, B., Sheffer, T., ... & Hasson, U. (2023). Deep speech-to-text models capture the neural basis of spontaneous speech in everyday conversations. *bioRxiv*, 2023-06.

Minor comments

Clarification questions

- “By correlating the weights of the semantic model for production and comprehension (2,816 dimensions) for each voxel that exhibited significant predictions (Fig. 1d), we found positive correlations within the amodal regions ($P < 0.05$, two-sided, FDR corrected), including clusters in STG/STS (Fig. 4a and Extended Data Fig. 6). In contrast, negative correlations were observed throughout the cortex, both inside and outside the amodal regions ($P < 0.05$, two-sided, FDR corrected), encompassing clusters in the CeS and SMA (Fig. 4b and Extended Data Fig. 7).”

o the statement “positive correlations within amodal regions” but negative correlations [...] inside [...] the amodal regions” is confusing; please clarify where positive/negative correlations were found

Typos

- Page 2: listening to speech listening
- Page 13: First, both PC1 > should be “both PC1s”

Miscellaneous

- The term “weights of the semantic model” is unclear/ambiguous throughout the manuscript. In the Methods, semantic model refers to GPT, but in certain contexts, the term seems to refer to the learned regression weights, such as here:

o By correlating the weights of the semantic model for production and comprehension (2,816 dimensions) for each voxel that exhibited significant predictions [...]

We sincerely thank the reviewers for their insightful comments and constructive suggestions, which have greatly improved the quality of our manuscript. In response, we have made substantial revisions to enhance the clarity and robustness of our findings. Below, we summarize the key updates:

(i) Expanded analysis of layers and context lengths:

Previously, our analysis was limited to GPT layer 27 with a 1-second context length. In the revised manuscript, we performed a comprehensive analysis across a broader range of context lengths (1, 2, 4, 8, 16, 32 s) and layers (0, 3, 6, ..., 36). This extended analysis highlights how linguistic representations are modulated across different timescales and layers.

(ii) Three-step model assignment:

To enhance interpretability, we replaced the single-step ridge regression approach with a three-step banded ridge regression procedure:

- Step 1: Controlling for head motion.
- Step 2: Removing lower-level signals using random normal embeddings.
- Step 3: Modeling higher-level linguistic representations with contextual linguistic embeddings.

This approach effectively isolates higher-level linguistic representations from lower-level motor and sensory signals.

We believe that the revisions we have made significantly strengthen the manuscript and contribute to a fundamental understanding of how the human brain represents conversational content.

For ease of review, our responses are structured to address each reviewer comment in turn, with the reviewer comments presented in green.

Reviewer #1:

Comment 1-1: Why is Fig. 3 labeled as “semantic selectivity”? Also, wouldn’t this analysis be sensitive to whether both production and perception happened every TR? Can the authors elaborate on the statistics of the transcripts, like the word rate for production and perception, if there are TRs comprising only one condition and how such TRs were treated?

Answer: We address the questions in the following points.

1-1-1. Labeling as “semantic selectivity”:

The term “semantic” was adopted from a recent study (Tang et al., 2023, Nature Neuroscience; <https://www.nature.com/articles/s41593-023-01304-9>) that used voxelwise modeling with GPT features, similar to our approach, and referred to the analysis as “semantic reconstruction.” The term “selectivity” was derived from earlier studies that used voxelwise modeling to compare multiple features (e.g., de Heer et al., 2017; Deniz et al., 2019). However, after careful consideration, we have decided not to use the term “semantic selectivity” in our current manuscript. Instead, we use the broader term “linguistic” to encompass both semantic and syntactic information captured by GPT models, in line with recent research (Caucheteux et al., 2021; Oota et al., 2023):

“Throughout the paper, the term linguistic encompasses both semantic and syntactic dimensions, reflecting their tight interdependence (Caucheteux et al., 2021; Oota et al., 2023).” (lines 65–67 on page 3)

1-1-2. Sensitivity to production and perception occurring in every TR:

You raised an important point about the potential sensitivity of our analysis to the representation of both production and perception within each TR. Ideally, production and comprehension data should be equally represented in each TR for a fair comparison. However, in natural conversation settings, it is common for one person to speak while the other listens, resulting in an unequal distribution of production and

comprehension TRs in our fMRI data. Our analysis confirmed that the number of TRs with both production and perception was relatively small. We have included these statistics in Supplementary Fig. 1 to provide a clear picture of the distribution. Moreover, we have examined these effects:

“To mitigate potential biases in variance partitioning results due to disparities in sample sizes, we examined the correlation between production-to-comprehension sample size ratios and the corresponding variance explained. A significant correlation was observed for early layers at a 1-second context length (layers 6 and 15; Spearman’s rank correlation $\rho = 0.93$, $P < 0.05$, FDR corrected). Participants who produced more speech demonstrated greater variance explained by production under these conditions. Importantly, sample proportions were balanced overall, with four participants producing more speech (P3, P4, P6, P7) and the remaining four comprehending more (P1, P2, P5, P8). These findings confirm that variance partitioning results were not systematically biased toward either modality across participants.” (lines 207–215, pages 9–10)

Therefore, the potential effect on our results and conclusions (Figs. 3 and 4) was limited.

1-1-3. How TRs containing only one condition are treated:

For TRs containing only one condition (either production or comprehension) and TRs containing both conditions, we consistently applied variance partitioning in our analysis. This procedure is described in detail in the Methods section under "Variance partitioning":

“For TRs containing single (either production or comprehension) or overlapping conditions (production and comprehension), we applied variance partitioning consistently.” (lines 686–688 on page 27)

Comment 1-2: Can the authors elaborate on the lack of successful weight transfer between production and comprehension even in voxels that jointly represent the two? This is very closely related to the correlation experiment and should be grouped together. Also, aren't the strong "positive" correlations, suggestive of preserved semantic selectivity, only around auditory cortex (expectedly due to perception and auditory feedback during production)?

Answer: We have revised our manuscript to address these concerns.

1-2-1. Lack of successful weight transfer between production and comprehension:

In our current manuscript, we observed a greater number of voxels showing weight transfer between production and comprehension compared to the previous manuscript. This improvement results from our new modeling approach, which reduces the influence of lower-level factors such as motor and auditory processing. To specifically target brain activity related to higher-level linguistic representations, we removed the variance in BOLD responses predicted by head motion and random embeddings, as detailed in the Supplementary Results. After controlling for these confounding factors, we identified “cross-modal voxels” where the weights generalized across both production and comprehension (Fig. 2). Compared to our previous manuscript (Fig. 2 in the previous manuscript), we now report a larger number of cross-modal voxels, with weak but positive weight correlations between modalities (mean $r \sim 0.3$; Supplementary Fig. 10).

Regarding the limited weight transfer between production and comprehension, our revised manuscript addresses this issue more directly. We identified a set of “bimodal voxels” that are selectively tuned to both production and comprehension but show little to no correlation in weights between these two modalities ($-0.1 < \text{mean } r < 0.15$; Supplementary Fig. 12). This suggests that the linguistic representations in these bimodal voxels are largely independent across two modalities. These results are discussed in detail in the subsection “Dual linguistic representations in bimodal voxels” (lines 227–244, page 11).

1-2-2. Grouping of correlation experiments:

Following your suggestion, we grouped the correlation experiments to provide a clearer understanding of linguistic tuning across different voxel types. Our analysis revealed that cross-modal voxels exhibit weak positive weight correlations (mean $r \sim 0.3$; Fig. 2d), while linguistic, bimodal, production, and comprehension voxels show uncorrelated weights (Fig. 2d and Fig. 4d). These results indicate that linguistic representations are aligned in cross-modal voxels but remain independent in other voxel types.

1-2-3. Localization of strong positive correlations:

In the previous manuscript (Extended Data Fig. 2), we found that cross-modal voxels were primarily located around the auditory cortex, likely due to auditory feedback during speech production. However, in the current analysis, after controlling for low-level auditory factors, we found that shared representations extended to more distributed brain regions, including the prefrontal, temporal, and parietal cortices. These regions appear to be involved in higher-level semantic and syntactic processing, rather than simply reflecting auditory feedback. This suggests that the influence of low-level auditory feedback on our earlier findings was minimal.

Comment 1-3: Were the PCs estimated on weights of the joint model, or production and perception utterances separately? Which voxels went into the estimation for each condition? I.e., were the prod PCs estimated on “prod-only” and “both” voxels but not “comp only”, or, on all significantly predicted voxels in this condition?

Answer: In our current manuscript, we implemented a banded ridge regression approach instead of the ordinary ridge regression used in our previous manuscript. This change has led to slight modifications in how we perform PCA, which we detail below.

1-3-1. PCA on the weights of the joint model:

We performed PCA on weights of the joint model. In the revised manuscript, our joint model (the Separate Linguistic model) includes only GPT features for both production and comprehension (Fig. 1):

“To elucidate the linguistic organization underlying conversational content, we conducted principal component analysis (PCA) on the 2,816-dimensional Separate Linguistic model weights for each modality and participant.” (lines 275–277, page 13)

“Due to variability in conversational content, PCA was conducted separately for each participant and independently for production and comprehension, yielding modality-specific principal components (PCs).” (lines 278–281, page 13)

1-3-2. Voxels used in the PCA:

For the PCA, we included all cortical voxels identified for each participant. This is specified in Methods section:

“PCA was performed separately for each modality on these scaled weights in all cortical voxels (2,816 weights x all cortical voxels), yielding 2,816 orthogonal PCs.” (lines 695–696, page 28)

Comment 1-4: The correlation between interjections and ProdPC 1 seems to be very small. From Extended Fig. 10, it also seems that this PC is high in “comprehension only” regions like parts of the STG. How do the authors reconcile this with the claim that the first PCs capture turn-taking.

Answer: As we mentioned above (response 1-2-1), the results in our previous manuscript were influenced by low-level motor and auditory factors. As a result, it was difficult to separate these low-level processes from higher-level linguistic processes. To address this issue, we used random embedding features to control for these low-level motor and auditory contributions to the BOLD responses (Supplementary Fig. 4). After this adjustment, we found that what was previously interpreted as the first PC observed in both production

and comprehension was largely driven by these low-level processes. Therefore, we no longer claim that the first PCs capture turn-taking, as it is more accurate to attribute them to these lower-level contributions.

Comment 1-5: It could be interesting to plot the distribution of each PC across voxels that are “production-only”, “comprehension-only” or both. Are there consistent patterns here? One might expect that “prod-only” voxels have very high ProcPC1 and low CompPC1, for example.

Answer: In our revised manuscript, identifying consistent spatial patterns of PCs across participants is challenging for several reasons. Unlike our previous manuscript, where we performed group-level PCA under a single condition (context length of 1 second for GPT layer 27 in each modality), our current analysis involves performing PCA individually for each participant, layer, and timescale. This methodological change has substantially increased the number of PCA conditions (participant, modality, context length, layer = $8 \times 2 \times 6 \times 13 = 1,248$ combinations), making direct comparison of voxel patterns across conditions difficult. Even when we focus on a specific condition (e.g., context length of 1 second for layer 0, as shown in Fig. 5), there is substantial individual variability in the spatial distribution of PCA results. Given these individual differences, we have chosen not to present cortical surface maps of PCA results in our revised manuscript.

Comment 1-6: The labels of the “semantic” features are based on the authors’ qualitative interpretations as far as I can tell. I would recommend running a more systematic behavioral experiment across multiple labelers as in Huth et al., 2016. Can the authors also elaborate why they refer to these PCs as “semantic”? How can we be assured that these differences are purely semantic in nature? This also affects the speculative discussion on why these PCs might arise in the first place. Without a strong assessment for interpreting each PC, it is difficult to understand their cortical distributions.

Answer: We address the questions in the following points.

1-6-1. Systematic behavioral experiment:

We acknowledge that our previous manuscript relied on subjective interpretations rather than a systematic behavioral experiment. While such an experiment could provide more objective insights, it might also introduce variability due to differences in the labelers' backgrounds in linguistics and neuroscience. To address this, we utilized ChatGPT (GPT-4o) for a consistent interpretation process across all participants.

For our current analysis, we focused on a context length of 1 s for layer 0, which yielded the highest number of significant PCs across participants (Supplementary Fig. 18). Interpretation involved three steps:

- (i) Identification of correlated utterances: For each PC, the top 20 positively and negatively correlated utterances were identified for each participant (Figs. 5c and g).
- (ii) Interpretation using ChatGPT: Utterances and correlation coefficients were input into ChatGPT (GPT-4o) for consistent interpretations across PCs, modalities, and participants.
- (iii) Synthesis of common components: ChatGPT synthesized interpretations to identify common components across participants. These methods are now included in the Methods section (lines 703–709, page 28).

1-6-2. Why “semantic” PCs?:

We acknowledge that “semantic” was not an accurate term. We have revised it to “linguistic,” which encompasses both semantic and syntactic aspects. This change is supported by the fact that our analysis removed variance in BOLD responses primarily linked to low-level motor and auditory processes (Supplementary Results). Additionally, prior research suggests that both semantic and syntactic information is represented in overlapping brain regions (e.g., Caucheteux et al., 2021, PMLR, “Disentangling syntax and semantics in the brain with deep networks”), and that the alignment between brain representations and language models is largely driven by syntactic information (Oota et al., 2023, arXiv, “Joint processing of linguistic properties in brains and language models”).

Comment 1-7: While an interesting and novel analyses, the conclusion is not clear. What does it mean for the perception and production phases to have different PC interpretations? How does this relate to purported shared/different computational mechanisms in each voxel for the two conditions?

In the previous manuscript, group-level PCA revealed that turn-taking, primarily driven by lower-level sensory-motor information, emerged as PC1 for both production and comprehension. Given the opposing nature of speech production and comprehension behaviors, we interpreted this as evidence that the two processes are represented by opposing PCs.

However, in the revised manuscript, we excluded lower-level motor and auditory information in advance, and focused on higher-level linguistic features (as noted in our response to Comment 1-4). Additionally, we performed PCA separately for each participant, enabling us to capture PCs that more accurately reflect each participant's unique conversational content.

This methodological shift led to different findings from the previous manuscript. Across all participants, the largest number of significant PCs were identified in the layer-0 and 1-second context conditions for both production and comprehension (Fig. 5b, f). While we did not observe perfect alignment between production and comprehension PCs—likely because the content produced and perceived was different—the two processes were strongly influenced by shared elements such as backchannel responses and fillers (Fig. 5d and h).

We discuss these new PCA results in Discussion:

“Notably, our identified principal components reflected social interaction nuances, such as backchannels, confirmations, and fillers. These elements require minimal cognitive effort yet are vital for maintaining conversational flow (Knudsen et al., 2020; Clark and Brennan, 1991; Clark and Fox Tree, 2002; Fox Tree, 2001). In contrast, principal components linked to “*specific information provision*,” such as referring to locations or objects, were identified as opposite axis of the social components.” (lines 408–412, page 19)

Comment 1-8: What types of semantic concepts did the conversations cover? I am interested in understanding why the traditional semantic PCs uncovered during pure perception (for ex., “social” vs. “visual” vs. “tactile”) don’t come up here even in the later components- is this an artifact of not covering semantically-rich content, or, is it fundamental to real-world communication that earlier studies have missed? This also relates to a potentially critical claim the paper makes re “our results suggest that the foundational components of cortical semantic representations are formed based on universal systematic communication rules, regardless of the specific semantic content.” What does this result say about the purported function of voxels ascribed a specific semantic selectivity in prior work and the conversation-related selectivity found here?

Answer: We address these comments in the following points.

1-8-1. Semantic content in our experiment:
As described in Methods section:

“These topics were selected to cover a wide range of semantic domains relevant to daily life, such as knowledge, memory, imagination, and temporal and spatial cognition, referencing the Corpus of Everyday Japanese (CEJC).” (lines 589–560, page 22)

To enhance transparency, we have added a topic list in Supplementary Table 5. Here are some examples:

- spatial locations (e.g., directions to the experiment location)
- numerical expressions (e.g., arithmetic thinking and teaching methods)

- objects (e.g., packing an eco-bag)
- emotional expressions (e.g., regrets, dislikes, moments of happiness, moments of relaxation)
- cooperative scenarios (e.g., consensus game discussing how to escape from the jungle).

1-8-2. Real-world interactive language:

We do not believe that the differences between our findings and prior work are due to a lack of semantically rich content. Rather, we propose that these differences reflect essential features of real-world interactive communication, which may not have been the focus of earlier studies on passive language use.

In our revised manuscript, we consistently observed PCs related to emotional expression and the provision of specific information across participants. Notably, these distinctions are similar to those identified by Huth et al. (2016), such as:

- “humans and social interaction” versus “perceptual descriptions, quantitative descriptions, and setting.”

As shown in Fig. 5d and h, we identified components such as:

- “Empathy/emotional expression vs. Specific information provision,”
- “Maintaining fluency/structure vs. Specific information provision,”
- “Concise responses/confirmations vs. Detailed explanations/elaborations.”

These components reflect key features of interactive communication, such as backchanneling, fillers, and responses, which are crucial in maintaining dynamic conversation. These elements are less emphasized in non-interactive contexts like monologues, where turn-taking and speech planning are not as critical (Knudsen et al., 2020, *Frontiers in Psychology*). The absence of these interactive features in non-interactive language tasks may explain why these distinctions have not been prominent in previous studies.

However, we acknowledge that our experiment does not cover the full range of possible semantic concepts. We hope our study will contribute to future efforts aimed at describing neural linguistic representations in natural, interactive language contexts.

1-8-3. Consistency with prior findings:

We interpret these results (as summarized in 1-8-2) as an extension of previous findings into interactive social contexts. These new interpretations have been discussed in detail in the revised Discussion section:

“Our results potentially extend the seminal work of Huth and colleagues (Huth et al., 2016), which comprehensively mapped fine-grained semantic representations during natural speech listening using word embeddings. Specifically, that study identified the first principal component differentiating between “*humans and social interaction*” and “*perceptual descriptions, quantitative descriptions, and setting*,” thereby separating social content from physical content. Our conversational data offered a unique opportunity to examine the semantic space surrounding social words in greater depth.” (lines 402–408, pages 18–19)

“In contrast, principal components linked to “*specific information provision*,” such as referring to locations or objects, were identified as opposite axis of the social components. This suggests that interactive language enhances the neural representation of social content, highlighting the interplay between semantic representations and social cognition.” (lines 411–414, page 19)

Comment 1-9: Throughout this study, the visual cortex is significantly predicted by the encoding models and is even attributed specific colors in the RGB PC space. Could this be because of eye motion during the task that need to be regressed out?

Answer: We recognize that eye movements could have influenced the prediction accuracy in the visual cortex, as participants' eye movements were not specifically controlled. To assess the impact of low-level processes, we used random embeddings to predict brain activity during both speech production and comprehension. Significant predictions were indeed observed in the visual cortex using these random embeddings (Supplementary Fig. 4). This suggests that random embeddings help exclude fluctuations not directly related to higher-order linguistic processing. However, we have not explicitly investigated the extent to which eye movements may have contributed to these results.

Comment 1-10: Which areas correspond to the “language network” being referred to in lines 369-378? I believe it is not Ev Fedorenko’s definition but perhaps “all regions activated during language production and comprehension”?

Answer: We acknowledge that our use of the term “language network” was potentially misleading, because we did not rely on the ROI definition from Ev Fedorenko’s language network localizer task (Fedorenko et al., 2011, PNAS; <https://www.pnas.org/doi/abs/10.1073/pnas.1112937108>). Therefore, we have removed the term “language network” from our revised manuscript to avoid confusion. Moreover, we have referred this point in Discussion:

“We did not conduct functional localizer tasks to delineate specific functional networks, such as the language network and ToM network. Thus, our analysis could not precisely attribute voxel clusters to specific functional networks.” (lines 416–418, page 19)

Comment 1-11: Perhaps reducing the curvature contrast will improve the visibility of RGB colors in the flatmaps.

Answer: We have reduced the curvature contrast in cortical surface maps to improve the visibility.

Comment 1-12: It might be useful to add one sentence about the FIR model while explaining the encoding model in the main text.

Answer: We have added the following sentence to clarify the use of the FIR model in the Results section:

“A finite impulse response (FIR) model was employed to predict BOLD responses with delays ranging from 2 to 7 s (5,632 features x 6 delays = 33,792 features).” (lines 101–102, page 4)

Reviewer #2

Comment 2-1: Given that head motion can have a significant effect on acquired BOLD signal, it is important see more detailed analysis of potential motion artifacts and how they may be affecting the results. It would be a good start to examine the motion traces per individual in the cortical voxels. The motion parameters have been added to the analysis as nuisance regressors. However, I am very surprised to see in Extended Data Fig. 3 that motion (and other features but see my next comment for the other features) have no unique variance explained that seems to be very unlikely, especially for the production data. Did the authors plot head motion model separately in a ridge model? Did the authors have systematically and separately investigate the effect of speaking in the scanner?

Answer: We agree that head motion can systematically affect fMRI measurements, and it is essential to address these potential artifacts to ensure the accuracy of our results. Although we have not conducted a separate experiment specifically focusing on the effects of speaking inside the scanner, we have analyzed how head motion correlates with speech production and its potential impact on brain activity.

2-1-1. Head motion data:

We evaluated head motion by calculating frame-wise displacement (FD; Power et al., 2012) and compared the magnitude of head motion between fMRI volumes with and without speech production. Our analysis revealed no significant difference in mean FD values between speech production and non-speech conditions (Mann-Whitney U test, $P = 0.88$; Supplementary Fig. 2). This suggests that engaging in speech production during conversation does not necessarily result in increased head motion. We have added this information under “Removing head motion artifact” in Supplementary Results:

“We assessed head motion using framewise displacement (FD; Power et al., 2012) and compared FD across scans conducted with and without speech production (Supplementary Fig. 2). At the group level, no significant difference was observed between the two conditions (Mann-Whitney U test, $U = 30$, $P = 0.878$). During speech production, the mean FD was 0.239 ± 0.051 mm (range: 0.182–0.322 mm), compared to 0.244 ± 0.052 mm (range: 0.202–0.363 mm) in the absence of speech. Notably, the variance in head motion among individual participants was greater without speech production, indicating more frequent excessive motion in this condition (standard deviation of FD: production, 0.093–0.185 mm; non-production, 0.287–0.613 mm).”

2-1-2. Head motion model:

We developed a voxelwise encoding model to account for head motion, incorporating six motion parameters and the FD value. The model’s prediction performance was generally low across the cortex, indicating that head motion does not substantially explain the observed brain activity. Detailed results are presented in Supplementary Fig. 3 and “Removing head motion artifacts” in Supplementary Results:

“To minimize motion-related artifacts in BOLD responses, we removed head motion effects by modeling six motion parameters and FD. Prediction accuracy, evaluated by the Pearson’s correlation coefficient between predicted and observed values, was low across the cortex (mean prediction accuracy in Fisher z value = 0.0500 ± 0.0093 ; range: 0.0351–0.0633; see Supplementary Fig. 3 for individual participant data). The predicted timeseries were subtracted from the preprocessed fMRI data to isolate the neural signals of interest.”

2-1-3. Separate investigation of speaking effects in the scanner:

Although we did not conduct a separate systematic investigation specifically aimed at isolating the effects of speaking inside the scanner, our analyses suggest that head motion during speech production did not introduce significant artifacts (as described in 2-1-2). We acknowledge the importance of this issue and will consider it in future research.

Comment 2-2: I have a hard time following what the authors did in their variance partitioning. It seems that the authors only computed the difference between the full model and a reduced model. Has this been done for all the different feature spaces separately? The methods section lacks the appropriate detail here and needs improvement. Please see the cited papers Le Bel et al. 2021 and de Heer et al. 2017 for a better description. Based on this I am also curious whether the authors fit each model as a separate ridge model? Given de Heer et al. 2017 it is very surprising that the low-level features are very poorly predicting in speech-related regions (Extended Data Fig. 3). Given that there may be voxels that share representations of GPT features and e.g. part-of-speech features it would be more beneficial to use a banded ridge model [la Tour et al. 2022; Nunez-Elizalde et al. 2019] and examine the split-correlations of the prediction accuracies across different models.

Answer: We acknowledge that our Methods section needs further clarification, and we recognize the potential benefits of using a banded ridge model.

2-2-1. Variance partitioning for each feature space:

We performed unique variance partitioning by computing the difference between the full model and a reduced model. This analysis was conducted separately for each feature space to isolate the unique contributions of each set of features.

2-2-2. Clarification on methods:

In our revised manuscript, we have simplified the main model by excluding lower-level features (cochleagram, part-of-speech, morpheme count, and syllable count). The updated main model (the Separate Linguistic model) now focuses on higher-level linguistic features derived from GPT contextual embeddings (Fig. 1c). To quantify the unique contributions of these features, we computed the difference between the full model (including both production and comprehension embeddings) and reduced models (with embeddings for production or comprehension alone). Additionally, the shared variance between production and comprehension was measured by analyzing the intersection of their contributions. The Methods section has been updated accordingly, following de Heer et al. (2017):

“We performed variance partitioning to quantify the unique contributions of linguistic features to BOLD responses in production, comprehension, and their intersection. Following methods from previous voxelwise modeling studies (Lescroart et al., 2015; de Heer et al., 2017), we used three models: a production-only model, a comprehension-only model, and their combination (i.e., the Separate Linguistic model). We used set theory to calculate the unique and common variances explained as follows. Unique variance was calculated as follows:

$\text{Production} \setminus \text{Comprehension} = \text{Production} \cup \text{Comprehension} - \text{Comprehension}$

$\text{Comprehension} \setminus \text{Production} = \text{Production} \cup \text{Comprehension} - \text{Production}.$

Shared variance was calculated as follows:

$\text{Production} \cap \text{Comprehension} = \text{Production} + \text{Comprehension} - \text{Production} \cup \text{Comprehension}.$

To maintain consistency with correlation coefficients, square roots of variance components were reported. Variance partitioning was applied to all layer-context combinations. For TRs containing single (either production or comprehension) or overlapping conditions (production and comprehension), we applied variance partitioning consistently.” (lines 675–688, page 27)

2-2-3. Separate ridge models:

While we excluded some lower-level features, we retained head motion features and fitted a separate ridge model to evaluate their effects. This analysis is presented in Supplementary Fig. 3 (as noted in response 2-1-2). The revised results show that head motion has a broader effect across the brain than previously estimated.

2-2-4. Banded ridge regression:

We agree that using a banded ridge regression model is more appropriate for combining multiple feature spaces in a joint model. In our revised manuscript, we employed banded ridge regression to better account for shared representations across different features. This is now specified in Results and Methods sections:

“This joint model was fit to BOLD responses using banded ridge regression (Nunez-Elizalde et al., 2019; Dupré la Tour et al., 2022) for each voxel.” (lines 102–103, page 4)

“weights estimated using banded ridge regression, implemented via *Himalaya* package (Nunez-Elizalde et al., 2019; Dupré la Tour et al., 2022).” (line 650, page 26)

Comment 2-3: Related to my comment in (2) did the authors examine the stimulus-correlations? It seems that the GPT features are strongly correlated with the low-level features and that the model with a lot of dimensions in the ridge-model explains most of the variance. Have the authors examined the implications of feature dimensions on model weights further? Given the uncertainties in the main model, I also have difficulties following the interpretations made in Fig 5 about semantic representations.

Answer: We address these comments in the following points.

2-3-1. Correlated low-level features:

In conversations, the alternating interaction between production and comprehension can create similar temporal patterns within each modality and inverse correlations between them. While we did not directly examine stimulus correlations across all feature sets, we addressed this concern by systematically removing the influence of low-level sensorimotor features before predicting BOLD responses using higher-level linguistic features, as detailed below. Our analysis followed a stepwise regression approach, progressively removed variance associated with low-level features to ensure that the variance explained by GPT features reflects higher-level linguistic processing.

- First stage: We subtracted the variance in the BOLD response attributed to head motion (Supplementary Fig. 3).
- Second stage: We further subtracted the variance explained by random embeddings from the residual BOLD response. These random embeddings were fitted using a joint model that included both production and comprehension, maintaining the same dimensionality (2,816 dimensions) as the GPT embeddings. This model achieved high prediction accuracy (max $r \sim 0.6$) in cortical regions, such as the primary motor cortex involved in vocalization and the auditory cortex involved in speech perception (Supplementary Fig. 4).
- Third stage: We applied linguistic models with GPT features to the residual BOLD responses, after accounting for the variance from the second stage. The prediction accuracy of the Separate Linguistic model was lower than the main model from the previous manuscript (max $r \sim 0.3$ compared to max $r \sim 0.6$). Notably, the high prediction accuracy previously observed in the motor and auditory cortices disappeared, indicating that the variance associated with these areas had been removed in the second stage.

This multi-stage approach highlights that the GPT features capture distinct, higher-level linguistic variance, independent of low-level sensorimotor features. The lower prediction accuracy after removing low-level variance further supports this conclusion.

2-3-2. Feature dimensions:

We did not conduct specific experiments to examine the direct effect of feature dimensionality on model weights. Instead, we investigated the prediction accuracy using random embeddings of the same dimensionality (2,816 dimensions). The random model achieved a relatively high prediction accuracy (max $r \sim 0.6$). To address concerns about inflated prediction performance due to the large number of features, we subtracted the predicted BOLD response from this random model and used GPT features to predict the

residual BOLD response. This approach ensures that our estimation of the GPT model's prediction performance is not biased by the number of features.

Comment 2-4: The main results that the authors show is based on the reduced semantics model, however, I am not convinced that the low-level features explain the necessary low-level information (see my previous comments in (1-3) that are correlated in the results and therefore the conclusions hold.

Answer: We acknowledge that our previous manuscript underestimated the influence of low-level information. To address this, the revised manuscript systematically accounted for variance associated with head motion, motor, and auditory information, as mentioned in the previous comment (2-3-1). After subtracting these low-level effects, we assessed the prediction accuracy of GPT features on the residual brain activity data. This approach allowed us to more accurately isolate the contribution of high-level linguistic features, ensuring that our conclusions more robustly reflect the relationship between brain activity and linguistic representation.

Comment 2-5: The authors used as semantic features the features extracted from one specific GPT-layer (layer 27) with the reasoning that other studies have shown that “semantic features extracted from middle layers are more accurate in predicting brain activity”. Although I agree with the authors that this has been to some extent replicated using different large language models in different studies, layer-wise representational effects may remain in this unique dataset. In addition, the studies mentioned in the manuscript use smaller GPT-models or other contextual language models such as BERT and ELMo (LeBel et al. 2021, Caucheteux et al. 2023 use 12-layer GPT model, and Toneva and Wehbe, 2019 use other contextual language models such as BERT and ELMo). It would therefore still be important to run a systematic layer-by-layer analysis of the results. One possible way to present the results would be to show per layer the average prediction accuracy across significantly predicted voxels (per subject), and the average across subjects and this separately for comprehension and production. Only then can the importance of layer 27 vs. other layers or all-layers predictions can be systematically assessed, which is important for understanding how different layers relate to the semantic representation in both conversation models (comprehension vs. production) in this dataset.

Answer: We have addressed your comments and incorporated the suggested revisions throughout our revised manuscript.

2-5-1. Layer-wise representational effects and the influence of context lengths:

We agree with your suggestion to perform a systematic layer-by-layer analysis, which has been included in the revised manuscript. Additionally, we explored how different context lengths affect contextual embeddings extracted from different GPT layers (Fig. 1b). Since each transformer layer in GPT captures distinct syntactic and semantic information, we hypothesized that very short speech fragments (e.g., within a one-second window) may not fully capture the layer-wise differences. Due to the high computational costs, we analyzed every third layer of the 36-layer GPT model. This is presented in Results section:

“This GPT model comprises an input embedding layer and 36 Transformer layers, each containing 2,816 hidden units. We extracted embeddings from 13 hierarchical layers (the input layer and every third Transformer layer) across six context lengths (1, 2, 4, 8, 16, and 32 s).” (lines 94–97, page 4)

Our results indicate that both layer position and context length significantly impact mean prediction accuracy across the cortex. This is presented in Results section:

“To investigate whether the average prediction accuracy across the cortex was influenced by layer position and context length, we conducted a linear mixed-effects (LME) model analysis. Participants were specified as random effects, allowing for variations in the effects of context length and its squared term across participants (see Methods). We found an inverted U-shaped

relationship across timescales (context length: $b = 0.78$, $SE = 0.17$, $P = 0.0027$; context length²: $b = -0.46$, $SE = 0.094$, $P = 0.0017$, summarized in Supplementary Table 1) and across layers (layer position: $b = 0.13$, $SE = 0.013$, $P < 2e-16$; layer position²: $b = -0.27$, $SE = 0.015$, $P < 2e-16$). Additionally, we found a significant interaction effect between context length and layer position ($b = -0.080$, $SE = 0.013$, $P = 1.4e-9$).” (lines 114–122, page 5)

We have also included plots illustrating the average prediction accuracy across voxels for each layer, context length, and participant (Supplementary Fig. 8).

2-5-2. Layer-wise variance explained by production and comprehension:

We separately analyzed these effects for production and comprehension, as described in Results section:

“We next explored modality-specific linguistic representations by fitting Production-only and Comprehension-only Linguistic models. These models utilized modality-specific contextual embeddings to quantify the variance in BOLD responses that could be uniquely attributed to each modality. Variance partitioning (Lescroart et al., 2015; de Heer et al., 2017) was employed to assign variance to either production or comprehension using the following equations:

Production \ Comprehension = Production \cup Comprehension - Comprehension

Comprehension \ Production = Production \cup Comprehension - Production.

We found that production explained more variance at shorter timescales (1 to 4 s; Fig. 3a; see Supplementary Fig. 9 for individual layers and participants). The LME analysis revealed significant effects of context length ($b = -0.38$, $SE = 0.092$, $P = 0.0044$), layer position (layer position²: $b = -0.11$, $SE = 0.015$, $P = 5.8e-13$), and their interaction ($b = -0.055$, $SE = 0.014$, $P = 5.5e-5$). In contrast, 8 comprehension explained more variance at longer timescales (16 to 32 s). The LME analysis indicated an inverted U-shaped relationship for context length (context length: $b = 1.04$, $SE = 0.25$, $P = 0.0040$; context length²: $b = -0.45$, $SE = 0.18$, $P = 0.043$) and layer position (layer position: $b = 0.067$, $SE = 0.013$, $P < 4.5e-7$; layer position²: $b = -0.12$, $SE = 0.015$, $P = 6.3e-16$). Across participants, the context length that maximized prediction accuracy consistently varied between modalities, with production peaking at shorter timescales and comprehension at longer timescales (Fig. 3b). These findings suggest distinct timescale selectivity for production and comprehension.” (lines 176–193, pages 8–9)

This new analysis reveals relationships between layers and context lengths that were not apparent in our previous manuscript, which had focused solely on layer 27 and a one-second context window. The results highlight the substantial impact of timescales on prediction accuracy and the unique variance differently explained by each modality.

Comment 2-6: Following on my previous point in (5) it is advisable to create a correlation matrix of significantly predicted voxels between comprehension and production per-layer to examine the per-layer similarity between the modes. Especially, because given the complex and not-immediately interpretable representations of large language models, I am not sure why we should assume that a random middle layer 27 performs similarly for both comprehension and production. Also, have the author compared their contextual model to other contextual models?

Answer: We address the comments in the following points.

2-6-1. Per-layer similarity between production and comprehension:

We agree that it is essential to investigate the similarities between production and comprehension for each layer. We have created a correlation matrix that shows the similarity of significantly predicted voxels between production and comprehension for each layer. Specifically, we calculated the correlation coefficients of explained variance across various layer and context length combinations. The mean correlation coefficients across participants are displayed in Response-Fig. 1 for each context length. The results

Response-Fig. 1 | Mean correlation coefficients of variance explained between production and comprehension. For each participant, layer, and context length, we calculated the correlation coefficients of the variance explained between production and comprehension and averaged these across participants. As the context length increased, the strength of the correlation decreased. Within each context length, correlations between the same layers were the highest. Additionally, at longer timescales (16 and 32 s), relatively high correlation was observed between high production layers and low comprehension layers.

consistently indicate that the highest correlations are observed between the same layers across all timescales. Additionally, for longer timescales (16 and 32 s), we observed higher correlations between early layers (3 to 9) in comprehension and later layers (27 to 33) in production.

2-6-2. Comparison with a base GPT model without instruction tuning:

We also replicated our analysis using a base GPT model that was not instruction-tuned, to examine the potential impact of instruction tuning on our results. Instruction-tuning can modify how GPT models represent world knowledge and has been suggested to improve the prediction accuracy of brain activity (Aw et al., 2023, arXiv). To determine whether instruction tuning enhanced prediction accuracy, we performed the same analysis with a pre-trained GPT model without instruction tuning. Consequently, our linear mixed-effects analysis showed no significant differences in prediction accuracy between the instruction tuned and base models, suggesting minimal impact of instruction tuning. Additionally, we also evaluated the effect of instruction tuning on weight correlations and found no significant influence. These findings are included in the Results section:

“To ensure that these findings were not influenced by the instruction-tuned GPT model (Aw et al., 2023), we replicated the core analyses using a base GPT model before instruction tuning. The results were consistent, confirming that instruction tuning did not affect the observed effects (Supplementary Fig. 17).” (lines 270–272, page 13)

Comment 2-7: It reads like (L655-657) that the authors performed a parametric statistical test. However, this test may inflate the results and may be insufficient. A non-parametric permutation test while accounting for autocorrelation of the BOLD response is more appropriate.

Answer: We have updated our analysis to include a non-parametric permutation test that accounts for the autocorrelation inherent in the BOLD response. Specifically, we segmented each run (430 seconds) into 21 blocks of 20 seconds, preserving the temporal structure within each block. These blocks were randomly shuffled to generate permuted samples of the BOLD responses, and this shuffling was repeated 1,000 times to create a null distribution of correlation coefficients between the shuffled samples and the predicted values. We then evaluated the statistical significance of our original predictions by comparing them with this null distribution. The detailed explanation is added to Results and Methods sections:

“To account for autocorrelation, data were divided into 20-second blocks, permuted to estimate the null distribution, and correlations were calculated across 1,000 permutations to obtain P-values.” (lines 105–107, page 5)

“Statistical significance was determined through a one-sided permutation test. A null distribution was generated by permuting 20-TR blocks (20 s) of the left-out test data 1,000 times, recalculating the correlation for each permutation.” (lines 655–657, page 26)

Comment 2-8: Have the authors looked in more detail into speech errors? In day-to-day communications it is very common that we make speech errors, so did the authors investigate the distributions of participant’s speech errors and fit a speech error model?

Answer: Indeed, we observed speech errors made by both the participants and the experimenter during the dialogue experiments. While these errors might have impacted prediction accuracy, we transcribed the spoken utterances exactly as they occurred, without conducting an in-depth analysis of the distribution of speech errors or fitting a speech error model. We acknowledge that a closer examination of speech errors could provide further insights into the mechanisms of everyday communication. Investigating their impact and incorporating a speech error model would be a promising direction for future research.

Comment 2-9: Extended Data Fig. 6 & 7 show the opposite results for P3 and P4 do the authors have an explanation for this behavior?

Answer: We understand your concern regarding the opposite results for P3 and P4 in our previous manuscript. Upon further analysis, we discovered that these results were partially driven by inflated negative weight correlations due to the contrasting neural activity between vocalization and auditory perception. In the revised manuscript, we addressed this issue by removing lower-level motor and auditory factors from the analysis (as noted in 2-3-1), which subsequently reduced these negative weight correlations. As a result, the “opposite results for P3 and P4” are no longer observed in the updated analysis (Supplementary Figures 13 and 16).

Comment 2-10: The Semantic model construction section in Methods needs necessary details on how fine-tuning was performed. Also, the wording in L597-598 mentioned that the authors first pre-train a model wasn’t the huggingface model used?

Answer: To clarify, in both the previous and revised versions of the manuscript, we utilized a pre-trained Huggingface model that was fine-tuned by a third party, rinna (<https://huggingface.co/rinna/japanese-gpt-neox-3.6b-instruction-sft>). We did not perform any additional modifications or fine-tuning ourselves. Unfortunately, detailed information regarding rinna's fine-tuning process is not publicly available. The only available detail, which we have included in our manuscript, pertains to the dataset used for the fine-tuning:

“To extract contextual embeddings from the content of conversations, we utilized an instruction-tuned language model (GPT) fine-tuned specifically for Japanese (Sawada et al., 2024; <https://huggingface.co/rinna/japanese-gpt-neox-3.6b-instruction-sft>). This model is built on the open-

source GPT-NeoX architecture (Black et al., 2023) and was pre-trained to predict the next word based on preceding context using 312.5 billion tokens from various Japanese text datasets: Japanese CC-100, Japanese C4, and Japanese Wikipedia. For comparative purposes, we also replicated our analysis using the non-instruction tuned version of the model (<https://huggingface.co/rinna/japanese-gpt-neox-3.6b>) as detailed in Supplementary Fig. 17. Instruction-tuning was performed using datasets translated into Japanese, including Anthropic HH RLHF data, FLAN Instruction Tuning data, and the Stanford Human Preferences Dataset. The resulting model architecture comprises 36 transformer layers with hidden unit dimensions of 2,816.” (lines 603–612, page 24)

Reviewer #3:

Comment 3-1: In the Methods section, the authors report that voxel responses are modeled using an FIR model, and that the slow hemodynamic response is accounted for through concatenating features from the previous 2-7s to account for the delay (TR=1s). This means that instead of predicting a voxel's response from the 2,816-dimensional vector representation of GPT-NeoX (or even the concatenation of production & comprehension representations: $2 * 2,816 = 5,632$ features), the voxel response is instead predicted using $5,632 * 6 = 34,962$ features (concatenating the production & comprehension features within the 6s in the delay window, from seconds 2-7 before the current TR). That is a lot of features!

--Of course, using a lagged feature representation for FIR modeling is well-established. However, what is the justification of using concatenation over averaging or summing the feature representations within a window (e.g., Wehbe et al., 2014; Caucheteux, Gramfort & King, 2021), or down-sampling them (e.g., Jain & Huth, 2018)? And how do either of the other methods fare?

Answer: We address your questions in the following points.

3-1-1. Alignment of language stimuli with fMRI data:

In previous studies (e.g., Wehbe et al., 2014; Jain & Huth, 2018), additional numerical manipulations were needed to align word embeddings with fMRI data due to differences in sampling rates. For example, Wehbe et al. (2014) presented words visually at a fixed rate (every 500 ms), with four words appearing within each fMRI acquisition (TR = 2,000 ms). To match the word embeddings with the fMRI volumes, they summed the embeddings for these four words. Similarly, Jain & Huth (2018) worked with naturally spoken stimuli, transcribing each word and converting it into embeddings, which were then down-sampled to align with the fMRI data's sampling rate (TR = 2,000 ms). In contrast, our study extracts GPT embeddings for utterances (words or sentences) occurring within each fMRI TR (1,000 ms) or a longer context window, eliminating the need for additional manipulations such as summing or down-sampling the embeddings.

3-1-2. Finite impulse response (FIR) function to account for the delay in the hemodynamic response:

Our approach of concatenating features across multiple time points is consistent with the FIR modeling methods used in all the studies mentioned (Wehbe et al., 2014; Caucheteux, Gramfort & King, 2021; Jain & Huth, 2018). This technique, which allows us to account for the delay in the hemodynamic response and thus give more accurate predicted response time courses, is standard across these studies.

Comment 3-2: --For the PCA analysis, the authors indeed change the representation and average the features within a given time lag ("to eliminate temporal information"). Why is this approach reasonable here but not elsewhere?

Answer: As explained in our previous response (3-1-2), the FIR function is a standard method commonly used across studies to account for the systematic delay in fMRI measurements. In our analysis, we used PCA to project the linguistic information represented by the estimated weights onto a lower-dimensional space, following prior studies (e.g., Huth et al., 2012; Huth et al., 2016). By applying PCA to the weights for each time delay from 2 to 7 s ($w_{-2}, w_{-3}, \dots, w_{-7}$), we could better understand the linguistic representation at each delay. However, since our primary interest is not in the specific differences between time delays, averaging the weights across delays allows us to focus on the overall linguistic representation. This approach enhances interpretability by eliminating temporal specifics while still capturing aggregated linguistic features.

Comment 3-3: --Further, given the large number of features, how does a control model, such as an untrained model of the same architecture, or a random embedding with the same number of features fare? Is the sheer number of parameters maybe already enough to achieve high predictivity? These controls are critical to be able to interpret the reported results.

Answer: We have incorporated this suggestion into our revised manuscript and provided additional analysis in the Supplementary Results (“Removing low-level sensorimotor signals”).

To address the concern about the large number of features, we have examined whether a random embedding model with the same number of features could predict brain activity (as noted in 2-3-1). For each utterance stimulus per TR, we generated 2,816 random numbers (ranging from 0 to 1; Zada et al., 2024). After controlling for head motion effects, we fitted this random embedding model and observed that it achieved high prediction accuracy, particularly in the primary motor and auditory cortices (max $r \sim 0.6$; Supplementary Fig. 4). This prediction accuracy is comparable to what we observed in our previous results (see Fig. 1 in the previous manuscript). These findings indicate that lower-level processes, such as vocalization and speech listening, significantly impact the BOLD response in dialogue-based fMRI experiments. Based on these results, we subsequently fitted higher-level linguistic models after subtracting the predicted BOLD responses from the random embedding model to isolate the specific contributions of linguistic features.

Comment 3-4: Likewise, the variance partitioning analysis, aimed at isolating the variance explained by different “low-level linguistic features”, leaves several questions unanswered.

--First, these features are represented as just one, or very few features in the model (e.g., number of morphemes (an integer), or part of speech (a string)). Given the large number of what the authors call “semantic features” (i.e., GPT features) it seems unsurprising that removing these few features, while keeping the full set of GPT features (~35,000!), would lead to a negligible drop in predictivity.

Answer: We acknowledge that our previous joint model, which combined eleven different features and used a simple ridge regression for estimation, likely underestimated the impact of low-level features. To address this, we adopted a stepwise approach (see our response 2-3-1 to Comment 2-3), fitting voxelwise models in three steps. This allowed us to separately estimate the variance explained by lower-level sensorimotor features and that explained by higher-level linguistic features (i.e., GPT embeddings).

Comment 3-5: --This is especially true considering that many of these features are represented (in a distributed way) in the GPT features, too. Thus, arguing that such low-level linguistic features do not contribute to the LLM-to-brain alignment is unwarranted, based on these results. In fact, other approaches have arguably tried to test similar claims more directly, by eliminating/marginalizing over information related to specific linguistic properties in the language model representations (see, e.g., Caucheteux et al., 2021 (<https://proceedings.mlr.press/v139/caucheteux21a/caucheteux21a.pdf>); Oota et al., 2022 (<https://arxiv.org/pdf/2212.08094.pdf>)).

Answer: These references highlight that many features within GPT embeddings indeed correlate with specific syntactic and semantic dimensions. We have now cited these studies in the Introduction of our revised manuscript to better contextualize our findings:

“Throughout the paper, the term linguistic encompasses both semantic and syntactic dimensions, reflecting their tight interdependence (Oota et al., 2023; Caucheteux et al., 2021).” (lines 65–67, page 3)

Comment 3-6: The manuscript underspecifies how the input was provided to the GPT model, how representations were obtained, and whether, empirically (not only conceptually), the use of fine-tuning the model as a conversational agent makes a difference. Specifically:

--The model used was trained using a specific input format (<https://huggingface.co/rinna/japanese-gpt-neox-3.6b-instruction-sft#io-format>). Were inputs prepared in the same way? I.e., indicating different speakers? It looks like the production/comprehension streams were just separated directly and fed individually to the model; see, e.g., Fig. 1). If the nested production/comprehension models did not have access to the interlocutor’s content, and thus, the larger conversational context, how much do we benefit

from using the instruction-tuned model? Using this kind of model seems conceptually reasonable, but the choice should be justified better, including through prior literature (e.g., Aw et al, 2023, see below for full reference), and possibly through a comparison to a base GPT model without instruction tuning.

Answer: We address these questions in the following points.

3-6-1. Specific input format:

We did not use the specific input format designed for fine-tuning GPT for interactive purposes, which distinguishes between system and user. For our analysis, which focused on extracting contextual embeddings, this format was not necessary.

3-6-2. Broader conversational context:

We have revised our manuscript to include a Unified Linguistic model, where both production and comprehension inputs are combined in the GPT model to extract contextual embeddings (Fig. 1e). This unified approach allows the model to potentially capture a more comprehensive conversational context compared to treating production and comprehension streams separately (as in the Separate Linguistic model). However, our results showed no significant difference in prediction accuracy between the Unified and Separate Linguistic models. The results from the Unified Linguistic model are now included in Fig. 1e, 2b, and 3c, as well as in Supplementary Figs. 7, 8, and 14. The main results of the model is specified in Results section:

“Although these results demonstrated generalizable linguistic representations across modalities, another critical question arises: is a unified linguistic representation sufficient for accurate predictions? To address this, we developed a Unified Linguistic model that extracted GPT embeddings from combined transcripts (Fig. 1e). No significant differences in prediction performance were observed between the Unified and Separate Linguistic models (Fig. 2b, Supplementary Figs. 7 and 8 for individual layers and participants). The LME analysis of the Unified Linguistic model revealed significant effects of layer position ($b = 0.09$, $SE = 0.022$, $P = 3.9e-5$) and its interaction with context length ($b = -0.10$, $SE = 0.022$, $P = 6.6e-6$), but no significant main effect of context length ($b = 0.31$, $SE = 0.16$, $P = 0.093$). These findings further support the existence of a shared linguistic representation across modalities.” (lines 150–158, page 7)

3-6-3. Instruction-tuning:

As mentioned by Aw et al. (2023), instruction-tuning can modify how GPT models represent world knowledge, which may improve the prediction accuracy of brain activity. To determine whether instruction-tuning enhances prediction accuracy, we performed the same analysis using a pre-trained GPT model without instruction-tuning. Consequently, our linear mixed-effects analysis showed no significant difference in prediction accuracy between the two models, suggesting that instruction-tuning had no meaningful impact. Additionally, we also examined whether instruction-tuning affected weight correlation and found no significant effect. Therefore, our findings suggest that instruction-tuning had minimal influence on prediction accuracy of brain activity. We added these results in the Results section:

“To ensure that these findings were not influenced by the instruction-tuned GPT model (Aw et al., 2023), we replicated the core analyses using a base GPT model before instruction tuning. The results were consistent, confirming that instruction tuning did not affect the observed effects (Supplementary Fig. 17).” (lines 270–272, page 13)

Comment 3-7: --It is unclear how model representations were obtained for the “semantic chunks”? Is the representation of the last token obtained (as is common for GPT models?), or is the representation of tokens summed, averaged, concatenated? This information should be added to the methods.

Answer: We acknowledge that our previous manuscript lacked sufficient detail on the methodological approach to obtaining model representations. In both our previous and current analyses, we averaged the GPT embeddings across all tokens within each segment corresponding to the fMRI TR. We have now added this additional detail in the revised Results section:

“We averaged the embeddings across all tokens within each segment (or fMRI volume, TR = 1,000 ms), resulting in 78 feature combinations per modality (13 layers × 6 context lengths).” (lines 97–99, page 4)

Comment 3-8: Further, while the previous literature indeed shows that middle layers typically perform best in ANN-to-brain comparisons, the choice of layer 27 seems arbitrary. Especially given the novel paradigm, an across-layers comparison should be performed to confirm this choice empirically.

Answer: This concern overlaps with the issue raised in Comment 2-5. To avoid redundancy, we kindly refer you to our detailed response to Comment 2-5.

Comment 3-9: Lastly, we were surprised to see the model’s very high (and, to the best of our knowledge, not normalized relative to a ceiling, which is typically computed by how well a given participant’s data can be predicted from other participants’ data) predictivity performance (~.6). This is substantially higher than most other papers report, which suggests that some overfitting is likely happening. What is the signal-to-noise ratio in the dataset? I.e., how much variance can be explained overall? The high predictivity is especially surprising to see this given the unscripted conversations and the leave-one-session out cross-validation. Presumably most content discussed in this left out session will not be shared (except for backchanneling, etc., more on this point below).

Answer: We agree that the high prediction accuracy reported in our previous manuscript could suggest potential overfitting. To address this, we have made several improvements in the revised manuscript.

3-9-1. Very high prediction performance:

This concern overlaps with the issue raised in Comment 3-3. To avoid redundancy, we kindly refer you to our detailed response to Comment 3-3.

3-9-2. Noise ceiling:

Due to the nature of our experimental paradigm—spontaneous conversations between participants and the experimenter—it is not feasible to collect identical data repeatedly or apply uniform stimuli across participants. Therefore, calculating the signal-to-noise ratio in the traditional sense, as done in voxelwise modeling studies, is not possible. As a result, we could not determine the prediction accuracy ceiling using the standard noise ceiling method.

Comment 3-10: Through correlating the weights of the semantic model for production and comprehension, the authors report evidence in favor of “modality-invariant semantic processing” in the amodal regions, including STG/STS. However, in the section before they argue that these regions are selective for comprehension and defend a separation between production and comprehension. How do these results fit together?

Answer: We agree that the term “modality-invariant semantic processing” may be misleading, as it suggests that semantic representations are identical across modalities. In our revised manuscript, we have replaced “modality-invariant” with “modality-aligned” to more accurately reflect that while there are similarities in representations across modalities, they are not identical. This term is used in Discussion:

“Modality-aligned representations were primarily localized to brain regions involved in processing word- and sentence-level linguistic information over shorter timescales” (lines 420–422, page 19)

To clarify, we classify voxels as “cross-modal” if the model weights show a degree of generalization across modalities. This generalization reflects similarity, not equality, of model weights. In fact, our revised manuscript shows relatively low correlations of weights in cross-modal voxels ($0.1 < \text{mean } r < 0.3$; Supplementary Fig. 10), indicating some similarity in linguistic tuning for production and comprehension, but far from identical tuning. Additionally, cross-modal voxels may show a stronger preference or selectivity for one modality over the other. In our classification, voxels are divided into production voxels (blue), comprehension voxels (red), and bimodal voxels (green), based on the variance explained by these variables (Fig. 4). For example, a cross-modal voxel showing 0.1 variance explained for production and 0.09 for comprehension, with 0.09 for the intersection of the two, would be categorized as a production voxel due to the slightly higher variance explained for production.

Comment 3-11: The authors report the intriguing finding that “semantic representations in a single voxel can differ between production and comprehension, even when using the same words or phrases”. This conclusion is drawn based on the drop in prediction accuracy for a mapping model with swapped regression weights between production vs. comprehension at test time. This analysis is clever and the results are intriguing, but

--what data support this result (even when using the same words or phrases)? Did the authors subset the data for overlap with training utterances and showed that for this dataset, too, prediction performance dropped?

Answer: We acknowledge that the statement “semantic representations in a single voxel can differ between production and comprehension, even when using the same words or phrases” was not directly tested by comparing prediction accuracy for exactly the same words or phrases between production and comprehension. Our intended meaning was that different regression weights, applied within the same embedding space, would lead to different predictions even for the identical words or phrases. To prevent confusion and potential misunderstanding, we have removed this claim from our revised manuscript.

Comment 3-12: --Later in the paper the authors show that different voxels seem to be responsible for representing production vs. comprehension semantic information. What do these results look like when subsetting for voxel preference? Would the expectation be that voxels which encode both “modalities” are better predicted by this model? And what are the implications if this is/is not so?

Answer: We apologize for any misunderstanding, but we are unclear about the specific intent of this question. However, in our new results, we conducted variance partitioning to separately examine voxels tuned to production-only, comprehension-only, and both modalities. We analyzed the average variance explained across voxels with respect to different layers and timescales. Our results show that voxels selective to one modality are more prevalent at shorter time scales for production, while longer time scales are more relevant for comprehension. We hope these findings address your question sufficiently.

Comment 3-13: The authors claim that they “aimed to identify the specific semantic content that is predominantly represented in each voxel.”. However, they use a group-level analysis. Given the anatomical differences of individuals’ brains it seems unlikely that the same voxels would represent the same information, leaving the aim unsupported.

Answer: First, we would like to clarify that our revised manuscript does not employ a group-level analysis as described in the comment. Instead, we perform principal component analysis (PCA) separately for each participant (i.e., individual PCA).

Anatomical variability among individual does not pose a problem here because we aimed to identify low-dimensional spaces in GPT features rather than in specific voxels. In our previous manuscript, we performed a group-PCA by aggregating voxel weights across all participants (2,816 GPT features x 80,000 voxels, with

10,000 voxels per participant across eight participants). The goal was to identify lower-dimensional spaces within the 2,816-dimensional embeddings, such as distinguishing between ‘social vs. physical language’. For this purpose, we conducted PCA with GPT features as variables and voxels as data samples. These GPT features are consistent across participants, while voxels were different across participants.

Conversely, if our goal had been to identify specific anatomical regions or networks that uniformly represent linguistic components across individuals, such as distinguishing between the superior temporal and medial prefrontal cortex, we would have performed PCA using voxels as variables (after transforming them into standard space) and GPT features as data samples.

Comment 3-14: The methodology and interpretation of the PC analysis are unclear. This is especially important given that the PC analysis (while only being one of many analyses presented in the paper) is the main focus of the discussion section, accounting for ~3/4 of that section.

Answer: In the revised manuscript, we have made several improvements to clarify both the PCA methodology and its interpretation. In the previous version, we conducted PCA only at the group level, without accounting for individual differences in conversational content or performing PCA at the individual level. In the revised version, we now conduct PCA separately for each participant, aligning the PCs with each individual’s conversational content to capture more personalized insights:

“Due to variability in conversational content, PCA was conducted separately for each participant” (lines 278–279, page 13)

Moreover, while the previous manuscript relied heavily on subjective interpretations of the PCs, we have now employed ChatGPT to assist in interpreting these components objectively, ensuring greater transparency and consistency in our analysis (noted in response to comment 3-17 below). Moreover, we have revised the Discussion section to provide a more balanced presentation of our findings, significantly reducing the overemphasis on PCA results.

Comment 3-15: The part-of-speech weights estimated in the voxel-wise encoding model (that were then discarded for the encoding analysis) were correlated with PC scores for the participants. Why were they included here when they were shown to not contribute to the alignment? This point is critical given that most of the conclusions are based on the analysis of the POS (and number of morphemes).

Answer: We agree the inconsistency in including part-of-speech (POS) data in the PCA interpretation, given that it was excluded from the encoding analysis due to its limited contribution to prediction accuracy. To address this issue and enhance clarity, we have removed the POS data from our analysis. Instead, we have utilized ChatGPT to interpret the PCA results based solely on the content of the conversation.

Comment 3-16: The PC analysis returns backchanneling as a primary factor. How much do the authors think this result is an artifact of the methodology? In particular, in leave-one-out cross-validation, backchanneling arguably provides some of the few utterances that would be shared across training and test sets, and they are usually quite frequent, too. If the authors want to make some claim about this factor being important, they need to rule out this trivial explanation.

Answer: We acknowledge that fillers and backchannels are frequently used, regardless of the conversation content, and may have significantly contributed to predicting brain activity across sessions. However, our PCA analysis evaluates whether the variance explained by the weight PCA is significantly greater than that explained by the stimulus (GPT embedding) PCA. Therefore, the presence of backchannels in the the PCs is not simply due to their frequent occurrence across sessions.

Comment 3-17: Further, some of the conclusions drawn from the correlation analyses are premature. For example: “both PC1 for production and comprehension displayed a negative correlation solely with the weights of interjections (Fig. 5 c, f). Simultaneously, they exhibited a strong positive correlation with the weights of other parts of speech. Moreover, these components showed little association with specific semantic content. This suggests that both PC1s represent turn-taking” > why is this necessarily the case? Many other explanations are possible.

Answer: We agree that correlations between PCs and lower-level linguistic features can be unstable, and drawing conclusions based solely on correlation values may be premature. In our revised manuscript, we removed low-level features (e.g., part-of-speech) from the analysis because they contribute little to the interpretation of the PCA results. Moreover, we utilized ChatGPT (GPT-4o) to provide a more nuanced interpretation of our findings, as detailed in our response 1-6-1 to Comment 1-6.

Regarding the interpretation of the PC1s, we no longer observed this component in our current results. By subtracting predicted BOLD responses related to head motion and random embeddings, we focused our analysis on higher-level linguistic processing using GPT embeddings (as described in 2-3-1). This new approach eliminated our previous observation of high prediction accuracy in primary motor and auditory cortex, which had suggested that PC1 might represent turn-taking.

Comment 3-18: Many of the categories identified through the PC analysis do not seem immediately meaningful: “there was a preference for proper nouns, particularly names of universities or specific locations, with fewer morphemes” >> why should this be represented? What are the implications of such a result? This and other such findings are just stated without any discussion/elaboration.

Answer: We acknowledge that the interpretations of the PCA results in the previous manuscript were based on subjective criteria, and the discussion lacked depth and elaboration.

In the revised manuscript, we addressed this issue by semi-automating the interpretation process using ChatGPT, as noted in response to Comment 3-17, to reduce subjectivity. Additionally, we expanded the discussion to better contextualize our findings within the broader literature. Specifically, we highlight two key points: (i) the extension of traditional semantic representations for natural speech comprehension, and (ii) the previously overlooked significance of backchannels and fillers.

(i) Extensions of traditional semantic representations for natural speech comprehension:

“Our results potentially extend the seminal work of Huth and colleagues (Huth et al., 2016), which comprehensively mapped fine-grained semantic representations during natural speech listening using word embeddings. Specifically, that study identified the first principal component differentiating between “*humans and social interaction*” and “*perceptual descriptions, quantitative descriptions, and setting*,” thereby separating social content from physical content. Our conversational data offered a unique opportunity to examine the semantic space surrounding social words in greater depth.” (lines 402–408, pages 18–19)

(ii) Previously overlooked significance of backchannels and fillers:

“Notably, our identified principal components reflected social interaction nuances, such as backchannels, confirmations, and fillers. These elements require minimal cognitive effort yet are vital for maintaining conversational flow (Clark and Brennan, 1991; Fox Tree, 2001, Clark and Fox Tree, 2002, Knudsen et al., 2020).” (lines 408–410, page 19)

By expanding on these two points, we provide a more objective interpretation of the PCA results, grounding our findings in existing research and addressing the implications of previously underappreciated aspects of interactive language, such as backchannels and fillers.

Comment 3-19: The idea of speaker- vs. listener-specific semantic interpretations needs more discussion. What does it mean, theoretically, if participants represent the same content differently when they speak it/ listen to it? How does this relate to current proposals about the functional architecture of language and meaning, most of which assume that the meaning representations are shared between comprehension and production?

Answer: We revised our manuscript to clarify these points and reflected this comment in Introduction and Discussion. As noted in the Introduction, prior research on neural processing of language has primarily focused on comprehension:

“The neural underpinnings of language processing have been extensively investigated using functional MRI (fMRI), particularly in the context of naturalistic narrative comprehension. Comprehension can be viewed as the transformation of low-level sensory inputs (e.g., speech sounds, written text) into high-level hierarchical linguistic structures (Leaner et al., 2011; Regev et al., 2013; de Heer et al., 2017; Deniz et al., 2019; Popham et al., 2021, Deniz et al., 2023; Caucheteux et al., 2023). These structures are created by integrating linguistic information—both semantic and syntactic—across multiple timescales, ranging from word-level semantics to sentence meaning, ultimately culminating in a coherent narrative (Leaner et al., 2011; Regev et al., 2013; de Heer et al., 2017; Deniz et al., 2019; Popham et al., 2021, Deniz et al., 2023; Caucheteux et al., 2023; Huth et al., 2016; Schrimpf et al., 2021; Goldstein et al., 2022; Chen et al., 2024). While these studies have highlighted the brain's capacity to encode hierarchical linguistic structures across various sensory modalities (e.g., listening, reading) and timescales (words, sentences, discourse, narrative arcs), the functioning of these mechanisms during real-time conversations remains largely unexplored.” (lines 42–50, page 2)

To discuss linguistic representations during conversation in this context, we referred to these studies. Previous research suggests that a hierarchical structure, from word to sentence to narrative meaning, is represented across overlapping cortical regions, irrespective of sensory modality (e.g., listening or reading).

Our revised manuscript proposes that this hierarchical structure, previously observed in comprehension studies, may differ in production. This is supported by the contrasting timescale tuning we found:

“Across participants, the context length that maximized prediction accuracy consistently varied between modalities, with production peaking at shorter timescales and comprehension at longer timescales (Fig. 3b). These findings suggest distinct timescale selectivity for production and comprehension.” (lines 190–193, page 9)

Moreover, previous studies comparing neural representations between production and comprehension have typically used context-free stimuli, such as isolated sentences. Recent research, however, suggests that such stimuli evoke weaker brain responses compared to natural narratives (Deniz et al., 2023). Since our study focuses on spontaneous language production in natural contexts, it may provide more reliable insights into how neural representations function in real-world communication.

Comment 3-20: The paper does not consider many relevant, recent references, which makes it difficult to situate the paper within the recent research discourse. We are providing a (non-exhaustive) list of relevant references below:

Answer: We have incorporated the suggested papers into our revised manuscript.

- ANN-to-brain modeling / disentangling the contribution of different features

- o Caucheteux, C., Gramfort, A., & King, J. R. (2021, July). Disentangling syntax and semantics in the brain with deep networks. In *International conference on machine learning* (pp. 1336-1348). PMLR.

Answer: In our previous manuscript, we described GPT embeddings represent semantics. However, it is more accurate to state that these embeddings capture both semantics and syntax. Understanding how both are represented in the brain is a key question in our research. This paper proposes a method to mathematically disentangle syntax and semantics, demonstrating that they are represented in overlapping regions of the association cortex. In our revised manuscript, we clarify that GPT encodes linguistic representations encompassing both semantics and syntax, and we reference this paper in the Introduction:

“Throughout the paper, the term linguistic encompasses both semantic and syntactic dimensions, reflecting their tight interdependence (Caucheteux et al., 2021; Oota et al., 2023).” (lines 65–67 on page 3)

- o Oota, S. R., Gupta, M., & Toneva, M. (2022). Joint processing of linguistic properties in brains and language models. *arXiv preprint arXiv:2212.08094*.

Answer: This paper is also similar to the approach of Caucheteux et al. (2021). This paper utilized BERT and GPT models to extract semantic and syntactic information and showed that removing such information reduces the prediction accuracy of brain activity. Importantly, they found that syntactic components contributed the most to this prediction accuracy. This finding suggests that GPT encodes linguistic representations that include both semantics and syntax, with syntax playing a key role in aligning brain activity with language models. We reference this paper in the Introduction:

“Throughout the paper, the term linguistic encompasses both semantic and syntactic dimensions, reflecting their tight interdependence (Caucheteux et al., 2021; Oota et al., 2023).” (lines 65–67 on page 3)

- o M. Schrimpf, I. A. Blank, G. Tuckute, C. Kauf, E. A. Hosseini, N. Kanwisher, J. B. Tenenbaum, and E. Fedorenko. The neural architecture of language: Integrative modeling converges on predictive processing. *Proceedings of the National Academy of Sciences*, 118(45), 2021.

Answer: This work comprehensively investigated the relationship between model quality and brain activity prediction accuracy by comparing multiple language comprehension datasets, various large language models, and behavioral indicators of language processing. The results demonstrated a correlation between the accuracy of next-word prediction and both brain activity prediction accuracy during language comprehension and behavioral data. This supports the idea that language comprehension involves predictive processing to extract semantic information. We reference this paper in the Introduction:

“These structures are created by integrating linguistic information—both semantic and syntactic—across multiple timescales, ranging from word-level semantics to sentence meaning, ultimately culminating in a coherent narrative (e.g., Schrimpf et al., 2021; etc.).” (lines 45–47, page 2)

- Instruction-tuning improves LLM-to-brain encoding performance

- o Aw, K. L., Montariol, S., Alkhamissi, B., Schrimpf, M., & Bosselut, A. (2023). Instruction-tuning Aligns LLMs to the Human Brain. *arXiv preprint arXiv:2312.00575*.

Answer: This work demonstrated that instruction-tuning for answering to given prompts improves their representation of world knowledge, contributing to enhanced brain activity prediction accuracy. The conversational GPT model used in our current study may similarly improve its representation of world

knowledge compared to ordinary GPT models. However, we found no significant improvement using instruction-tuned GPT model. We reference this paper in the Results:

“To ensure that these findings were not influenced by the instruction-tuned GPT model (Aw et al., 2023), we replicated the core analyses using a base GPT model before instruction tuning. The results were consistent, confirming that instruction tuning did not affect the observed effects (Supplementary Fig. 17).” (lines 270–272, page 13)

- Production and comprehension load onto the same network

- o Menenti, L., Gierhan, S. M., Segaert, K., & Hagoort, P. (2011). Shared language: overlap and segregation of the neuronal infrastructure for speaking and listening revealed by functional MRI. *Psychological science*, 22(9), 1173-1182.

Answer: This work is a pioneering effort to explore the neural processing commonalities between language production and comprehension, particularly at higher semantic and syntactic levels. Using an fMRI adaptation paradigm, this work examined brain activity variations in response to prepared controlled stimuli. We reference this paper in the Introduction and Discussion:

“This hypothesis aligns with evidence of shared neural representations for semantic and syntactic information across modalities (Menenti et al., 2011; Hu et al., 2022; Giglio et al., 2022; Patel et al., 2023; Goldstein et al., 2023).” (lines 54–55, page 2)

“Previous neuroimaging studies, employing the within-subject approach, have manipulated the semantic and syntactic dimensions of stimuli to reveal shared representations (Menenti et al., 2011; Giglio et al., 2022; Hu et al., 2022; Patel et al., 2023).” (lines 338–339, page 16)

- o Hu, J., Small, H., Kean, H., Takahashi, A., Zekelman, L., Kleinman, D., ... & Fedorenko, E. (2023). Precision fMRI reveals that the language-selective network supports both phrase-structure building and lexical access during language production. *Cerebral Cortex*, 33(8), 4384-4404.

Answer: We also cited this work in our previous manuscript, and also refer it in our revised manuscript. This work examined whether the language network, which has been associated primarily with language comprehension, are also involved in language production. This work demonstrated that the language network is activated during language production as well, suggesting that both language production and comprehension are processed in overlapping brain regions. We refer this paper in the Introduction and Discussion:

“This hypothesis aligns with evidence of shared neural representations for semantic and syntactic information across modalities (Menenti et al., 2011; Hu et al., 2022; Giglio et al., 2022; Patel et al., 2023; Goldstein et al., 2023).” (lines 54–55, page 2)

“Previous neuroimaging studies, employing the within-subject approach, have manipulated the semantic and syntactic dimensions of stimuli to reveal shared representations (Menenti et al., 2011; Giglio et al., 2022; Hu et al., 2022; Patel et al., 2023).” (lines 338–339, page 16)

- fMRI responses to dialogue vs. monologue are similar

- o Olson, H. A., Chen, E. M., Lydic, K. O., & Saxe, R. R. (2023). Left-Hemisphere Cortical Language Regions Respond Equally to Observed Dialogue and Monologue. *Neurobiology of Language*, 1-36.

Answer: This work examined brain activity during third-person observation of conversations and monologues. This work found that comprehending conversations engages not only language-related areas but also theory of mind networks. These findings are discussed:

“In contrast, at longer timescales (16 to 32 seconds), spanning multiple conversational turns, shared representations were distributed across broader cortical regions, exhibiting notable inter-individual variability. Some participants (P1, P2, P4, P6, and P7) demonstrated shared representations extending into brain regions associated with the default mode network (DMN) and the theory of mind (ToM) network. The DMN has been implicated in representing higher-order discourse and narrative frameworks by integrating extrinsic information (e.g., utterances) with intrinsic information (i.e., prior context and memory) (Lerner et al., 2011; Yeshurun et al., 2017; Yeshurun et al., 2021). Similarly, the ToM network supports reasoning about others' mental states, a critical function in both language production and comprehension during conversations (Bašnáková et al., 2014; Paunov et al., 2022; Fedorenko et al., 2024). This network is particularly engaged in inferring the mental states of conversational partners, thereby facilitating pragmatic inferences about that particular individual (Bašnáková et al., 2014; Kuhlen et al., 2017; Olson et al., 2023). These findings suggest that shared representations at longer timescales support the integration of incoming conversational content with prior conversational context, as well as with broader social knowledge and beliefs. Such integration may support the formation of a psychological model of the situation, enabling inferences about the interlocutor's intended meaning (“speaker meaning”). Individual differences in the spatial distribution of these shared representations may reflect variability in discourse-level integration strategies.” (lines 362–377, page 17)

- **Neural basis of speech in everyday conversations**

- o Goldstein, A., Wang, H., Niekerken, L., Zada, Z., Aubrey, B., Sheffer, T., ... & Hasson, U. (2023). Deep speech-to-text models capture the neural basis of spontaneous speech in everyday conversations. *bioRxiv*, 2023-06.

Answer: This work recorded brain activity using ECoG during daily conversations and modeled the activity using a speech-to-text language model to identify speech and linguistic representations and compare them between production and comprehension. Although this work addressed similar questions to ours, it used different methodologies and devices. In contrast, our study uses fMRI to capture whole-brain signals, incorporating longer timescale contexts over tens of seconds, as opposed to the short-term transient responses recorded via ECoG. This allowed us to address issues that were not covered in this work. We refer this paper in the Discussion:

“Previous neuroimaging studies, employing the within-subject approach, have manipulated the semantic and syntactic dimensions of stimuli to reveal shared representations (Menenti et al., 2011; Hu et al., 2022; Giglio et al., 2022; Patel et al., 2023). Recent research utilizing spontaneously generated sentences and conversations has enhanced ecological validity (Nastase et al., 2020; Hamilton and Huth, 2020), uncovering shared semantic and syntactic representations during natural language use (Goldstein et al., 2023; Giglio et al., 2024). For instance, recent research (Goldstein et al., 2023) employed electrocorticography (ECoG) during natural conversations and modeled transient neural activity before and after word onset, identifying overlapping regions for word production and comprehension. However, two critical questions remain unresolved: (i) whether linguistic representations generalize across modalities and (ii) how these shared representations vary across timescales. Our study addresses these gaps, demonstrating the generalizability of shared representations and their modulation by the amount of contextual information.” (lines 338–347, page 16)

Comment 3-21: “By correlating the weights of the semantic model for production and comprehension (2,816 dimensions) for each voxel that exhibited significant predictions (Fig. 1d), we found positive correlations within the amodal regions ($P < 0.05$, two-sided, FDR corrected), including clusters in STG/STS (Fig. 4a and Extended Data Fig. 6). In contrast, negative correlations were observed throughout the cortex, both inside

and outside the amodal regions ($P < 0.05$, two-sided, FDR corrected), encompassing clusters in the CeS and SMA (Fig. 4b and Extended Data Fig. 7).”

o the statement “positive correlations within amodal regions” but negative correlations [...] inside [...] the amodal regions” is confusing; please clarify where positive/negative correlations were found

Answer: We agree that the description of brain regions with positive/negative correlations needs to be clearer.

3-21-1. Clarification of “amodal regions”:

In the previous manuscript, we used the term “amodal regions” without explicitly defining the specific voxels included in this category. To improve clarity, we have removed the term “amodal regions” from the revised manuscript.

3-21-2. Voxelwise modeling approach:

Voxelwise modeling allows us to examine the fine-grained spatial distribution of prediction accuracy at the voxel level. While many studies define regions of interest using functional localizers (e.g., Huth et al., 2016; de Heer et al., 2017), we did not employ localizer tasks. Instead, in line with other studies without localizers (e.g., Deniz et al., 2023; Chen et al., 2024), we visually defined the regions of interest based on coarse anatomical landmarks such as the prefrontal cortex (PFC), medial parietal cortex (MPC), auditory cortex (AC), and early visual cortex (EVC). These regions are consistently represented in all cortical surface maps included in the revised manuscript.

3-21-3. Weight correlations presented separately for cross-modal and bimodal voxels:

In the previous manuscript, we displayed positive and negative correlations across all voxels with significant prediction accuracy (Fig. 4). In the revised manuscript, we now present these correlations separately for cross-modal voxels (Fig. 2c, Supplementary Fig. 13) and bimodal voxels (Fig. 4e, Supplementary Fig. 16). This change provides a clearer visualization of how these correlations are distributed across different types of voxels.

Comment 3-22: Typos

- Page 2: listening to speech listening
- Page 13: First, both PC1 > should be “both PC1s”

Answer: Thank you for pointing out these typographical errors. We have corrected them in the revised manuscript.

Comment 3-23:

- The term “weights of the semantic model” is unclear/ambiguous throughout the manuscript. In the Methods, semantic model refers to GPT, but in certain contexts, the term seems to refer to the learned regression weights, such as here:
 - o By correlating the weights of the semantic model for production and comprehension (2,816 dimensions) for each voxel that exhibited significant predictions [...]

Answer: We carefully revised the use of the term “weights” in the manuscript to avoid ambiguity. Whenever we mention “weights,” we now clearly specify whether they refer to the Separate or Unified Linguistic models.

Reviewer #2 (Remarks to the Author):

First, I would like to thank the authors for revising the manuscript to systematically address for variance associated with head motion information. I am also happy to see the layer-by-layer representations and context-length analysis. In the following I have a few remaining comments:

Comment 1

It is very surprising that the without production condition produced more head motion. How do the authors explain this?

In Supplemental Fig. 2 due to the outliers, the differences between the medians are hard to compare. Could the authors create this figure without the red outlier circles.

It would also be helpful to compute the statistical difference in the individual subject level instead of group average.

Comment 2

I appreciate that the authors build a separate model where they remove the head motion artifacts, as from Supplemental Fig. 3 one can see that this model predicts well in language related areas (e.g. P2, P3, P5, P7, P8).

However, the description of how the motion parameters were considered needs a better explanation than the following sentence:

“The predicted timeseries were subtracted from the preprocessed fMRI data to isolate the neural signals of interest.”

Comment 3

The authors write in L617-618

“As a control to account for predictions potentially driven by low-level sensory or motor brain activity, we generated random normal embeddings with the same dimensionality as the GPT embeddings (2,816 features).”

Although this model is a neat model, and looking at Supplementary Fig 4 suggests that sensory regions are well predicted. I doubt it is a good model for low-level sensory information. There are remaining stimulus correlations that won't be addressed with such a random model.

Comment 4

Regarding the description about model fitting:

The authors write in the revised Results section (L676-679):

“We performed variance partitioning to quantify the unique contributions of linguistic features to BOLD responses in production, comprehension, and their intersection. Following methods from previous voxelwise modeling studies^{16,40}, we used three models: a production-only model, a comprehension-only model, and their combination (i.e., the Separate Linguistic model).”

So, did the authors run a simple regression model with only comprehension and only production data and the corresponding embeddings separately?

$$Y_{\text{comprehension}} = X_{\text{comprehension}} * w + \text{epsilon}$$
$$Y_{\text{production}} = X_{\text{production}} * w + \text{epsilon}$$

What do the authors mean by the following to statements:

- L685-686: “To maintain consistency with correlation coefficients, square roots of variance components were reported.”

- L686-688: “For TRs containing single (either production or comprehension) or overlapping conditions (production and comprehension), we applied variance partitioning consistently.”

Comment 5

L627-628: Following statement needs citations:

“A distance of 50 mm between the cerebral cortex and the head center was assumed in accordance with prior studies.”

Comment 6

EVC labels in all the Figures where the cortical flatmaps are shown are quite misplaced.

We sincerely thank the reviewers for their valuable comments. For ease of review, our responses are structured to address each reviewer comment in turn, with the reviewer comments presented in blue.

Comment 1

It is very surprising that the without production condition produced more head motion. How do the authors explain this?

In Supplemental Fig. 2 due to the outliers, the differences between the medians are hard to compare. Could the authors create this figure without the red outlier circles.

It would also be helpful to compute the statistical difference in the individual subject level instead of group average.

Answer: We thank the reviewer the cautious comment. We discovered an error in the initial calculation of head motion: the first scan of each run had mistakenly been assigned motion parameters from a different run. We have corrected this by assigning a value of zero to the first scan's head motion in each run.

The revised figure (below; Supplementary Fig. 2) reflects this correction. We have retained the outliers in the figure because they provide valuable information about the full distribution of motion values. With the correction applied, the the medians are now more clearly seen. Furthermore, we have added the effect sizes (Cohen's d) of the difference for each participant. These updates have been incorporated into the Supplementary Results:

“We assessed head motion using framewise displacement (FD; Power et al., 2012) and compared FD between scans conducted with and without speech production (Supplementary Fig. 2). At the individual level, six participants showed increased head motion during speech production, with effect sizes ranging from small (Cohen's $d < 0.3$; P1, P3, P5, P7) to medium ($0.3 < \text{Cohen's } d < 0.5$; P2, P6). In contrast, two participants (P4, P8) showed slightly greater head motion in the absence of speech production, both with small effect sizes (Cohen's $d < 0.3$). At the group level, no significant difference in head motion was observed between conditions (Mann-Whitney U test, $U = 40$, $P = 0.44$). The FD during speech production was 0.239 ± 0.047 mm (mean \pm standard deviation; range: 0.182–0.322 mm), compared to 0.219 ± 0.044 mm (range: 0.175–0.323 mm) in the absence of speech production.”—(Removing head motion artifacts in Supplementary Results)

Comment 2

I appreciate that the authors build a separate model where they remove the head motion artifacts, as from Supplemental Fig. 3 one can see that this model predicts well in language related areas (e.g. P2, P3, P5, P7, P8).

However, the description of how the motion parameters were considered needs a better explanation than the following sentence:

“The predicted timeseries were subtracted from the preprocessed fMRI data to isolate the neural signals of interest.”

Answer: As described in our response to Comment 1, we have updated the FD values and accordingly reanalyzed the Head Motion model:

“To minimize motion-related artifacts in the BOLD responses, we regressed out six rigid-body motion parameters along with FD. Model performance, evaluated by the Pearson correlation between predicted and observed timeseries, was low to medium across the cortex (mean \pm standard deviation of Fisher z-transformed correlation = 0.0566 ± 0.0099 ; range: 0.0386 – 0.0699 ; see **Supplementary Fig. 3** for individual cortical maps).” —(Removing head motion artifacts in Supplementary Results)

We have added interpretation of the results of the Head Motion model:

“These findings indicate that head motion can partially account for brain activity during conversation, with relatively high prediction accuracy in primary motor cortex. To isolate higher-level linguistic contributions directly related to conversational content, the motion-predicted timeseries was regressed out in subsequent analyses.” —(Removing head motion artifacts in Supplementary Results)

Comment 3

The authors write in L617-618

“As a control to account for predictions potentially driven by low-level sensory or motor brain activity, we generated random normal embeddings with the same dimensionality as the GPT embeddings (2,816 features).”

Although this model is a neat model, and looking at Supplementary Fig 4 suggests that sensory regions are well predicted. I doubt it is a good model for low-level sensory information. There are remaining stimulus correlations that won't be addressed with such a random model.

Answer: To address this concern, we have developed an additional low-level feature model that included acoustic and linguistic features such as syllable counts, morpheme counts, and cochleagram-based features. This model was applied to the residual BOLD response after regressing out the predictions from the random normal embedding model.

We found that the low-level feature model yielded minimal prediction accuracy across the cortex, suggesting that the random normal embedding model had already accounted for a substantial portion of the variance related to basic sensory and motor processes. Therefore, for the main analyses, we used BOLD responses after regressing out both head motion artifacts and predictions from the random normal embedding model.

We have added these results in “Removing low-level sensorimotor signals” in the Supplementary Results:

“To further validate that the random normal embeddings captured low-level sensorimotor signals, an additional low-level sensorimotor model was fit to the residual BOLD responses using explicit features. These included syllable and morpheme counts per utterance, as well as auditory features derived from a cochleagram based on Lyon's passive cochlea model (<https://github.com/theunissenlab/tlab/tree/master/AuditoryToolbox>), which employs logarithmically spaced filters with a bandwidth defined by $BW = \sqrt{cf^2 + ebf^2}/Q$, where cf is the characteristic frequency, ebf is the ear break frequency (set to 1,000 Hz), and Q is the quality factor (set to 8). This cochlear filter bank produced 80 frequency channels spanning 264–7,630 Hz, spaced at 25% of their respective bandwidths. To match the Nyquist frequency of fMRI data (0.5 Hz), we applied a low-pass filter followed by a Lanczos filter. The resulting low-level model yielded very low prediction accuracy (mean \pm standard deviation of Fisher z-transformed correlation = 0.0137 ± 0.0074 ; range: 0.0048 – 0.0268 ; see Supplementary Fig. 5 for individual cortical maps). These results confirm that the random normal embedding model had already accounted for a substantial portion of the variance related to basic sensory and motor processes. Therefore, for the main analyses, we used BOLD responses after regressing out both head motion artifacts and predictions from the random normal embedding model.”

Comment 4

Regarding the description about model fitting:

The authors write in the revised Results section (L676-679):

“We performed variance partitioning to quantify the unique contributions of linguistic features to BOLD responses in production, comprehension, and their intersection. Following methods from previous voxelwise modeling studies^{16,40}, we used three models: a production-only model, a comprehension-only model, and their combination (i.e., the Separate Linguistic model).”

So, did the authors run a simple regression model with only comprehension and only production data and the corresponding embeddings separately?

$Y_{\text{comprehension}} = X_{\text{comprehension}} * w + \text{epsilon}$

$Y_{\text{production}} = X_{\text{production}} * w + \text{epsilon}$

Answer: Yes, that is correct. We performed separate ridge regression analyses for production and comprehension, respectively.

What do the authors mean by the following to statements:

- L685-686: “To maintain consistency with correlation coefficients, square roots of variance components were reported.”

Answer: We have revised the Methods section to clarify this statement:

“While variance partitioning is typically reported using R-squared values, we report the square roots of these values to align with our primary evaluation metric—correlation coefficients—thereby facilitating direct comparison and consistent interpretation across all reported results.”

- L686-688: “For TRs containing single (either production or comprehension) or overlapping conditions (production and comprehension), we applied variance partitioning consistently.”

Answer: We have revised the Methods section to clarify this statement:

“In principle, variance partitioning assumes equal sample sizes across conditions. However, in our naturalistic dialogue experiment, individual fMRI frames (TRs) may correspond to production, comprehension, both, or neither, resulting in inherent unequal sample sizes across conditions. To preserve the ecological validity of the dataset and avoid imposing artificial constraints, we applied variance partitioning uniformly across all TRs.”

Comment 5

L627-628: Following statement needs citations:

“A distance of 50 mm between the cerebral cortex and the head center was assumed in accordance with prior studies.”

Answer: We appreciate the reviewer’s suggestion. Although this assumption is commonly used in the field, we have now added a citation to the original article that proposed this approach (Power et al., 2012, Neuroimage).

“A distance of 50 mm between the cerebral cortex and the head center was assumed in accordance with a prior study (Power et al., 2012, Neuroimage).”

Comment 6

EVC labels in all the Figures where the cortical flatmaps are shown are quite misplaced.

Answer: We thank the reviewer’s observation. In the revised figures, we have replaced the term “EVC” with “VC” (visual cortex), which more accurately reflects the region shown and avoids potential confusion.